# Multi-Accurate CATE is Robust to Unknown Covariate Shifts

**Christoph Kern**[*]                                              *christoph.kern@stat.uni-muenchen.de*

*Department of Statistics*
*Ludwig-Maximilians-University of Munich*
*Munich Center for Machine Learning (MCML)*

**Michael Kim**[*]                                                         *mpk@cs.cornell.edu*
*Department of Computer Science*
*Cornell University*

**Angela Zhou**[*]                                                         *zhoua@usc.edu*
*Department of Data Sciences and Operations*
*University of Southern California*

**Reviewed on OpenReview:** *https://openreview.net/forum?id=VOGlTb27ob*

## Abstract

Estimating heterogeneous treatment effects is important to tailor treatments to those individuals who would most likely benefit. However, conditional average treatment effect predictors may often be trained on one population but possibly deployed on different, possibly unknown populations. We use methodology for learning multi-accurate predictors to post-process CATE T-learners (differenced regressions) to become robust to unknown covariate shifts at the time of deployment. The method works in general for pseudo-outcome regression, such as the DR-learner. We show how this approach can combine (large) confounded observational and (smaller) randomized datasets by learning a confounded predictor from the observational dataset, and auditing for multi-accuracy on the randomized controlled trial. We show improvements in bias and mean squared error in simulations with increasingly larger covariate shift, and on a semi-synthetic case study of a parallel large observational study and smaller randomized controlled experiment. Overall, we establish a connection between methods developed for multi-distribution learning and achieve appealing desiderata (e.g. external validity) in causal inference and machine learning.

## 1 Introduction

Causal inference studies how to make the right decision, at the right time, for the right person. Extensive recent literature on *heterogeneous treatment effects*, also called conditional average treatment effects (CATE), studies the estimation of personalized causal effects, rather than only population-level average treatment effects. Estimating CATE can inform better triage of resources to those who most benefit in healthcare, social services, e-commerce, and many other domains.

In these consequential domains, many firms/decision-makers face treatment decisions where other firms also need to make the same decision, although perhaps each with slightly different data distributions. For example, problems of clinical risk prediction, such as risk of a heart disease or medication treatment guidelines, are

---

[*]Authors listed in alphabetical order.

shared widely across hospitals, but each has its own distribution of patients in addition to idiosyncratic reporting, testing, and treatment patterns that can hinder external validity (Caruana et al., 2015). Indeed, off-the-shelf, relatively simple clinical risk calculators developed on one population are often broadly deployed as a decision support tool in many locations, without the ability to share the originating individual-level data, or with data drift over time. In social settings, the Arnold Public Safety Assessment (PSA), trained on a proprietary dataset and used in hundreds of jurisdictions (Goel et al., 2021), is an example of a widely deployed tool. Its accompanying decision-making matrix is another example of a treatment recommendation rule made more widely available (Laura and John Arnold Foundation, 2016), which can introduce disparities and poor treatment efficacy (Zhou, 2024). Other examples include the design of algorithmic profiling in active labor market programs (Crépon and Van Den Berg, 2016; Bach et al., 2023; Körtner and Bonoli, 2023): many different jurisdictions run different active labor market programs, and policymakers face questions about how to learn from what works elsewhere and how to scale up programs across heterogeneous locations.

A key challenge in these settings is to certify valid predictive performance of personalized causal effects for unknown deployment settings, each of which could have a different *target* covariate distribution. For example, predictive risk calculators, such as those for chronic heart disease, learned on a specific population might induce biased estimation for different locales with different populations. As one example, the widely used Framingham risk score overestimates risk for Asian populations (Badawy et al., 2022). This problem is not limited to earlier risk scores, but also affects modern ones: a sepsis predictive risk score provided by Epic, a major healthcare IT provider, fell short in a study of external validity on another population (Habib et al., 2021).

External validity, generalizability, and transportability are also important questions for causal inference (Tipton, 2014; Tipton and Hartman, 2023; Bareinboim and Pearl, 2013). Heterogeneous causal effect estimates might also be similarly learned on one population, but made more widely available, hence vulnerable to unknown covariate shifts. Spini (2021) studies the potential impacts of shifts in population for generalizing results from the Oregon Health Insurance Experiment, while Shyr et al. (2024) studies potential shifts in effect heterogeneity across multiple cancer studies.

On the other hand, we do want to leverage predictive information when it is available. How can we develop methods for heterogeneous treatment effect estimation so that a new hospital, without its own large database or in-house machine learning team, is still assured guarantees of low predictive bias on its own population, that might differ in unknown ways from a proprietary risk score that does not publish the original data? Importantly, while some methods guarantee validity of prediction models for a *known* target distribution, our previous examples highlight important cases when we want to guarantee low prediction bias under *many unknown target distributions.*

In this paper, we show how methods from multi-accurate learning (Hébert-Johnson et al., 2018; Kim et al., 2019) can endow conditional average treatment effect estimation with robustness to unknown covariate shifts. Indeed, the problem of confounding itself is a covariate shift problem, from the treated or control population to some target population (Johansson et al., 2022). Multi-accurate learning is a powerful and flexible framework that, by ensuring low predictive bias over a test function class, is also robust to combinations of these covariate shifts: those induced by confounding *or* unknown covariate shifts in the target population. Although multi-calibrated and accurate learning originated from fairness motivations for ensuring calibration/low prediction bias over rich subgroups, in this work we show how the adversarial test functions in the formulation also confer broad robustness against covariate shift. To highlight this flexibility, we use multi-accurate calibration on an extremely small clinical trial to correct a predictor from an confounded observational study.

Though there is extensive work on establishing external validity and transportability of causal effects, most of this work assumes information about a target population. Drawing inspiration from Kim et al. (2022), which studied "universal adaptability" of averaged outcome estimators with bias robust to unknown covariate shifts, we learn *CATE* estimates that will maintain unbiased predictions under *unknown* target populations.

Although causal inference and machine learning has witnessed significant methodological innovation either in orthogonal/statistical learning or other machine learning adaptations (Kennedy, 2023; Nie and Wager, 2020; Chernozhukov et al., 2018; Wager and Athey, 2018; Shalit et al., 2017; Hill, 2011), to name just a few, multi-accurate learning (Kim et al., 2019) introduces a different methodological toolkit related to boosting/adversarial formulations of conditional moment conditions (Dikkala et al., 2020; Bennett and Kallus, 2023; Ghassami et al., 2022). We conduct a thorough empirical study comparing finite- and large-sample performance of multi-accurate learning and other causal machine learning techniques more specifically tailored for causal structure. To summarize, we find that multi-accurate methods grant additional robustness against unknown covariate shifts while being competitive with more advanced causal machine learning methods in finite-samples. There is a robustness-efficiency tradeoff: the latter methods are designed to exploit in-distribution efficiency, which multi-accurate learning "off-the-shelf" does not. Nonetheless, our work connects these two previously unrelated lines of work and shows how multi-accurate learning "off-the-shelf" can address the problem of robust CATE estimation. Multi-accurate learning reduces prediction bias from model misspecification, just as is required for conditional average treatment effect estimation.

In our thorough empirical study we find that our proposed multi-accurate T- and DR-learner perform well under unobserved covariate shift. Although our work does not suggest multi-accurate learning as a replacement for state-of-the-art causal machine-learning for in-distribution estimation, due to differences in variance reduction or hyperparameter selection subroutines, it does provide evidence that could inform further methodological improvements and variance reduction of multi-accurate learning for CATE estimation. In summary: Multi-accurate learning can be used "off-the-shelf" to post-process CATE estimates based on differenced outcome regressions to endow them with robustness to unknown covariate shift. Such post-processing can be appealing because it can work with only *black-box* access to predictors and original data. (Our earlier examples indicate many situations where only black-box access to predictors is available). Alternative approaches to robustness against unknown shifts, like distributionally robust optimization, could change the robust-optimal predictor to a risk-sensitive one rather than the true CATE, but multi-accurate learning does not.

The contributions of our work are the following.

- We propose multi-accurate post-processing of differenced-regression (T-learner) and pseudo-outcome (the doubly-robust score in the DR-learner) based CATE estimation to obtain unbiased prediction on unknown deployment populations. This approach can also flexibly adapt to a variety of covariate shifts from confounding to adversarial/unknown shifts: we illustrate by postprocessing a CATE estimator that combines large observational/small randomized data. We theoretically establish identification.

- In Proposition 3 we show that multi-accurate post-processing of simple CATE estimates (T-learner) with a *richer* test function class can approximate a less-multi-accurate/less-robust but more-advanced CATE estimator, i.e. a multi-accurate DR-learner under a *simpler* test function class. This establishes theoretically favorable estimation properties: that our method not only provides robust bias control, but is also comparable to (weaker implementations of) CATE estimators that pursue efficient estimation. This is an important contribution since the multi-accuracy framework alone only guarantees robust bias control.

- We show in extensive experiments with simulations and real-world observational and randomized data from the Women's Health Initiative how our approach achieves finite-sample gains in ensuring robust bias control (and correspondingly, MSE) under unknown distribution shifts. Hence we empirically verify our theoretical contributions.

In Section 2, we introduce the formal problem setup. In Section 3 we describe the background context and most closely related work to our methodological developments (other related work is discussed in Section 6). We develop and analyze our methodology in Section 4. In Section 5, we provide extensive empirical

comparisons in simulated data and a case study of the Women's Health Initiative parallel clinical trial and observational study.

## 2  Problem setup

We overview the problem setup and describe directly related prior work on multi-calibration/multi-accuracy. See Section 6 for discussion of other methodological approaches.

**Data.**  The dataset $\mathcal{D} = \{(X_i, T_i, Y_i(T_i))\}_{i=1}^n$ comprises of covariates, treatment $\in \{0,1\}$, and (potential) outcomes $Y(T)$.

In different applications it will satisfy different assumptions, so we will later define different variants of $\mathcal{D}$. We first assume it arises from a randomized controlled trial or observational study under the assumption of weak ignorability, so that the following assumption about selection on observables holds.

**Assumption 1** (Unconfoundedness (ignorability))**.**

$$\{Y(1), Y(0)\} \perp T \mid X$$

Assumption 1 is a generally untestable assumption that permits causal identification. For example, it holds in randomized trials by design, and in observational studies if the observed covariates are fully informative of selection into treatment. Later on, we will jointly consider access to both a large-scale observational study (with potential violations of unconfoundedness) and a small randomized trial.

We denote treatment-conditional outcome regressions, and the propensity score as:

$$\mu_t(x) = \mathrm{E}[Y \mid X = x, T = t], e_t(x) = P(T = t \mid X = x)$$

These are the typical so-called nuisance estimation functions used in common estimators. We use $\hat\mu, \hat e$ to denote estimates.

Throughout we also assume standard assumptions of consistency, SUTVA, and overlap.

**Assumption 2** (Consistency, SUTVA, and overlap)**.** *We assume that $Y_i = Y_i(T_i)$ (consistency and SUTVA), and that there exists $\nu > 0$ such that $\nu \le e_1(x) \le 1 - \nu$.*

Finally, we have a mild assumption that outcomes are bounded.

**Assumption 3** (Bounded outcomes)**.** *We assume that $|Y(1)|, |Y(0)| \le B$.*

Many prior works have also assumed bounded outcomes, including but not limited to Kennedy (2020); Dudík et al. (2014). Relaxing the assumption of bounded outcomes is a mild technicality: it is used in the convergence analysis of multi-accuracy as a sufficient but not necessary condition and as a condition for Chernoff inequalities, which could be replaced with stronger concentration inequalities under weaker assumptions. See also Devroye and Wise (1980); Gayraud (1997) for canonical estimators for estimating the support of a distribution.

**Estimands.**  The common estimand in causal inference is the *average treatment effect*, (ATE) $\mathrm{E}\,[Y(1) - Y(0)]$. In regimes with posited heterogeneous treatment effects, that are predictable given covariates $X$, a (functional) estimand of interest is the *conditional average treatment effect* (CATE)

$$\tau(X) = \mathrm{E}\,[Y(1) - Y(0) \mid X].$$

**Performance assessment.**  The convention for benchmarking estimation of CATE is the mean-squared error (MSE) with respect to the true $\tau(X)$ CATE function:

$$\mathrm{E}[(\hat\tau(X) - \tau(X))^2].$$

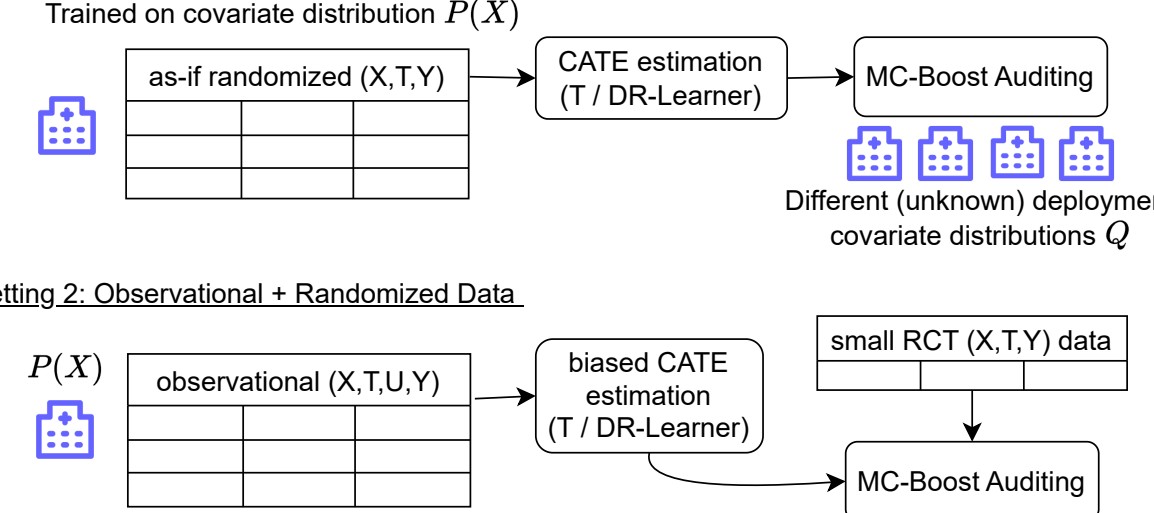

Figure 1: Schematic of setting 1 (external shift), and setting 2 (learning from large observational and small RCT data). We propose multi-accuracy (MC-Boost) auditing as an "off-the-shelf" procedure to improve the downstream robustness of CATE learners to unknown covariate shifts in both settings.

Further, estimators for CATE will all eventually involve different regressions that implicitly minimize predictive error marginalized over the dataset's distribution of $X \sim P_X$.

Later on, our work will focus on providing guarantees on conditional bias achieved by a CATE estimate $\hat{\tau}$ marginalized under a target covariate distribution $X \sim Q_X$ that can be *different* from the distribution $X \sim P$ upon which the CATE estimate was trained:

$$|\mathrm{E}_Q[(\hat{\tau}(X) - \tau(X))]| \tag{1}$$

The bias is of course a component of the MSE: multi-accuracy methods provide guarantees on the absolute bias; later in Section 5 we extensively empirically evaluate the mean squared error as well.

We write $Q_X(x), P_X, P_{X_1}, P_{X_0}$ for the marginal distribution of $X$ under $Q$, the marginal distribution of $X$ under $P$, and $P_{X_1}, P_{X_0}$ under $X \mid T = 1, X \mid T = 0$ on the observed data, respectively. We also denote $E_P[\cdot], E_{P_1}[\cdot], E_{P_0}[\cdot]$ to denote marginalization over $X$ in the training data, $X \mid T = 1$, or $X \mid T = 0$, respectively. For brevity we write $E_Q[\cdot]$ to denote expectations under the *unknown target distribution* $Q_X$ on $X$ (which can be extended to accommodate shifts beyond the typical covariate shift assumption in a slight abuse of notation). For example, $\mu_t \in \arg\min_g E_{P_{X_t}}[(Y - g(X))^2]$, e.g. by default, regression in each treated arm minimizes the MSE under the covariate distribution of each treatment arm.

**Notation conventions.** When, for example describing the multi-accuracy criteria without reference to the dataset's distribution, we write $E[\cdot]$ when referring to the distribution of the training data.

We next introduce the shift scenarios (i.e. combinations of assumptions) under which we seek guarantees on CATE estimation. See Figure 1 for an informal illustration.

## 2.1 Robustness to unknown deployment shifts

**Setting 1** (Unknown *external covariate shifts*)**.** *Suppose Assumption 1, that unconfoundedness holds, and Assumption 2. Consider valid likelihood ratios with respect to the marginal distribution of $X$ in observational data, $P_X$:*

$$\mathcal{L}_1 := \left\{ \frac{dQ_X(x)}{dP_{X_1}(x)} : \mathbb{P}\left\{ \frac{dQ_X(x)}{dP_{X_1}(x)} < \infty, \forall x \ s.t. \ P(x) > 0 \right\} = 1, \mathrm{E}[\frac{dQ_X(X)}{dP_{X_1}(X)}] = 1 \right\}$$

$$\mathcal{L}_0 := \left\{ \frac{dQ_X(x)}{dP_{X_0}(x)} : \mathbb{P}\left\{ \frac{dQ_X(x)}{dP_{X_0}(x)} < \infty, \forall x \ s.t. \ P(x) > 0 \right\} = 1, \mathrm{E}[\frac{dQ_X(X)}{dP_{X_0}(X)}] = 1 \right\}$$

*We seek an estimator $\hat{\tau}(X)$ with low bias under $Q : |\mathrm{E}_Q[(\hat{\tau}(X) - \tau(X))]| \leq \epsilon$.*

$\mathcal{L}_1$ is the set of valid covariate shift likelihood ratios, such that $Q_X$ is absolutely continuous w.r.t $P_{X_1}$, i.e. the likelihood ratio is finite whenever $P_{X_1}$ has nonzero support (almost surely, with probability 1 under the measure $P_{X_1}$), and marginalizes to 1. We will call this type of unknown deployment shift an "external shift". This is analogous to the unknown shift setting studied in Kim et al. (2022), as well as other literature on unknown covariate shifts (Jeong and Namkoong, 2020; Subbaswamy et al., 2021; Hatt et al., 2021). In contrast to an extensive literature on transportability and external validity, we focus on the case of a-priori *unknown* deployment shifts.

If suitably nonparametric CATE estimation indeed recovered the Bayes-optimal predictor in finite samples, there would be no issue of unknown deployment shifts. But because in finite samples it generally does not, modifying estimation to protect against unknown deployment shifts can protect against misspecification and finite-sample issues. For example, misspecified CATE estimation is vulnerable to unknown covariate shift. Furthermore, the conventional mean-squared error *MSE* can be nonzero for the *Bayes-optimal* predictor $\mu_1(X) = \mathrm{E}[Y(1) \mid X]$. If the conditional bias or variance in $Y$ is heteroskedastic (i.e. varies in $x$), the *prediction MSE* changes as external shifts change the marginalizing covariate distribution. Under misspecification and nontrivial residual variance of the Bayes regressor, different weighting functions can indeed change the population minimizer (Shimodaira, 2000; Buja et al., 2019). Later on we will use multi-accurate learning to post-process CATE estimates to ensure robustness against covariate shifts represented by a function class of likelihood ratios.

## 2.2 Unobserved confounding: observational data with RCT

We consider a different setting where unknown covariate shifts may arise: a large observational dataset and small randomized trial. The observational study may be subject to unobserved confounding. On the other hand, the sample size of the randomized data may be small, so that learning conditional causal effects solely from randomized data is unsupported. This regime is common in clinical settings, such as the parallel Women's Health Initiative observational study and clinical trial (Machens and Schmidt-Gollwitzer, 2003); see also (Colnet et al., 2020; Yang et al., 2020) and (Bareinboim and Pearl, 2013) for identification results for the related setting of data fusion.

The data setting is as follows. The observational dataset may have been collected under unobserved confounders, $\mathcal{D}_{obs}^* = (X, U, T, Y)$, but we only observe $\mathcal{D}_{obs} = (X, T, Y)$. Hence unbiased causal estimation is not possible from the observational dataset alone. On the other hand, we also have a randomized controlled study, $\mathcal{D}_{\mathrm{rct}} = (X_r, U_r, T_r, Y_r)$. We summarize our assumptions about the sample size, and shift regimes of observational/randomized data below. The punchline is that multi-accuracy provides robustness against these potentially unknown shifts, under assumptions of well-specification of the test function class. (All shifts are in the "causal", rather than "anti-causal" setting).

In the below setting, we aim to learn a valid CATE estimator $E[Y(1) - Y(0) \mid X]$ for the covariate distribution of the observational study or additional unknown covariate shifts.

**Setting 2** (Observational and randomized study)**.** *Assume Assumption 2. Suppose an observational dataset $\mathcal{D}_{obs}$, collected under violations of Assumption 1 (the observational data were collected under unobserved confounders), and a randomized dataset $\mathcal{D}_{rct}$, where Assumption 1 holds (the randomized data are unconfounded). For covariate shift, consider likelihood ratio functions with respect to a target population Q. Again we seek an estimator $\hat{\tau}(X)$ with low bias under $Q : |E_Q[(\hat{\tau}(X) - \tau(X))]| \leq \epsilon$.*

**Assumption 4** (Small RCT, large observational study)**.** $|\mathcal{D}_{obs}| \gg |\mathcal{D}_{rct}|$

Assumption 4 is not necessary for identification, but it describes the relevant regime where the method is helpful: if instead $|\mathcal{D}_{rct}| \gg |\mathcal{D}_{obs}|$, unbiased CATE estimation is possible from the randomized data alone.

## 3  Background and related work

**Conditional average treatment effect estimation.**  We briefly discuss a few options for estimating $\tau$, upon which we will build multi-calibrated approaches in differing shift scenarios. The *T-learner* differences two regressions for the conditional means of Y for treated and untreated:

$$\hat{\tau}(x) = \hat{\mu}_1(x) - \hat{\mu}_0(x).$$

Implicitly in the definition of these methods, both of these basic approaches for CATE estimation learn predictive models $\mu_t(X)$ by minimizing the mean-squared error, evaluated over some distribution of covariates $X$. Namely, for the $T$-learner,

$$\mu_t(x) \in \arg\min_{g(x) \in \mathcal{G}} E_P[(Y - g(X))^2 \mid T = t], t \in \{0, 1\}$$

Although many other advanced machine learning and causal inference methods have been developed based on advanced estimating equations (Nie and Wager, 2020; Kennedy, 2023; Oprescu et al., 2019; Wager and Athey, 2018; Semenova and Chernozhukov, 2021), or other machine-learning adaptations (Shalit et al., 2017; Shi et al., 2019), we will first instantiate our post-processing method with the $T$-learner and describe how it can be used with more advanced methods based on pseudo-outcome regression (such as the *DR*-learner; Kennedy 2020; Semenova and Chernozhukov 2021).

Our meta-algorithm is based on post-processing CATE estimation (the $T$- or $DR$-learner) with algorithms for multi-calibration. Next, we describe multi-calibration and its prior use for universal adaptability.

**Universal Adaptability via Multicalibration.**  Recent work of Kim et al. (2022) introduced the concept of *universal adaptability.* Much work on inference under (external) covariate shift assumes that the shift is known at the time of estimation. Instead, universal adaptability builds a prediction function from a source dataset whose average outcome estimate incurs small bias on any downstream covariate distribution, within a broad class of unknown shifts. The work of Kim et al. (2022) establishes the feasibility of universal adaptability via a connection to the notions of multi-calibration/multi-accuracy, originally introduced in the literature on algorithmic fairness (Hébert-Johnson et al., 2018). Following this line of research, we show how multi-accuracy can be used, off-the-shelf, to address unknown (external and internal) shifts in the context of CATE.

The multi-calibration criterion was originally motivated to provide guarantees over a variety of subpopulations, such as valid calibration over arbitrary subgroups (Hébert-Johnson et al., 2018). The related, but somewhat weaker, notion of multi-accuracy ensures low prediction bias within arbitrary subgroups (Kim et al., 2019). Throughout this paper, we focus on multi-accuracy (although analogous results hold for the stronger criterion of multi-calibration).

**Definition 1** (Multi-accuracy). *For $c(X)$ in a set of functions $\mathcal{C}$, a predictor $\tilde{p} : \mathcal{X} \to [-B, B]$ is $(\mathcal{C}, \alpha)$ multi-accurate if*

$$\max_{c \in \mathcal{C}} | \operatorname{E}[(Y - \tilde{p}(X))c(X)]| \leq \alpha,$$

Interpretations depend on the specification of the function class $\mathcal{C}$. When $\mathcal{C}$ is a class of subgroup indicator functions, $\mathcal{C} = \{\mathbb{I}[x \in C] : C \in \tilde{C}\}$, with $\tilde{C}$ a set of subsets of $\mathcal{X}$, then the multi-accuracy criterion ensures low prediction bias over a rich set of subpopulations. The class $\tilde{C}$ could indicate sublevel sets of functions with a finite VC-dimension. For example, if $\mathcal{C}$ is the space of all decision trees of depth 4, it has a finite VC-dimension and can describe complex subpopulations.

A growing line of work has developed algorithms with guarantees to (approximately) satisfy multi-calibration and multi-accuracy criteria (Hébert-Johnson et al., 2018; Kim et al., 2019; Gopalan et al., 2022b; Pfisterer et al., 2021) via boosting. When initialized from scratch, multi-calibration/accuracy can be viewed as a learning algorithm, but it can also be used to post-process a given predictor, as we do in this paper. Our meta-algorithms leverage these existing algorithms for obtaining *multi-accurate* predictors by post-processing.

Specifically, we use the MCBoost algorithm (Pfisterer et al., 2021), pseudocode included in Algorithm 4. MCBoost (Pfisterer et al., 2021) takes as input a given *initial predictor $p$*, *test-function class $\mathcal{C}$*, *approximation parameter $\alpha$*, and *post-processing datasets $\mathcal{D}_{post}$* for calibration and validation. For historical reasons, because the codebase was developed initially for multi-calibration which requires discretization of the interval $[0, 1]$, we standardize the prediction range, under assumption 3. This pre-processing is completely without loss of generality. Later on in our meta-algorithms, to be concise we will refer to this as running $\text{MCBoost}(p, \mathcal{C}, \alpha, \mathcal{D}_{post})$. As a brief summary, MCBoost is a boosting procedure that proceeds via a series of *auditing steps*: given the initial predictor $p$, it then solves a least-squares problem over the calibration dataset to find a $c \in \mathcal{C}$ that maximizes correlation with the residuals. The process iterates until the total miscalibration or accuracy error drops below a stopping criterion. Next, we discuss the auditing step of this procedure in more detail.

**Auditing.** The auditing step solves a subproblem of finding a subgroup with worst-case multi-accuracy error (bias).

**Definition 2** (Multiaccuracy auditing). *Let $\alpha > 0$ and $\mathcal{H}$ be a test class of functions. Suppose $D_{post} \sim \mathcal{D}$ is a set of independent samples. A predictor $\tilde{p} : \mathcal{X} \to [-B, B]$ passes $(\alpha)$-multiaccuracy auditing if for $c^* \in \arg\min_{c \in \mathcal{C}} \operatorname{E}[((\tilde{p}(X) - Y) - h(X))^2]$,*

$$|\operatorname{E}[(Y - \tilde{p}(X))c^*(X)]| \leq \alpha.$$

In practice, evaluation of the multi-calibration or multi-accuracy criterion over discrete subgroups is implemented via regression, reminiscent of twicing (Tukey et al., 1977). "Twicing" is a one-step boosting procedure: given a predictor function $p(x)$, fit the residual errors $c^* \in \arg\min_{c \in \mathcal{C}} \operatorname{E}[((Y - p(X)) - c(X))^2]$ and return $p(x) + c^*(x)$ as the final prediction. (In the conditional moment literature, these audit test functions would be called instrument functions). That is, the algorithm often audits over real-valued functions $c(x) : \mathcal{X} \mapsto \mathbb{R}$. These can be connected to the subpopulation motivation by viewing $c(x) : \mathcal{X} \mapsto [0, 1]$ as a relaxation of indicator functions; and real-valued functions as a rescaling of the former. (Later on, we will relate the real-valued weight functions directly to IPW weight functions (i.e. Riesz representers) in causal inference estimators).

The MCBoost algorithm incorporates these worst-case subgroup error functions to improve the predictor. Given a predictor $p_k(x)$ at some iteration $k$ of the algorithm, the auditing step learns a test function $c$ that best correlates with the residual function $p_k(x) - y$. For auditing over regression prediction functions, for step $k$, this is a regression problem: the auditing step solves $\min_{c \in \mathcal{C}} \operatorname{E}[((Y - p_k(X)) - c(X))^2]$. The predictor is then updated with a multiplicative-weights update based on the worst-case test function $c^*(x)$, that is, the next predictor $p_{k+1}(x)$ is upweighted or downweighted based on the predicted error $c^*(x)$, $p_{k+1}(x) \propto e^{-\Delta_c \cdot c^*(x)/2} \cdot p_k(x)$, where $\Delta_c$ is the multi-accuracy error computed on a validation set, $\operatorname{E}_V[c^*(X)(Y - p_k(X))]$.

Therefore the MCBoost procedure adjusts the original predictor based on the predicted errors $(c^*(x))$, adjusting more strongly in regions of higher error, as well as if the predictor incurs higher error on average. Auditing and postprocessing occurs in a different held-out dataset: we will refer to this as the *post-processing dataset*, including both calibration and validation sets. If the multi-accuracy criterion is not met for this test function $c \in \mathcal{C}$, the algorithm takes a boosting step and adds a multiplicative update with this test function. If the multi-accuracy criterion is met, the algorithm terminates.

**Remark 1** (Relation to conditional moment restrictions). *A reader in causal inference or econometrics may notice connections to conditional moment formulations. We expect that our later analysis, which is focused on multi-calibration/accuracy algorithms, also hold for adversarial formulations of conditional moments. (For example, Greenfeld and Shalit (2020) observes that adversarial moment conditions, in their case HSIC for independence of residuals, imply robustness to covariate shift). Recent advances in machine learning for conditional moment equations (Dikkala et al., 2020; Bennett and Kallus, 2023; Ghassami et al., 2022) typically develop min-max estimation algorithms that are unstable in practice; besides, the theory of conditional moment restrictions typically identifies finite-dimensional parameters rather than entire functions like the CATE. (See Section 6 for more extensive discussion of related work.)*

**Multi-accuracy vs. multi-calibration.** Throughout this paper, we have focused on a weaker multi-accuracy criterion (Kim et al., 2019), which is nonetheless sufficient for the robustness guarantees we seek for external validity and robustness to shifts. On the other hand, the stronger criterion of multi-calibration is somewhat more often studied in the literature. In this paper, we *do not* investigate the *calibration* properties of heterogeneous treatment effects, focusing instead on *robustness to external shift*. On the other hand, these are not completely unrelated: with slightly different notions, Wald et al. (2021) studies connections between multi-domain calibration and out-of-distribution generalization (without a focus on causal estimation). Other papers have investigated calibration for heterogeneous treatment effects (Xu and Yadlowsky, 2022), including orthogonalized isotonic regression (Van Der Laan et al., 2023).

## 4 Method

In this section, we develop and analyze our meta-algorithm that uses multi-accurate post-processing on the black-box regression models for a $T$-learner of the CATE. We first describe the meta-algorithm. To justify its use, we start with basic results focusing on ATE estimation. First we show how regression adjustment with multi-accurate outcome models approximates an augmented doubly-robust estimator (AIPW), in the *absence* of unobserved confounding. (This relates to estimation properties of our final approach). Second, we discuss the *target-independent identification* properties of ATE estimation, under the potential presence of unobserved confounding, which relates to our guarantees under unknown covariate shifts.

Identification implies we can write our target causal estimand in terms of functionals of the observational distribution alone. Estimating the regression-adjustment-identified functional with *multi-accurate post-processing* will result in certain equivalences with *other* estimation approaches, such as double robustness, with corresponding function classes. We establish results for the different settings we consider by combining these properties.

We then discuss the specific settings of unknown covariate shifts (external validity), observational/randomized data, and CATE estimation therein. Finally, a natural question remains: we established the properties of a multi-accurate $T$-learner, but what about more complex approaches? We provide extensions to recently studied pseudo-outcome regressions and prove approximate equivalence of a multi-accurate $T$-learner with a "weaker" multi-accurate $DR$-learner, suggesting that expanding the test function class can capture some of the improvements of a more complex estimation target.

---
**Algorithm 1** Multi-accuracy for CATE estimation for Setting 1, unknown covariate shifts
---
1: Input: $\mathcal{D} = \{(X_i, T_i, Y_i)\}_{i=1}^n$ unconfounded data, $\mathcal{F}$ auditor function class, $\mathcal{G}$ function class for outcome functions.
2: Split $\mathcal{D}$ into $\mathcal{D}_{est}$ and $\mathcal{D}_{post}$
3: Fit treatment-conditional outcome functions from the observational dataset $\mathcal{D}_{est}$:

$$\hat{\mu}_t(x) \leftarrow \arg\min_{g(x) \in \mathcal{G}} \mathrm{E}[(g(X) - Y)^2 \mid T = t], \text{ for } t \in \{0, 1\}$$

4: Post-process $\hat{\mu}_t(X)$ for $t \in \{0, 1\}$ by multi-accuracy:
   $\tilde{\mu}_t(x) \leftarrow \mathrm{MCBoost}(\hat{\mu}_t, \alpha, \mathcal{F}, \mathcal{D}_{post}^t)$, where $\mathcal{D}_{post}^t$ is the subset of $\mathcal{D}_{post}$ where $\mathbb{I}[T = t]$.
   so that $\max_{f \in \mathcal{F}} |\mathrm{E}_P[f(X) \cdot (Y - \tilde{\mu}(X)) \mid T = t]| \leq \alpha$.
5: Return $\tilde{\tau}(x) = \tilde{\mu}_1(x) - \tilde{\mu}_0(x)$
---

### 4.1 (Meta)-Algorithm: Multi-accurate post-processed T-learner

We describe the meta-learner in Algorithm 1. We learn CATE estimates based on the $T$-learner (i.e. differencing outcome regression models). We post-process the outcome models with multi-accuracy. Then the multi-accurate CATE estimate is:

$$\tilde{\tau}(X) = \tilde{\mu}_1(X) - \tilde{\mu}_0(X) \tag{2}$$

It also admits a natural regression-adjustment estimate for the ATE:

$$\mathrm{E}[\tilde{\tau}(X)] = \mathrm{E}[\tilde{\mu}_1(X) - \tilde{\mu}_0(X)]. \tag{3}$$

This represents estimating the ATE by imputing potential outcomes via regression: hence its name *regression adjustment* in the statistics literature, the "direct method" in the computer science literature, etc. Notably, the estimator *does not* explicitly use propensity scores. Nonetheless, we show how the robust test functions in multi-accurate auditing can nonetheless approximate inverse propensity score functions.

### 4.2 Warmup: Multicalibration, universal adaptability, and the ATE

We introduce properties of multi-calibration/multi-accuracy algorithms for estimation of the CATE and ATE. We begin by establishing that regression adjustment estimation with multi-accurate outcome predictors can approximate an AIPW estimator with certain nuisance functions. Later on, we combine these estimation properties with robust identification of the ATE and CATE under external covariate shifts.

**Robust estimation of the ATE under unconfoundedness: regression adjustment with sufficiently post-processed multi-accurate outcome predictors = doubly-robust estimation.** As a warm-up, we first consider estimation when unconfoundedness holds. We show in the causal context, that multi-calibration/multi-accuracy can be viewed as *finding a boosted predictor* whose marginalization satisfies estimating equations for the average treatment effect. In addition, multi-calibration/multi-accuracy as an algorithmic scheme *expands* the functional complexity of the original predictor it is initialized with. Interestingly, regression adjustment with a multi-calibrated/multi-accurate predictor approximates the doubly-robust estimator for the ATE, and hence is consistent if either the original predictor is well-specified, the inverse propensity score is within the auditor function class, *or* if the prediction function is within the *expanded* function class output by multi-calibration/multi-accuracy.

The doubly-robust augmented inverse-propensity weighting estimator (AIPW) is a canonical estimator highlighting improved estimation opportunities for causal inference (Robins et al., 1994). It has the following

form, for a given outcome and propensity model $\mu, e$:

$$\mathrm{E}[Y(1) - Y(0)] = \sum_{t \in \{0,1\}} \mathrm{E}\left[\frac{\mathbb{I}[T = t]}{e_t(X)}(Y - \mu_t(X)) + \mu_t(X)\right] \tag{4}$$

It enjoys improved estimation properties, such as the mixed-bias property (only requiring one of outcome or propensity model to be consistent for consistent estimation of the ATE) or rate double-robustness.

The approximation relies on conducting multi-accurate learning with an auditor function class containing the inverse propensity score. As a note, the below statements hold up to an additional misspecification error (as shown in Kim et al. (2022)). Because the auditor function class is typically large (i.e. contains functions beyond the inverse propensity score), this is a "robust" way to conduct doubly-robust estimation.

**Proposition 1** (Multi-accuracy implies robust estimation of the ATE via regression adjustment). *Suppose Assumptions 1 to 3. Consider an auditor class $\mathcal{H}$ that is closed under affine transformation. Assume unconfoundedness holds. Consider the estimator $\mathrm{E}[\tilde{\tau}(X)]$ where $\tilde{\tau}(x)$ is the output of Algorithm 1 with auditor class $\mathcal{H}$, approximation parameter $\alpha$, initial outcome model estimators $\hat{\mu}_1(x), \hat{\mu}_0(x)$, and $\mathcal{D}_{post}$ from the same distribution as the data.*

*If at least one of the following is true: (1) the original outcome models $\hat{\mu}_1(x), \hat{\mu}_0(x)$ are consistent estimators, (2) $e_1(X)^{-1}, (1 - e_1(X))^{-1} \in \mathcal{H}$, or (3) if using multi-accuracy, the true $\mu_1(x), \mu_0(x)$ are in the linear span of $\mathcal{G} + conv(\mathcal{H})$, then*

$$\mathrm{E}\left[\tilde{\tau}(X)\right] = \mathrm{E}\left[\tilde{\mu}_1(X) - \tilde{\mu}_0(X)\right] = \mathrm{E}[Y(1) - Y(0)] + 2\alpha,$$

*i.e. we obtain $2\alpha$-consistent estimation of the ATE.*

Although identification of the ATE with doubly-robust estimator is standard, note that our estimator that uses multi-accurate outcome models *does not* explicitly estimate the propensity scores. Nonetheless, the novelty of Prop. 1 is that averaging multi-accurate outcome models, because multi-accuracy implies $\mathrm{E}[(Y - \tilde{\mu}_t(X))h(X)] \leq \epsilon, \forall h \in \mathcal{H}$ leads to direct approximation of a doubly-robust estimator with the *multi-accurate regression-based* estimator $\mathrm{E}\left[\tilde{\mu}_1(X) - \tilde{\mu}_0(X)\right]$, under conditions on how the test function class $\mathcal{H}$ relates to inverse propensity score functions. Because this is a novel perspective, we establish it for ATE estimation first before showing how it applies to the robust covariate shifts that motivate our method.

This proposition connects the use of multi-accurate estimation to doubly-robust estimates and therefore establishes variance reduction properties, which is important because the multi-accuracy criterion itself is characterized via bias reduction on subgroups alone, without directly discussing the mean-squared error or estimation variance.

Although the beneficial estimation properties of AIPW are well-established (Robins et al., 1994; Kennedy, 2016), recall that our multi-accurate T-learner was not explicitly constructed as an AIPW estimator. It is therefore an example of an outcome-regression based approach that nonetheless satisfies doubly-robust estimating equation properties, in the spirit of (Bang and Robins, 2005), but algorithmically different. This additional perspective on how multi-accurate learning may robustly approximate a doubly-robust estimator is *new* in our paper, beyond previous analysis of universal adaptability. It also relates in spirit to recent work studying connections between regression adjustment and IPW methods, as (Chattopadhyay and Zubizarreta, 2023) does for linear regression. We build on this characterization in Proposition 3 to relate our later developments to doubly-robust CATE estimators.

**Robust target-independent identification and estimation of the ATE under universal adaptability.** Next we show how estimation with multi-accurate learning can robustly identify and estimate the ATE under potential violations of unconfoundedness. This is a re-interpretation of "universal adaptability" in Kim et al. (2022) which studied missing data under unknown shifts, which directly implies robust identification for causal inference.

*If* we had known the true propensity score function $1/e_1(X, U)$, we would obtain identification with respect to the observable marginalization of $\mathrm{E}[1/e_1(X, U) \mid X]$:

$$\begin{aligned}
\mathrm{E}[Y(1)] &= \mathrm{E}[\mathrm{E}[Y\mathbb{I}[T=1] \mid X, U] \cdot \mathrm{E}[1/e_1^*(X, U) \mid X, U]] = \mathrm{E}[Y\mathbb{I}[T=1]\,\mathrm{E}[1/e_1(X, U) \mid X]] \\
&= \mathrm{E}[Y\mathbb{I}[T=1]W_1^*(X)]
\end{aligned} \tag{5}$$

where in the first equality we apply ignorability conditional on $(X, U)$ and iterated expectations to obtain identification via the observable marginalization of

$$W_1^*(X) \coloneqq \mathrm{E}[1/e_1(X, U) \mid X].$$

(Note that $1/e_1 \neq \mathrm{E}[1/e_1(X, U) \mid X]$ due to Jensen's inequality). Robust identification of the ATE follows from Equation (5), i.e. that $\mathcal{H}$ contains (approximately) $\mathrm{E}[1/e_1(X, U) \mid X]$. Essentially, assuming that the auditor function class contains the (unknown) identifying weight, we can re-interpret the multi-accurate criterion as an approximation of adversarial IPW. We summarize this in the following corollary.

**Corollary 1.** *Suppose Assumptions 2 and 3. Suppose that $W_1^*(X), W_0^*(X) \in \mathcal{H}$. Run Algorithm 1 up to $(\alpha)$-multiaccuracy on $\mathcal{D}$ (possibly with unobserved confounders) over auditor function class $\mathcal{H}$ and outcome function class $\mathcal{G}$ to obtain $\tilde{\tau}(X) = \tilde{\mu}_1(X) - \tilde{\mu}_0(X)$. Then*

$$|\mathrm{E}[\tilde{\tau}(X)] - \mathrm{E}[Y(1) - Y(0)]| \leq 2\alpha \tag{6}$$

The result follows from the multi-accuracy criterion, which implies that $|\mathrm{E}[\mathbb{I}[T=t]W_t^*(X)\,(Y - \tilde{\mu}_t(X))]| \leq \alpha$, which obtains identification as in eq. (5) and the triangle inequality.

Of course, we have not gained identification for free: we *cannot* verify the assumption that $W_1^*(x), W_0^*(x) \in \mathcal{H}$ from observational data alone, just as we *cannot* test the unconfoundedness assumption from data alone. However, multi-accuracy/multi-calibration methods already work with quite flexible function classes, which could be nonparametric (RKHS, etc).

This is how multi-accuracy confers general robustness to distribution shift, whether from the data generating process such as unobserved confounders, or from external covariate shifts at the time of deployment.

### 4.3 External validity: unknown deployment shift

**Identification under Setting 1** The robust identification argument for "universal adaptability" re-interprets the test functions $c(X) \in \mathcal{C}$ as potential adversarial likelihood ratios for distribution shift.

However, the same properties of multi-accuracy also imply robustness to external shift. In this subsection, we indeed suppose Assumption 1, unconfoundedness. Recall that our goal was to control the predictive bias on a target covariate distribution $Q_X$, potentially unknown, $|\mathrm{E}_Q[(\hat{\tau}(X) - \tau(X))]|$. Note that each of $\mu_1, \mu_0$ are learned on a treatment conditional distribution, so we have that the valid likelihood ratio, which we denote $w_t(x)$, is defined as:

$$w_t(x) = \frac{dQ_X(x)}{dP_{X_t}(x)} = \frac{dQ_X(x)}{dP_{X_t}(x)} \frac{P(T=t)}{e_t(x)} \tag{7}$$

Obtaining robust identification for a "universally adaptable" CATE function instead interprets adversarial test functions as a product function class $\mathcal{F} = \mathcal{C} \times \mathcal{H}$ for *both* the subpopulations that identify CATE, *and* the adversarial likelihood ratio function. Our next proposition gives conditions on the weight functions $w_t \in \mathcal{H}, t \in \{0, 1\}$ to satisfy robust CATE estimation under unknown covariate shifts.

**Proposition 2.** *Suppose Assumptions 1 to 3. Let $\mathcal{C}$ denote a test function class for subgroup membership and $\mathcal{H}$ a test function class for likelihood ratios. Run Algorithm 1 for $\alpha$-multi-accurate postprocessing of the outcome models of a T-learner estimate. Then, for all target covariate distributions $Q$ such that the likelihood ratios $w_1, w_0 \in \mathcal{H}$,*

$$\forall c(x) \in \mathcal{C}, \ |\mathrm{E}_Q[\{\tilde{\tau}(X) - (Y(1) - Y(0))\}c(X)]| \leq 2\alpha.$$

---

**Algorithm 2** Multi-accuracy for CATE estimation for calibrating CATE on small Randomized Controlled Trial data

---

1: Input: $\mathcal{D}_{\mathrm{obs}} = (X, T, Y)$ confounded observational data, $\mathcal{D}_{\mathrm{rct}} = (X, T, Y)$ unconfounded randomized data, $\mathcal{F}$ auditor function class, $\mathcal{G}$ function class for outcome functions
2: Fit treatment-conditional outcome functions from the observational dataset:

$$\hat{\mu}_t(x) \leftarrow \arg \min_{g(x) \in \mathcal{G}} \mathrm{E}_{\mathrm{obs}}[(g(X) - Y)^2 \mid T = t], \text{ for } t \in \{0, 1\}$$

3: For $t \in \{0, 1\}$, use multi-accurate learning with $\mathcal{D}_{\mathrm{rct}}$ as validation set, i.e.
        $\tilde{\mu}_t(x) \leftarrow \mathrm{MCBoost}(\hat{\mu}_t(x), \mathcal{F}, \alpha, \mathcal{D}_{\mathrm{rct}})$.
4: Return $\tilde{\tau}(x) = \tilde{\mu}_1(x) - \tilde{\mu}_0(x)$

---

Because the guarantee holds for all functions $f \in \mathcal{F}$, it holds for complex subpopulations $c(X)$ and vacuous likelihood ratios with $h(X) = 1$, as well as the inverse: complex $h(X)$ and vacuous subgroups (i.e. $c(x) = 1$). Our assumption is that $\mathcal{F}$ is sufficiently well-specified to cover the product of these relevant functions, but we are generally agnostic as to the precise complexity of its constituent classes $\mathcal{C}, \mathcal{H}$. And, in practice, following the algorithmic implementation of MCBoost, we work with auditor function classes such as ridge regression, rather than direct products of subpopulations and other test functions.

Observe that although similar arguments apply, obtaining conditional guarantees for CATE estimation requires a richer test function class than for universal adaptability of the ATE alone. This illustrates that the case of learning CATE is indeed statistically harder than that of "universal adaptability" of the ATE. For CATE estimation, we need to choose a richer auditor function class than we would for ATE estimation.

### 4.4   Observational and randomized data (Setting 2)

**(Meta)-Algorithm.**   In this setting, we learn confounded outcome regressions from the observational data. We use the smaller randomized controlled trial data as *post-processing* datasets in MCBoost (the boosting paradigm for multi-calibrated and multi-accurate predictors). In Algorithm 2 we describe the meta-algorithm.

**Identification and estimation of CATE.**   Identification and estimation for the CATE follows by interpreting the auditing functions $c(X) \in \mathcal{C}$ as subpopulations. Achieving multi-accuracy on the RCT data hence identifies the CATE. That is, multi-accuracy assures us that

$$|\mathrm{E}[(Y - \mu_t(X))c(X) \mid T = t]| \leq \alpha, \; \forall c(x) \in \mathcal{C}$$

and we can evaluate this criterion on the unconfounded RCT data. On the unconfounded RCT data, we indeed have that $\mathrm{E}[Y \mid X, T = t] = \mathrm{E}[Y(t) \mid X]$ so that the $T$-learner identifies CATE.

The intuition for why our meta-algorithm improves upon directly running the $T$-learner on the randomized data alone is that we can learn a low-variance, high-bias (due to unobserved confounding) estimate of the true outcome model $\mathrm{E}[Y(t) \mid X]$ by outcome modeling on the observational data to obtain $\mathrm{E}_{\mathrm{obs}}[Y \mid T = 1, X]$. On the other hand, although randomized data is available, the finite-sample estimate of $\mathrm{E}_{\mathrm{rct}}[Y \mid T = 1, X]$ can be high-variance (though unbiased) under Assumption 4. We do note that the analysis of the boosting algorithm in Hébert-Johnson et al. (2018) is not tight enough to provably show faster convergence from warm-starting on the confounded regressions on the observational data, relative to multi-calibrating on the randomized data alone. However, we show benefits in later experiments.

**Identification of target-independent CATE.**   In complete analogy to the external shift setting, changing our interpretation of the target functions allows us to infer robustness to external shifts. Multi-accuracy ensures that, for all target covariate distributions $Q_X$ such that the likelihood ratios $w_t(x) \in \mathcal{H}$, $t \in \{0, 1\}$,

running Algorithm 2 with auditor function class $\mathcal{F} = \mathcal{C} \times \mathcal{H}$ results in a multi-accurate and deployment-shift robust CATE estimate:

$$\forall\, c(x) \in \mathcal{C}, \; |\mathrm{E}_Q[\{\tilde{\tau}(X) - (Y(1) - Y(0))\}c(X)]| \leq 2\alpha.$$

### 4.5 Extension to CATE pseudo-outcome regression

A natural question given our work on the $T$-learner is whether we can provide similar guarantees for an estimation-improved CATE learner, since the $T$-learner generally does not enjoy any improved estimation properties in causal inference. The causal inference and machine learning literature has developed many improved orthogonal/semiparametrically efficient procedures such as (but not limited to) the $R$-learner (Nie and Wager, 2020), $DR$-learner (Kennedy, 2023), or other machine-learning adaptations (Wager and Athey, 2018; Shalit et al., 2017).

Namely, some CATE estimation procedures give a pseudo-outcome $\psi(O; e, \mu)$, where $O$ denotes data tuples, i.e. $O = (X, T, Y)$, such that $\mathrm{E}[\psi(O; e, \mu) \mid X] = \tau(X)$. (It is designated as a pseudo-outcome because regressing upon it identifies the CATE or functional of interest, although it is not exactly an outcome itself). One such pseudo-outcome is the doubly-robust score. Pseudo-outcome regression of it as a CATE estimator was recently studied in Semenova and Chernozhukov (2021); Kennedy (2020).

$$\hat{\varphi}(O; \hat{e}, \hat{\mu}) = \frac{T - e_1(X)}{e_1(X)\{1 - e_1(X)\}}\,(Y - \hat{\mu}_T(X)) + \hat{\mu}_1(X) - \hat{\mu}_0(X) \tag{8}$$

Regressing upon pseudo-outcomes with favorable properties like orthogonality therefore confers these favorable properties to the estimated functional, such as improved statistical rates of convergence. Our arguments for external validity can naturally be extended for pseudo-outcome based CATE regression, so long as the pseudo-outcome's conditional expectation is the CATE function.

We multi-accurately postprocess the pseudo-outcome regression step. That is, we learn $\tilde{\tau}$ such that:

$$\mathrm{E}[\{\hat{\varphi}(O; \hat{e}, \hat{\mu}) - \tilde{\tau}(X)\}f(x)] \leq \epsilon, \forall f \in \mathcal{F}$$

Next, we instantiate such a procedure when the pseudo-outcome is the doubly-robust score.

**Multi-accurate DR-learner.** We give the algorithm for obtaining a multi-accurate $DR$-learner estimate in Algorithm 3. To summarize: we do need four folds of data $(\mathcal{D}_{1a}, \mathcal{D}_{1b}, \mathcal{D}_2, \mathcal{D}_3)$; the first three for sample-splitting of the nuisance estimates and pseudo-outcome evaluation and the last for validation/calibration for MCBoost. Estimate the nuisance functions on the first two folds $\mathcal{D}_{1a}, \mathcal{D}_{1b}$ and on $\mathcal{D}_2$, evaluate the pseudo-outcome value $\hat{\varphi}(O; \hat{e}, \hat{\mu})$ and regress $\hat{\tau}(x) = \hat{\mathrm{E}}_n[\hat{\varphi}(O; \hat{e}, \hat{\mu}) \mid X = x]$. Finally, we conduct post-processing via multi-accurate learning upon the $DR$-learner estimate $\hat{\tau}$, to obtain a multi-accurate $\tilde{\tau}$.

Again we will interpret the input auditor function class $\mathcal{F} = \mathcal{C} \times \mathcal{H}$ as a product function class of subgroup envelope functions $c \in \mathcal{C}$ and likelihood ratios $\in \mathcal{H}$. (Likelihood ratios are assumed to transport from the marginal distribution of $X$ to the new distribution). Then, (robust) identification of the predictions follows exactly as in Proposition 2.

Proposition 1 establishes that under specification assumptions, the multi-accurate regression adjustment estimator is (robustly) equivalent to the doubly-robust estimator up to $\epsilon$ approximation error, connecting multi-calibration with doubly-robust estimation. This implies basic (robust) doubly-robust properties of the multi-accurate $T$-learner. We now strengthen this connection by showing that multi-accurate post-processing of the $T$-learner over a *richer* function class (containing the true propensity score, and additional functions) implies that $\tilde{\mu}_t$ is also a *multi-accurate* estimate of the $DR$-learner over the additional functions.

**Proposition 3.** *Suppose Assumptions 1 to 3. Suppose that $\tilde{\mu}_t(x) \leftarrow MCBoost(\hat{\mu}_t, \alpha, \mathcal{F}, \mathcal{D}_{post})$, over auditor function class $\mathcal{F}$ such that $\overline{\mathcal{F}}_t \subseteq \mathcal{F}$, where $\overline{\mathcal{F}}_t = \left\{1, \frac{\mathbb{I}[T=t]}{e_t(X)}\right\} \times \mathcal{C} \times \mathcal{H}$. That is, $\tilde{\mu}_1 - \tilde{\mu}_0$ comprise an*

---

**Algorithm 3** Multi-accurate DR-learner (Equation (8)) for unknown covariate shift

---

1: Input: $(\mathcal{D}_{1a}, \mathcal{D}_{1b}, \mathcal{D}_2, \mathcal{D}_3)$ four independent samples of $n$ observations of $O_i = (X_i, T_i, Y_i)$ ($\mathcal{D}_3^n$ can be smaller). Auditor function class $\mathcal{F}$, approximation parameter $\alpha$.

2: Learn nuisance functions: Estimate propensity scores $e_t$ on $D_{1a}^n$. Estimate outcomes $(\hat{\mu}_0, \hat{\mu}_1)$ on $\mathcal{D}_{1b}$.

3: Pseudo-outcome regression: Construct the pseudo-outcome which takes as input observation $O = (X, A, Y)$ and nuisance functions $\hat{e}, \hat{\mu}$

$$\hat{\varphi}(O; \hat{e}, \hat{\mu}) = \frac{T - e_1(X)}{e_1(X)\{1 - e_1(X)\}}(Y - \hat{\mu}_T(X)) + \hat{\mu}_1(X) - \hat{\mu}_0(X)$$

and regress it on covariates $X$ in the test sample $\mathcal{D}_2$.

4: Post-process pseudo-outcome regression: run $\text{MCBoost}(\hat{\tau}_{dr}, \mathcal{F}, \alpha, \mathcal{D}_3)$ to obtain multi-accurate $\tilde{\tau}$ such that

$$\mathrm{E}[\{\hat{\varphi}(O; \hat{e}, \hat{\mu}) - \tilde{\tau}(X)\}f(X)] \le \epsilon, \forall f \in \mathcal{F}$$

5: Cross-fitting (optional)[1]

---

*$\alpha$-multi-accurate T learner. Then*

$$\left| \max_{ch \in \mathcal{C} \times \mathcal{H}} \mathrm{E}[\{\varphi(e, \tilde{\mu}; X) - \tau(X)\}c(X)h(X)] - \max_{ch \in \mathcal{C} \times \mathcal{H}} \mathrm{E}[\{\tilde{\tau}(X) - \tau(X)\}c(X)h(X)] \right| \le 2\alpha$$

*That is, the multi-accurate T-learner $\tilde{\mu}_1 - \tilde{\mu}_0$ is, up to $2\alpha$ additive approximation error, a multi-accurate DR-learner with outcome model $\tilde{\mu}$, post-processed over the function class $\mathcal{C} \times \mathcal{H}$.*

*Proof.* Consider a function class richer than that needed for Proposition 2. Define

$$\overline{\mathcal{F}}_t = \left\{ 1, \frac{\mathbb{I}[T = t]}{e_t(X)} \right\} \times \mathcal{C} \times \mathcal{H}$$

Consider a multi-accurate T-learner $\tilde{\tau} = \tilde{\mu}_1 - \tilde{\mu}_0$ where each $\tilde{\mu}_t$ is $\alpha$-multi-accurate over an auditor function class $\mathcal{F}_t$ so that $\overline{\mathcal{F}}_t \subset \mathcal{F}$.

Note that

$$\max_{c \times h \in \{\mathcal{C} \times \mathcal{H}\}} \left| \mathrm{E}\left[\left\{\left\{(Y - \tilde{\mu}_1(X))\frac{\mathbb{I}[T=1]}{e_1(X)} + (Y - \tilde{\mu}_0(X))\frac{\mathbb{I}[T=0]}{e_0(X)}\right\} + \tilde{\mu}_1(X) - \tilde{\mu}_0(X)\right\} - \tau(X)\}c(X)h(X)\right]\right|$$

$$- \max_{c \times h \in \{\mathcal{C} \times \mathcal{H}\}} \left| \mathrm{E}[\{\{\tilde{\mu}_1(X) - \tilde{\mu}_0(X)\} - \tau(X)\}c(X)h(X)]\right|$$

$$\le \sum_{t \in \{0,1\}} \max_{c \times h \in \{\mathcal{C} \times \mathcal{H}\}} \mathrm{E}\left[\left\{(Y - \tilde{\mu}_t(X))\frac{\mathbb{I}[T=t]}{e_t(X)}\right\}c(X)h(X)\right]$$

$$\le 2\alpha$$

by the triangle inequality and multi-accuracy of $\tilde{\mu}_t$ over the richer function class. $\square$

The interpretation is that post-processing a simple $T$-learner for multi-accuracy over a richer (yet well-specified) function class can approximate a $DR$-learner that was post-processed for multi-accuracy over a weaker function class. The population criterion for multi-accuracy confers some nonparametric robustness to bias over the specified test function class. Although this is a different estimation approach than causal machine learning estimates, we relate them formally here, and investigate empirically and thoroughly in Section 5. So, although multi-accurate post-processing of a $T$-learner appears on its face as a basic CATE estimator, in fact, the judicious choice of a richer function class for post-processing can approximate a more advanced estimator.

Interestingly, concurrent with the preparation of this work, Bruns-Smith et al. (2023) study augmented balancing weights and find a certain target-independent property of the augmented estimator related to the universal adaptability of Kim et al. (2022). Studying connections further would be an interesting direction for future work.

## 5 Experiments

We previously provided identification arguments and meta-algorithms for leveraging multi-accurate learning to learn CATE subject to unknown covariate shifts. To be sure, modern estimation of CATE prescribes nonparametric estimation that, in the infinite-data limit, is immune to external covariate shifts if CATE estimation recovers the Bayes-optimal predictor. Of course, real-world datasets are often smaller so that our methods can improve robustness in finite samples. To illustrate this, we conduct extensive empirical studies, testing our proposed multi-calibrated CATE estimation algorithms in comparison to other CATE learners in simulation scenarios that follow the previously introduced settings – unknown deployment shifts (Setting 1) and observational data with RCT (Setting 2).

### 5.1 Simulations

For both settings, we simulate data according to pre-specified propensity score functions, true outcome functions and external shift functions, with different degrees of complexity. For the external shift setting (simulation 1a and 1b), we assume access to training data from an observational study without unobserved confounding and a small auditing sample from the same distribution. The test data used for evaluation, however, is externally shifted by deliberately sampling with weights given by the shift function (and different shift intensities) from the original distribution. In this setting, we implement two simulations that differ in the complexity of the true CATE and propensity score functions (simulation 1a: linear CATE, beta confounding, simulation 1b: full linear CATE, logistic confounding). In the joint observational/RCT setting, we assume access to both a large observational training data set and a small RCT; and an external shift between both data sources. We implement two simulations in this setting that differ in whether both data sources (simulation 2a: confounded observational data and RCT) or only the observational training data (simulation 2b: total shift between observational data and RCT) are affected by unobserved confounding. The test data used for evaluation follows the covariate distribution of the observational training data. In simulation 2a, the problem is that of covariate shift alone; while in simulation 2b, the underlying conditional model (true $E[Y(t) \mid X]$ also changes. Simulation 2b illustrates the usefulness of the framework to simultaneously handle a variety of shifts. We present the simulation framework and comparisons to a broader set of baseline methods in supplementary material (D.2).

**Methods.** In both simulation settings, we use causal forests (CForest-OS) and random forest-based T-learner (T-learner-OS) and DR-learner (DR-learner-OS) trained in the observational training data as benchmark methods. T-learner-OS and DR-learner-OS also serve as the input for post-processing with MCBoost using ridge regression in the auditing data in simulation 1a and 1b (T-learner-MC-Ridge, DR-learner-MC-Ridge). In simulation 2a and 2b, post-processing is implemented with ridge regression-based auditing in the RCT data. In these simulations, we present causal forests trained in the RCT data (CForest-CT) as an additional baseline. To prevent overfitting with small auditing data, we regularized multi-accuracy boosting using small learning rates and a limited fixed number of boosting iterations (see Table 2 in D.2).

Comparing to T-learner-OS establishes the robustness benefits of our methods. On the other hand, our do-no-harm property holds with respect to the MSE of the best-in-class T-learner. Comparing to CForest-OS allows us to assess robustness-efficiency tradeoffs. CForest-OS is a representative state-of-the-art method that leverages the causal structure and modifies the estimation procedure of random forests; it is a very strong comparison point, but also very data-hungry. In contrast, our post-processing approach does not modify the estimation procedure. An interesting direction for future work is to achieve the robust bias guarantees of multi-calibration with other variance-reduced CATE estimators.

**Results.** We evaluate the outlined methods with respect to MSE of the estimated CATE in the test data in Figure 2 (external shift) and Figure 3 (observational data with RCT), over different sizes of initial training datasets and different intensities of covariate shift. The results of simulation 1a (Figure 2a) highlight how post-processing robustifies the initial T-learner and consistently improves over T-learner-OS in scenarios with moderate and strong external shift. When the observational training data is small, the multi-accurate T-learner also outperforms causal forests in these scenarios. With small training data, we see similar improvements of the multi-accurate DR-learner over DR-learner-OS. As the training data size increases, the naive DR-learner becomes more competitive and post-processing yields smaller gains.

In simulation 1b (Figure 2b), the more complex CATE function leads to higher MSE overall, while the previously observed pattern persists: The multi-accurate T-learner consistently improves over the naive T-learner, particularly under distribution shift. Our approach, DR-learner-MC-Ridge, is best in settings with strong unknown external shift and small dataset size: it then outperforms *both* T-learner and causal forest. Larger dataset sizes permit estimation over richer function classes and methods become asymptotically equivalent. We compare our approach to additional baselines, including shift-reweighted causal forests, T- and DR-learner in the supplementary material D.2.2.

Figure 3a shows that in simulation 2a, learning from both the observational training data and a small RCT via multi-accuracy boosting is beneficial across scenarios. The multi-accurate T-learner and DR-learner considerably improve over T-learner-OS and DR-learner-OS and in particular T-learner-MC-Ridge is competitive with CForest-OS. The improvement from multi-accuracy boosting can also be observed when post-processing was conducted with externally shifted RCT data and is similarly prominent for both T- and DR-learner in the "total shift" setting (Figure 3b). In both simulations, learning directly in the RCT data (CForest-CT) is only a viable option in the absence of a shift in the covariate distribution in the evaluation distribution (i.e. when only deploying on the smaller RCT population), and can incur considerable error otherwise. For results of additional CATE estimation techniques, see supplementary material D.2.3.

## 5.2 WHI data application

We next present a case study that draws on data from the Women's Health Initiative (WHI) studies (Machens and Schmidt-Gollwitzer, 2003). The WHI includes a large observational study and clinical trial data to investigate the effectiveness of hormone replacement therapy (HRT) in preventing the onset of chronic diseases. As the observational study has been suspected to suffer from various (unobserved) confounding phenomena (Kallus and Zhou, 2018), we study how utilizing both data sources in combination via multi-accuracy boosting compares to learning CATE from the observational or clinical trial data only. We focus on the effect of HRT treatment on systolic blood pressure and use age and ethnicity as covariates. Implementation details and results with an extended set of covariates is presented in the supplementary material D.3.

**Methods and Results.** We subsample the observational data to train causal forests (CForest-OS) and initial T-learner (T-learner-OS) and DR-learner (DR-learner-OS). We further sample from the clinical trial data to create CT training data with different sample sizes to post-process the initial T- and DR-learner with MCBoost using ridge regression (T-learner-MC-Ridge, DR-learner-MC-Ridge). We also train causal forests solely on the CT training data as an additional (strong) baseline (CForest-CT). Another sample from the CT data serves as the test set, with which we infer the (unobserved) "true" CATE using elastic net-based R-learner (Nie and Wager, 2020) and estimate the ATE as evaluation benchmarks.

We evaluate bias of the estimated ATE and MSE of the estimated CATE in Figure 4. Figure 4a shows how post-processing an initial T- and DR-learner with CT data can reduce bias, even if the auditing data is small. We similarly see improvements in MSE when comparing the multi-accurate learner to T-learner-OS and DR-learner-OS in Figure 4b. T-learner-MC-Ridge additionally improves over CForest-OS. Training only in the CT data leads to ATE estimates with low bias, but the MSE of CForest-CT is not competitive when model training is based on CT data with small sample sizes. Further results are presented in supplementary material D.3.

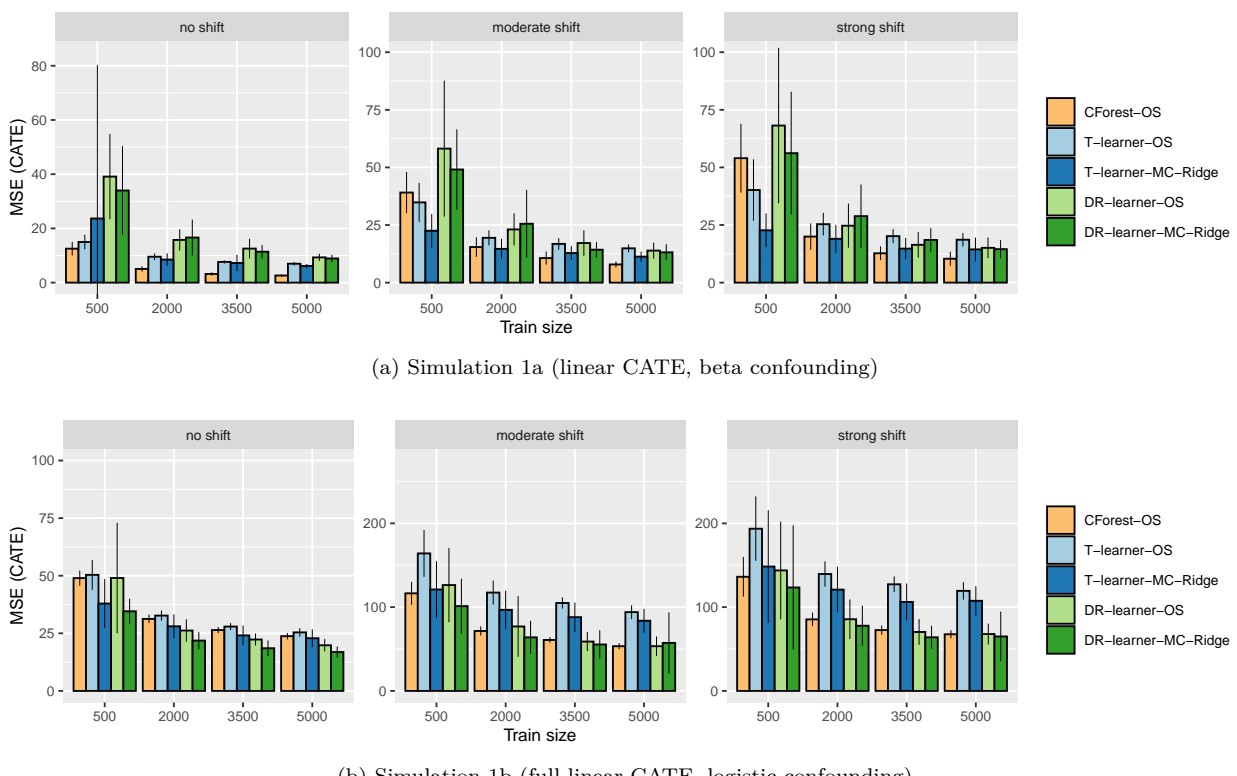

(a) Simulation 1a (linear CATE, beta confounding)

(b) Simulation 1b (full linear CATE, logistic confounding)

Figure 2: Average MSE of CATE estimation by shift intensity and training set size for post-processed (multi-calibrated) T- and DR-learner and benchmark methods in simulation studies (external shift setting).

## 6  Related work: further discussion

A popular approach for handling unknown shifts is to enforce robustness against a family of covariate shifts (e.g. unknown shifts parametrized by unknown covariate shift functions) (Liu and Ziebart, 2014; Wen et al., 2014; Chen et al., 2016). The goal is to find a robust hypothesis that maximizes the worst-case prediction risk (for example, squared error) evaluated with respect to unknown shifts within some class of covariate shifts. Parametrizations include distributionally robust optimization or linear basis functions. The work of Greenfeld and Shalit (2020) is motivated differently and penalizes with a Hilbert-Schmidt Independence Criterion loss; they show this implies some robustness to covariate shift. While most of this work is in the generic prediction setting, recent work also assesses ATE under covariate shift via distributionally robust optimization (Subbaswamy et al., 2021), use of the marginal sensitivity model for external shifts (Hatt et al., 2021), or variational characterizations of coherent risk measures (Jeong and Namkoong, 2020). Methodologically, some of this work is similar to work in causal inference that studies unobserved confounding under the lens of robust optimization adversarial likelihood ratios over some ambiguity set (Kallus et al., 2018a; Kallus and Zhou, 2021; 2020; Dorn et al., 2021; Zhao et al., 2019; Bruns-Smith and Zhou, 2023; Yadlowsky et al., 2018; Tan, 2006; Bruns-Smith and Zhou, 2023). This highlights the broad simultaneous interpretations of adversarial weight functions for handling unobserved confounding (in the generation of the data) in addition to robust adversarial covariate shifts (in the deployment of the predictor).

The approach based on multi-accuracy boosting, although it can be stated as a similar optimization problem in the abstract, differs from the previously mentioned works in a few important ways: (1) boosting couples the functional complexity of the post-processed predictor and the covariate shift function, and (2) under well-specification of the auditor function class and other conditions, boosting's asymptotic limit is the Bayes-

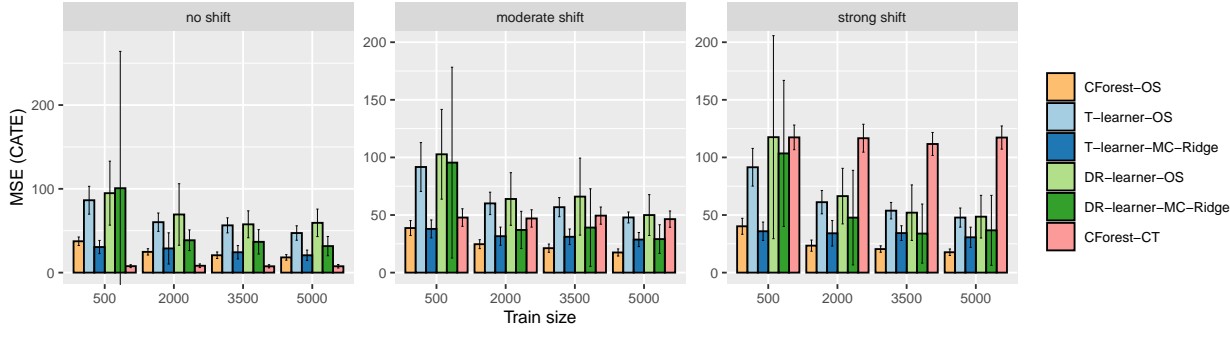

(a) Simulation 2a (confounded observational data and RCT)

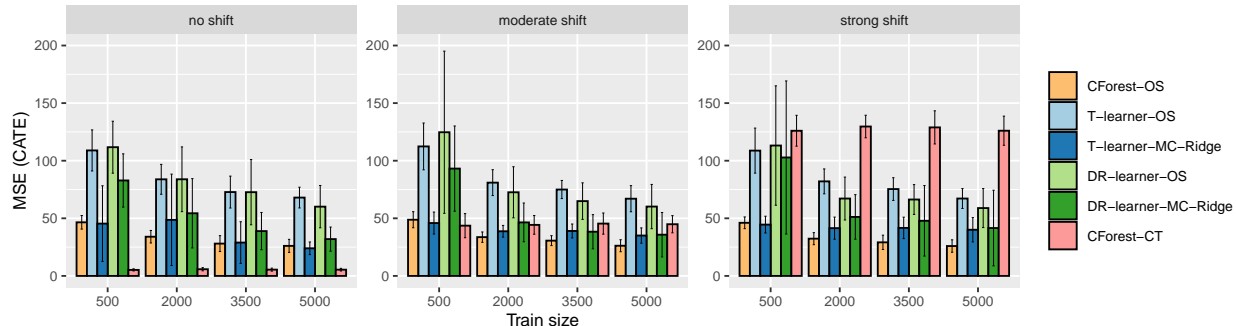

(b) Simulation 2b (total shift between observational data and RCT)

Figure 3: Average MSE of CATE estimation by shift intensity and training set size for post-processed (multi-calibrated) T- and DR-learner and benchmark methods in simulation studies (observational data with RCT setting).

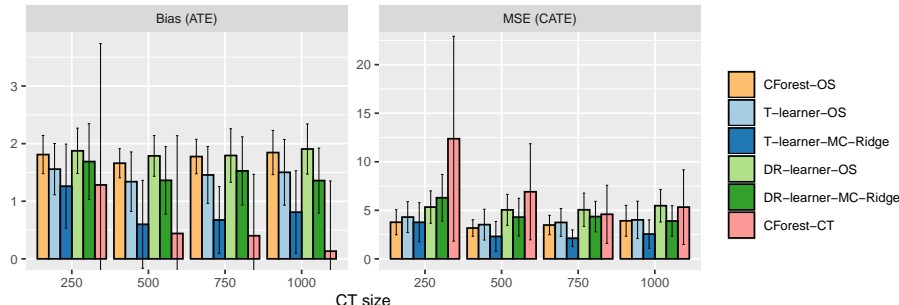

Figure 4: Average absolute bias and MSE by clinical trial sample size in WHI data application

optimal predictor, whereas robust optimization changes the asymptotic limit: typically to a coherent risk measure. In this sense, we expect that approaches based on multi-accuracy are *less conservative* within distribution. To the best of our knowledge, the only prior discussion of connections between boosting-style algorithms and distributionally robust optimization is Blanchet et al. (2019).

Approaches based on multi-calibration or multi-accuracy inherently couple the specification of the (expanded) hypothesis class of multi-calibrated predictors along with the specification of covariate shift functions, i.e. the boosting-type algorithm returns a predictor in the *sum class* of the original predictor and the classes of shifts. Approaches for robust covariate shift, to reduce the complexity of the adversary, require additional

moment constraints satisfied by valid likelihood ratios, i.e specifying a sharp set $\mathcal{C}$ of *only* valid covariate shifts. In robust optimization-based approaches to covariate shift, the hypothesis class and class of weight functions can be independently varied. But for multi-calibration, restricting the auditor function classes *also* simultaneously reduces the functional complexity of the hypothesis class of predictors. Distributionally robust objectives are equivalent to variance regularization or control of the tail risk, which couples statistically more difficult control of tail behavior with the control of ambiguous shift functions. Another important point of difference is that the Bayes-optimal predictor satisfies the multi-accuracy criterion, while a Bayes-optimal predictor with heteroskedastic noise may not satisfy desiderata of uniform performance implied by distributional robustness. For example, (Duchi and Namkoong, 2018, Example 2) discusses the example of linear well-specified models where the distributionally robust predictor coincides with the Bayes-optimal predictor; but in cases of model misspecification/heteroskedastic noise, this may not be the case. We leave a finer-grained comparison for alternative work.

Our discussion of the hybrid observational and randomized setting is more to highlight an "off-the-shelf" application of multi-accuracy, rather than the tightest analysis in this setting. Other works use more structure of this hybrid setting, or more heavily modify algorithms (i.e. learning shared representations) (Hatt et al., 2021; Yang et al., 2020; Kallus et al., 2018b); analogous adaptations with multi-accuracy are interesting directions for future work. See also Bareinboim and Pearl (2013) for a survey on transportability and external validity, Colnet et al. (2020) for a survey on learning from observational and randomized data. In this literature, Wu and Yang (2023; 2022) among others consider reweighting towards shifted distributions. Other approaches include Cheng et al. (2023).

## 7 Conclusion

In this work, we connect multi-accurate learning and conditional average treatment effect (CATE) estimation and show how off-the-shelf multi-accurate learning can be used for CATE estimation that is robust to unknown covariate shift. Although we empirically compare to more "state of the art" causal machine learning, these methods were designed for different purposes. Important directions for future work include "best-of-both-worlds" guarantees on *both* robustness and efficiency by improving variance reduction properties of Multi-CATE. A finer-grained analysis of the statistical implications of algorithmic implementations of boosting could also be relevant, in addition to improving hyperparameter tuning in the causal setting. In our work, we focus on establishing robustness properties.

**Acknowledgments** We thank the Simons Institute and the Simons Institute Program on Causality, where much of the work of this paper was conducted. AZ thanks the Foundations of Data Science Institute.

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

# A   Notation summary

| Notation/Object | Description |
| --- | --- |
| $X$ | Covariates |
| $T$ | Treatment, $T \in 0, 1$ |
| $Y(T)$ | Potential outcomes |
| $Y$ | Observed outcome, $Y = Y(T)$ |
| $U$ | Unobserved confounders |
| $e_t(x)$ | Propensity score, $P(T = t \mid X = x)$ |
| $\mu_t(x)$ | Outcome regression, $\mathbb{E}[Y \mid X = x, T = t]$ |
| $\tau(x)$ | Conditional average treatment effect (CATE), $\mathbb{E}[Y(1) - Y(0) \mid X = x]$ |
| $\mathcal{C}$ | Class of subsets of $X$ |
| $\mathcal{F}, \mathcal{G}, \mathcal{H}$ | Function classes |
| $\alpha$ | Multi-accuracy parameter |
| $\mathcal{D}_{\mathrm{obs}}$ | Observational dataset |
| $\mathcal{D}_{\mathrm{rct}}$ | Randomized controlled trial dataset |
| $\hat{\tau}(x)$ | Estimated CATE function |
| $\tilde{\tau}(x)$ | Multi-accurate/calibrated CATE estimator |

Table 1: Notation used in the paper.

# B   Details on algorithms

For completeness we describe the MCBoost algorithm for multi-calibration. See (Hébert-Johnson et al., 2018; Kim et al., 2019; 2022; Pfisterer et al., 2021) for more details, including theoretical analysis and implementation details. We describe the algorithm for a generic $(x, y)$ dataset (without reference to causal inference). See (Kim et al., 2019) for more details on the variant that achieves multi-accuracy (although ideas at a high level are similar.)

The key inputs include a regression algorithm for the boosting procedure, approximation parameter $\alpha$ which is a stopping condition (although in practice a finite limit on the number of iterations is used), and a validation/calibration set. When developing methods for Setting 1 (unknown covariate shifts), the calibration and validation set are drawn from the observational distribution. Our method for Setting 2 uses the (assumed small) RCT data as calibration/validations sets.

---
**Algorithm 4** MCBoost
---

**Given:**

$p_0 : \mathcal{X} \to [0, 1]$          `// initial predictor`
$\mathcal{A} : (\mathcal{X} \times [-1, 1])^m \to \mathcal{C}$      `// regression algorithm for functions in C`
$\alpha > 0$          `// approximation parameter`
$S = \{(x_1, y_1), (x_2, y_2), \ldots, (x_m, y_m)\}$      `// calibration set`
$V = \{(x_1, y_1), (x_2, y_2), \ldots, (x_v, y_v)\}$      `// validation set`

**Returns:**

$(\mathcal{C}, \alpha)$-multi-calibrated predictor $\tilde{\mu}$

Scale outcomes $y$ to $[0, 1]$

**Repeat:** $k = 0, 1, 2, \ldots$

$S_k \leftarrow \{(x_1, y_1 - p_k(x_1)), \ldots, (x_m, y_m - p_k(x_m))\}$      `// update labels in calibration set`
$c \leftarrow \mathcal{A}(S_k)$      `// regression over St`
$\Delta_c \leftarrow \frac{1}{|V|} \sum\limits_{(x,y) \in V} c(x) \cdot (y - p_k(x))$      `// compute miscalibration over V, validation set`
**if** $| \Delta_c | > \alpha$ **then**
     $p_{k+1}(x) \propto e^{-\Delta_c \cdot c(x)/2} \cdot p_k(x)$      `// multiplicative weights update`
**elsereturn** $\tilde{p} = p_k$ , rescaling the outcomes      `// return when miscalibration small`
**end if**

---

## C  Proofs

*Proof of Proposition 1.* (1a) Suppose $\hat{\mu}_1(x), \hat{\mu}_0(x)$ are consistent estimators and $e \in \mathcal{H}$. Then Equation (5) immediately implies $\epsilon$-consistency.

(1b) Suppose $\hat{\mu}_1(x), \hat{\mu}_0(x)$ are consistent estimators but $e \notin \mathcal{H}$. If $\hat{\mu}_1(x), \hat{\mu}_0(x)$ are consistent, they will asymptotically satisfy the multi-calibrated or multi-accurate criterion. See Hébert-Johnson et al. (2018) for related do-no-harm properties in this setting. Let $\mu_t^* = \mathrm{E}[Y \mid X, T = t]$ denote the true conditional expectation; it satisfies $\mu_t^* \in \arg\min \mathrm{E}[(Y - \mu_t(X))^2 \mid T = t]$ and that $\mathrm{E}[Y - \mu_t^*(X) \mid T = 1, X] = 0$, *a.s.* Hence $\forall f(X) \in \mathcal{F}$, $E[(Y - \mu_t^*(X))f(X) \mid T = 1] = 0$. Therefore $\mu_t^*(X)$ is feasible. Since the additive iterates of boosting approaches like MCBoost for multi-accuracy are commutative, (Gopalan et al., 2022a) characterizes multi-accuracy via a global optimization of squared loss over additive basis functions of $\mathcal{H}$. Since $\mu_t^*(X)$ is a optimal solution for the unconstrained problem, and feasible for the constrained problem, it is also optimal for the constrained problem.

(2) Suppose any of $\hat{\mu}_1, \hat{\mu}_0$ are not consistent estimators and $e_1^{-1}, (1 - e_1(X))^{-1} \in \mathcal{H}$. The implications of multi-accuracy with respect to $\mathcal{H}$ relate to the classical doubly-robust estimator:

$$\left| \sum_{t \in \{0,1\}} \mathrm{E}\left[ \frac{\mathbb{I}[T=t]}{e_t(X)}(Y - \tilde{\mu}_t(X)) + \tilde{\mu}_t(X) \right] - \mathrm{E}\left[ \tilde{\mu}_t(X) \right] \right| \leq 2\alpha \tag{9}$$

By properties of AIPW, the left hand term is consistent due to model double-robustness. By multi-accuracy, the CATE estimator is $2\alpha$-close to AIPW under well-specification.

(3) This follows via the same arguments given in (1b).

$\square$

## D Experiments

### D.1 Data and Software

We provide code of the simulation studies and the real data application for replication purposes in the following public OSF repository:

`https://osf.io/zxjvw/?view_only=a622c123414e4be6a218f121ded191d3`

Data preparations, model training and evaluation are conducted in `R` (3.6.3) (R Core Team, 2020) using the packages `ranger` (0.13.1) (Wright and Ziegler, 2017), `grf` (2.0.2) (Tibshirani et al., 2021) and `rlearner` (1.1.0) (Nie and Wager, 2020). The simulation studies heavily draw on the causal experiment simulator of the `causalToolbox` (0.0.2.000) (Künzel et al., 2019) package.

In all experiments, (initial) T-learner and DR-learner are post-processed using the MCBoost algorithm as implemented in the `mcboost` (0.4.2) (Pfisterer et al., 2021) package. More concretely, we make use of boosting for degree-2 multi-calibration, a (slightly) stronger notion than multi-accuracy, but computationally less demanding than full multi-calibration Gopalan et al. (2022b). The hyperparameter settings used for post-processing are listed as part of the following detailed presentation of the experiments (Table 2 and 13).

### D.2 Simulations

#### D.2.1 Setup

**Data** We follow the simulation setup of Künzel et al. (2019) in designing our experiments. Each of the following simulations is initialized by specifying the following components: Propensity score $e$, outcome functions $\mu_0^*$ and $\mu_1^*$, and external shift function $z$. We then simulate the following components:

- A 10-dimensional feature vector,

$$X_1, \ldots, X_{10} \sim \mathcal{N}(0, \Sigma)$$

  with modest correlations in $\Sigma$ (governed by `alpha` of the `vine` method (Lewandowski et al., 2009), which is set to 0.1).

- Potential outcomes are simulated according to the pre-specified covariate-conditional outcome functions $\mu_0^*$ and $\mu_1^*$,

$$Y_i(0) = \mu_0^*(x) + \varepsilon_i$$
$$Y_i(1) = \mu_1^*(x) + \varepsilon_i$$

  where $\varepsilon_i \sim \mathcal{N}(0, 1)$.

- Treatment assignment is simulated given the pre-specified propensity score $e$,

$$T_i \sim \text{Bern}(e_1(X))$$

  and the observed outcome is set to $Y_i = Y(T_i)$.

- A set of sampling weights is constructed given the external shift function $z$ (and shift intensity $s$),

$$w^{(s)}(x) = \left( \frac{z(x)}{1 - z(x)} \right)^s$$

  and used to simulate externally shifted observational data $\mathcal{D}_{os-shift}$ or shifted randomized control trial (RCT) data, $\mathcal{D}_{rct}$ (where $e_1(X) = 0.5$), depending on the simulation scenario.

We vary the shift intensity $s \in \{0, 0.25, \ldots, 2\}$ and training set size $\{500, 2000, 3500, 5000\}$, and run experiments for each combination 25 times. The size of the (audit/RCT) data used for multi-calibration boosting (500 observations) and the (test) data used for model evaluation (5000 observations) is fixed.

**Evaluation**  We compare and evaluate various techniques with respect to bias in ATE and MSE in CATE estimation. Bias is assessed based on the true ATE and the average of the estimated $\hat{\tau}(x)$ in the test data.

$$\text{Bias} = \tau - \frac{1}{n}\sum \hat{\tau}(x)$$

We further evaluate the true CATE $\tau(x)$ against $\hat{\tau}(x)$ of the respective CATE estimation method.

$$\text{MSE} = \frac{1}{n}\sum (\tau(x) - \hat{\tau}(x))^2$$

To quantify the external shift, we compute the multivariate Kullback-Leibler (KL) divergence (Kullback and Leibler, 1951) between the covariate distribution $(x_1 \ldots x_{10})$ of the training set (simulation 1a, 1b)/ the RCT (simulation 2a, 2b) $(\mathcal{N}_p^1)$ and the test set $(\mathcal{N}_p^0)$.

$$\mathrm{I}_{KL}(\mathcal{N}_p^0 \| \mathcal{N}_p^1) = \frac{1}{2}\left\{\mathrm{tr}(\boldsymbol{\Omega}_1\boldsymbol{\Sigma}_0) + (\boldsymbol{\mu}_1 - \boldsymbol{\mu}_0)^{\mathrm{T}}\boldsymbol{\Omega}_1(\boldsymbol{\mu}_1 - \boldsymbol{\mu}_0) - p - \ln|\boldsymbol{\Sigma}_0| + \ln|\boldsymbol{\Sigma}_1|\right\},$$

where $\boldsymbol{\Omega} = \boldsymbol{\Sigma}^{-1}$.

**MCBoost**  Multi-calibration boosting is conducted using the hyperparameter settings listed in Table 2. Hyperparameter settings of the baseline CATE learner are shown in Table 3.

### D.2.2  External Shift

In this initial setting, we simulate data that emulates an observational study with (observable) confounding. We additionally consider an external shift between the observational data that is available for initial model training, $\mathcal{D}_{os}$, and the distribution of the test (or deployment) data, $\mathcal{D}_{os-shift}$. We further assume access to an auditing sample from the original training distribution. The task is to estimate the true CATE function as evaluated the shifted test set, using models that either learned in the observational training data only or made additional use of the auditing data.

$$(X_{train}, T_{train}, Y_{train}) \sim \mathcal{D}_{os}, (X_{audit}, T_{audit}, Y_{audit}) \sim \mathcal{D}_{os}, (X_{test}, T_{test}, Y_{test}) \sim \mathcal{D}_{os-shift}$$

**Simulation 1a (external shift, linear CATE, beta confounding)**

$$\mu_0^*(x) = \begin{cases} x'\beta_l & \text{if } x_{10} < -0.4 \\ x'\beta_m & \text{if } -0.4 \leq x_{10} \leq 0.4 \\ x'\beta_u & \text{if } 0.4 < x_{10} \end{cases}$$

with $\beta_l \sim \mathrm{unif}([-5,5]^{10}), \beta_m \sim \mathrm{unif}([-5,5]^{10}), \beta_u \sim \mathrm{unif}([-5,5]^{10})$

$$\mu_1^*(x) = \mu_0^*(x) + 3x_1 + 5x_2$$

$$e_1(X) = \frac{1}{4}(1 + \mathcal{B}(x_1, 2, 4))$$

where $\mathcal{B}(x_1, 2, 4)$ is the beta distribution with parameters 2 and 4.

$$z(x) = \frac{1}{1 + e^{(-(x_1-0.5)-2(x_2-0.5)-0.5(x_1*x_2-0.5))}}$$

Table 2: Hyperparameter settings for post-processing using MCBoost. Default settings are used for parameters not listed.

(a) T-learner MC

| Method | Implementation | Hyperparameter | Value |
|--------|----------------|----------------|-------|
| Ridge | `mcboost` | `max_iter` | 5 |
| | | `alpha` | 1e-06 |
| | | `eta` | 0.5 |
| | | `weight_degree` | 2 |
| | `glmnet` | `alpha` | 0 |
| | | `s` | 1 |
| Tree | `mcboost` | `max_iter` | 5 |
| | | `alpha` | 1e-06 |
| | | `eta` | 0.5 |
| | | `weight_degree` | 2 |
| | `rpart` | `maxdepth` | 3 |

(b) DR-learner MC

| Method | Implementation | Hyperparameter | Value |
|--------|----------------|----------------|-------|
| Ridge | `mcboost` | `max_iter` | 5 |
| | | `alpha` | 1e-06 |
| | | `eta` | 0.1 |
| | | `weight_degree` | 2 |
| | `glmnet` | `alpha` | 0 |
| | | `s` | 1 |
| Tree | `mcboost` | `max_iter` | 5 |
| | | `alpha` | 1e-06 |
| | | `eta` | 0.1 |
| | | `weight_degree` | 2 |
| | `rpart` | `maxdepth` | 3 |

Note: `eta = 0.01` in simulation 2a and 2b (D.2.3).

Table 3: Hyperparameter settings of (baseline) CATE learners. Default settings are used for parameters not listed.

| Method | Implementation | Hyperparameter | Value |
|--------|----------------|----------------|-------|
| CForest | `grf` | `num.trees` | 2000 |
| | | `mtry` | `sqrt(p)+20` |
| | | `sample.fraction` | 0.5 |
| | | `honesty.fraction` | 0.5 |
| | | `min.node.size` | 5 |
| S-,T-, DR-learner | `ranger` | `num.trees` | 500 |
| | | `mtry` | `sqrt(p)` |
| | | `min.node.size` | 5 |
| | | `replace` | TRUE |
| | | `sample.fraction` | 1 |

**CATE estimation**   We use the following methods for estimating the CATE based on the observational training data. Shift-reweighting is conducted by training a logistic regression to predict sample membership in the observational training versus shifted test data and calculating propensity weights $\frac{1-\hat{p}}{\hat{p}}$ based on the predicted probability of membership in the training data $\hat{p}$.

- (**CForest-OS**) Causal forest (Wager and Athey, 2018) trained in the observational training data.

- (**CForest-wOS**) Causal forest trained in the shift-reweighted observational training data.

- (**S-learner-OS**) S-learner using random forest trained in the observational training data.

- (**S-learner-wOS**) S-learner using random forest trained in the shift-reweighted observational training data.

- (**DR-learner-OS**) DR-learner (Kennedy, 2023) using random forest trained in the observational training data.

- (**T-learner-OS**) T-learner using random forest trained in the observational training data.

- (**T-learner-wOS**) T-learner using random forest trained in the shift-reweighted observational training data.

We further estimate DR-learner and T-learner using multi-calibration boosting with a small set of auditing data.

- (**DR-learner-MC-Ridge**) DR-learner using random forest in the observational training data is post-processed with MCBoost using ridge regression in the auditing data.

- (**DR-learner-MC-Tree**) DR-learner using random forest in the observational training data is post-processed with MCBoost using decision trees in the auditing data.

- (**T-learner-MC-Ridge**) T-learner using random forest in the observational training data is post-processed with MCBoost using ridge regression in the auditing data.

- (**T-learner-MC-Tree**) T-learner using random forest in the observational training data is post-processed with MCBoost using decision trees in the auditing data.

**Evaluation**   We evaluate bias in ATE and MSE in CATE estimation in the externally shifted test data.

**Results**   We show the bias of the estimated average treatment effect (ATE) by shift intensity (column panels) and training set size (row panels) for each CATE estimation method in Figure 5 (see also Table 4 which includes quantifications of the distribution shifts using KL divergence). The results show that in the present setting all methods are able to produce unbiased estimates of the ATE in the non-shifted test data (first column). Introducing an external shift (second and third column), however, incurs bias across all methods with the shift-reweighted causal forest and shift-reweighted T-learner performing best. The ridge regression-based multi-accurate DR- and T-learner perform best among the shift-blind methods that had no access to the shifted test distribution.

We show the corresponding results for the MSE of the CATE estimation by shift intensity and training set size in Figure 6 (and Table 5). In the present setting, causal forest achieve the smallest MSE in the non-shifted test data as well as in settings with large initial training data (first column and third and last row). With increasing shift, however, ridge-based multi-accurate T-learner perform best in settings with small to moderately sized training data (upper right quadrant).

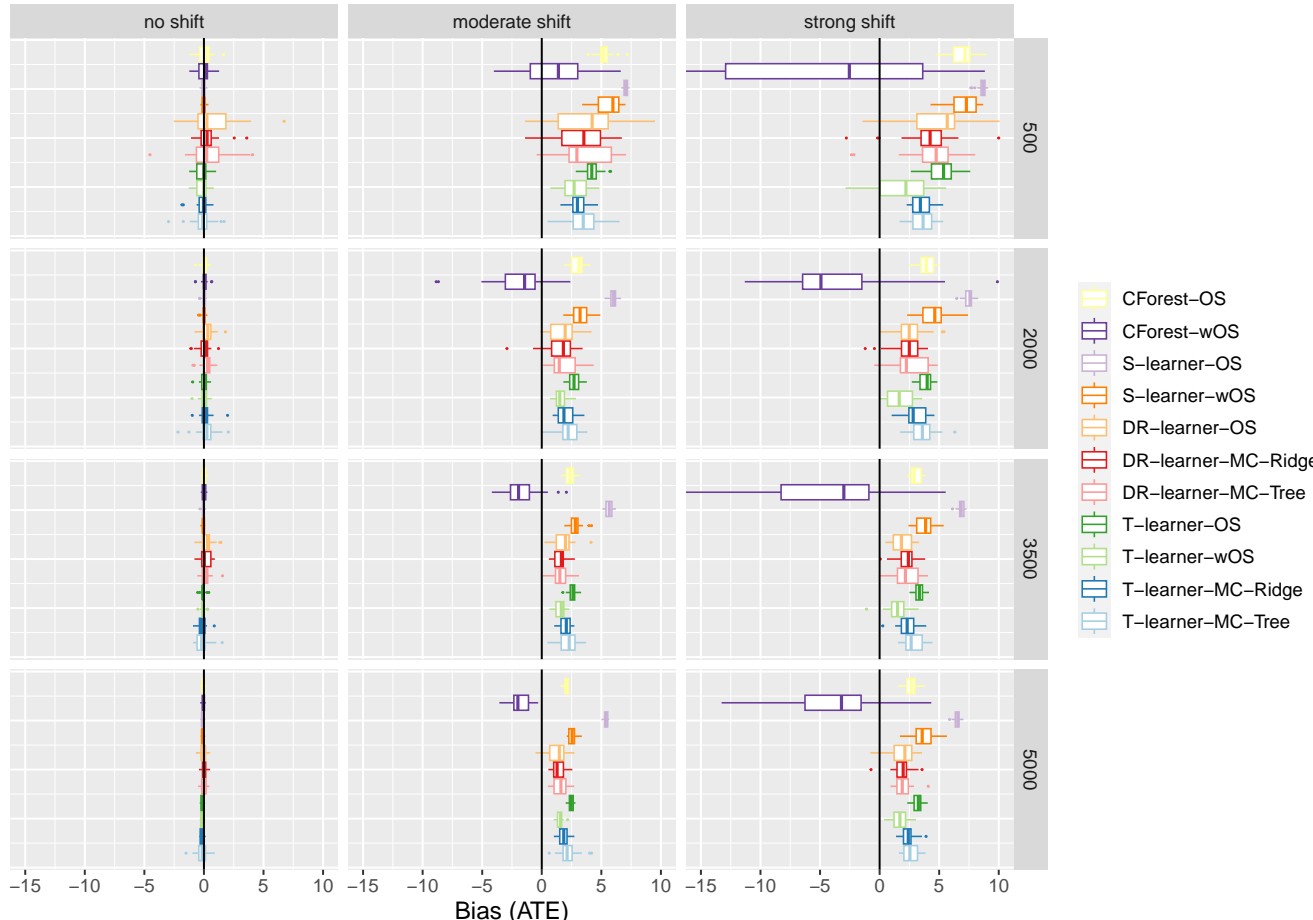

Figure 5: Bias of ATE estimation by shift intensity and training set size for different CATE estimation methods (Simulation 1a (external shift, linear CATE, beta confounding)). The distribution of bias scores over simulation runs is shown. Given an external shift between training and test data, DR-learner-MC-Ridge and T-learner-MC-Ridge perform best among the shift-blind methods that had no access to the shifted target distribution.

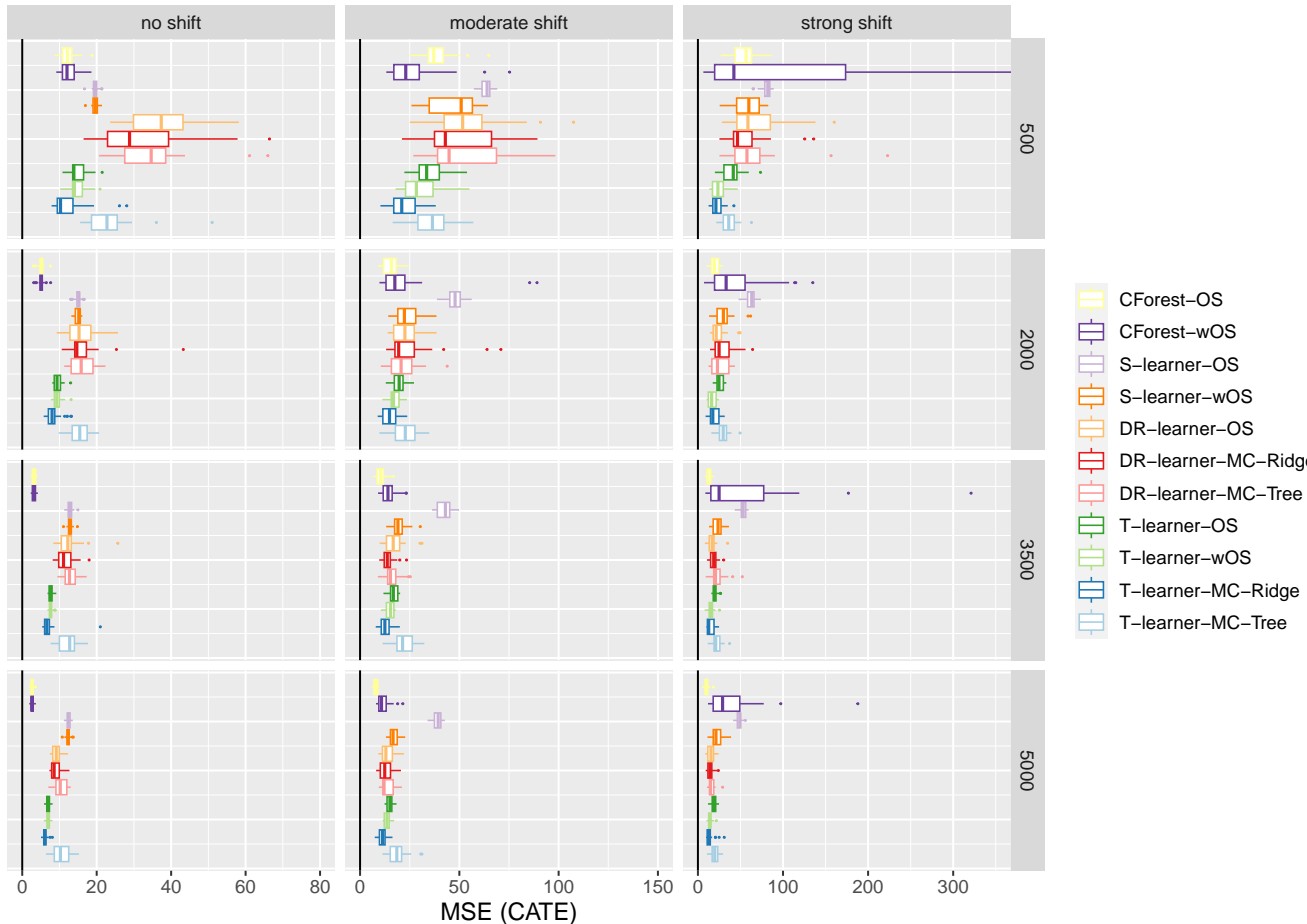

Figure 6: MSE of CATE estimation by shift intensity and training set size for different estimation methods (Simulation 1a (external shift, linear CATE, beta confounding)). The distribution of MSE scores over simulation runs is shown. T-learner-MC-Ridge performs best in settings with small to moderately sized training data and shifted test data.

**Simulation 1b (external shift, full linear CATE, logistic confounding)**

$$\mu_0^*(x) = 3x_1 + 5x_2$$

$$\mu_1^*(x) = \mu_0^*(x) + x'\beta, \, with \, \beta \sim \text{unif}([-5, 5]^{10})$$

$$e_1(X) = \frac{1}{1 + e^{(-2-2(x_1-0.5)-1(x_2-0.5))}}$$

$$z(x) = \frac{1}{1 + e^{(2(x_2-0.5)+(x_3-0.5))}}$$

**Evaluation**   Bias in ATE and MSE in CATE estimation is evaluated in the externally shifted test data.

**Results**   The results for bias of the ATE estimation in Figure 7 (and Table 6 including KL divergence) show that in absence of external shift (first column), causal forest-based estimators perform best and are able to achieve unbiasedness. Introducing an external shift between the observational training and test data (second and third columns) amplifies bias such that only the shift-reweighted causal forest is able to approximate the true ATE on average, given sufficient training data. The ridge regression-based multi-accurate DR-learner improve over the initial DR-learner and are competitive with shift-reweighted causal forest in settings with small initial training data and strong shift. Again note that, in contrast to the shift-reweighted methods, the multi-accurate learner had no access to the shifted test distribution during model training.

Figure 8 (and Table 7) shows results for the MSE of the estimated CATE. The ridge-based multi-accurate DR-learner consistently improve over the initial DR-learner and achieve the lowest MSE among all methods in the initial, non-shifted setting. With increasing shift, ridge-based multi-accurate and shift-reweighted learner perform well in settings with small to moderately sized training data. Causal forest is competitive in all settings and particularly as the training set size increases.

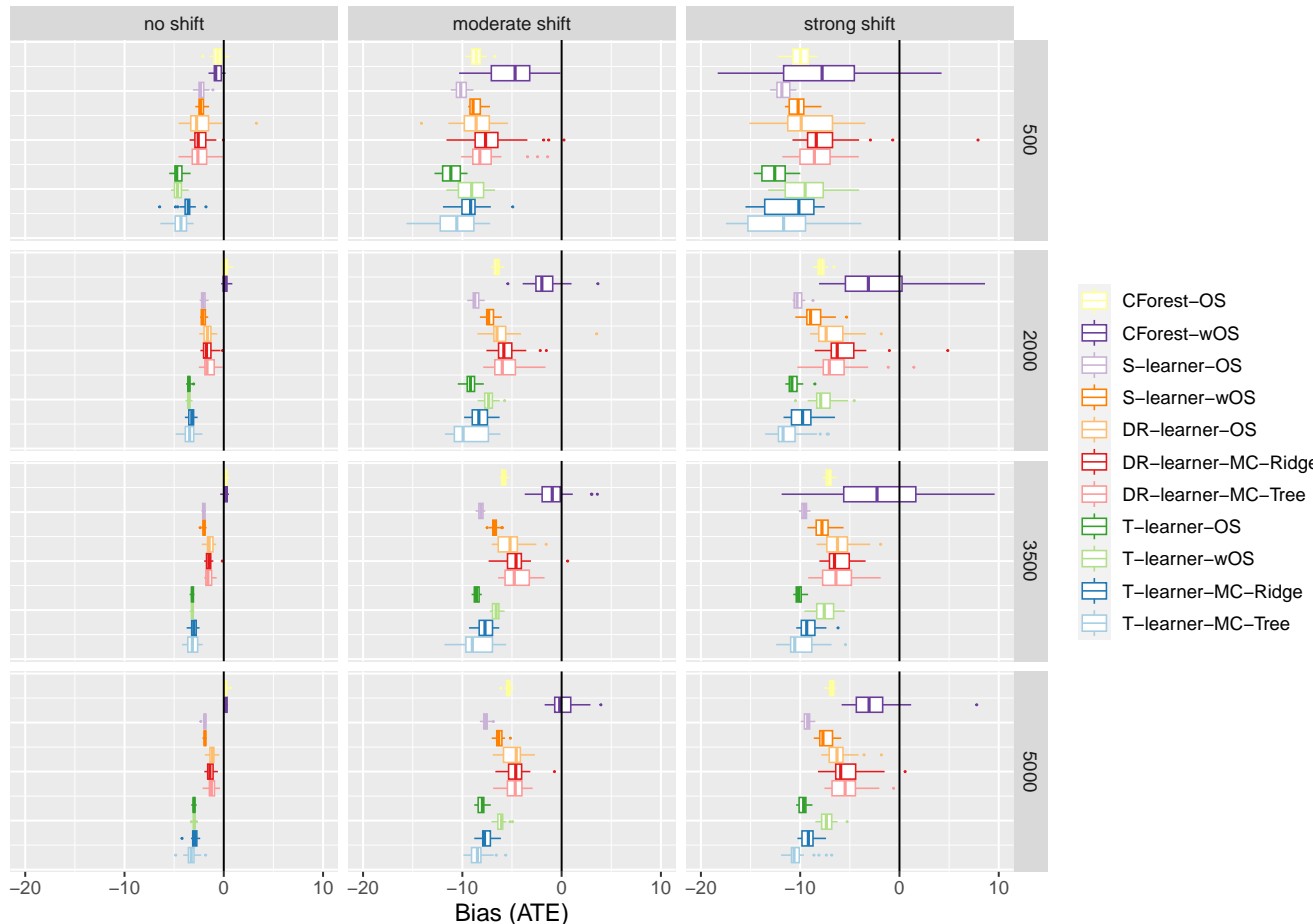

Figure 7: Bias of ATE estimation by shift intensity and training set size for different CATE estimation methods (Simulation 1b (external shift, full linear CATE, logistic confounding)). The distribution of bias scores over simulation runs is shown. Given an external shift between training and test data, DR-learner-MC-Ridge performs best among the shift-blind methods, particularly in settings with small initial training data.

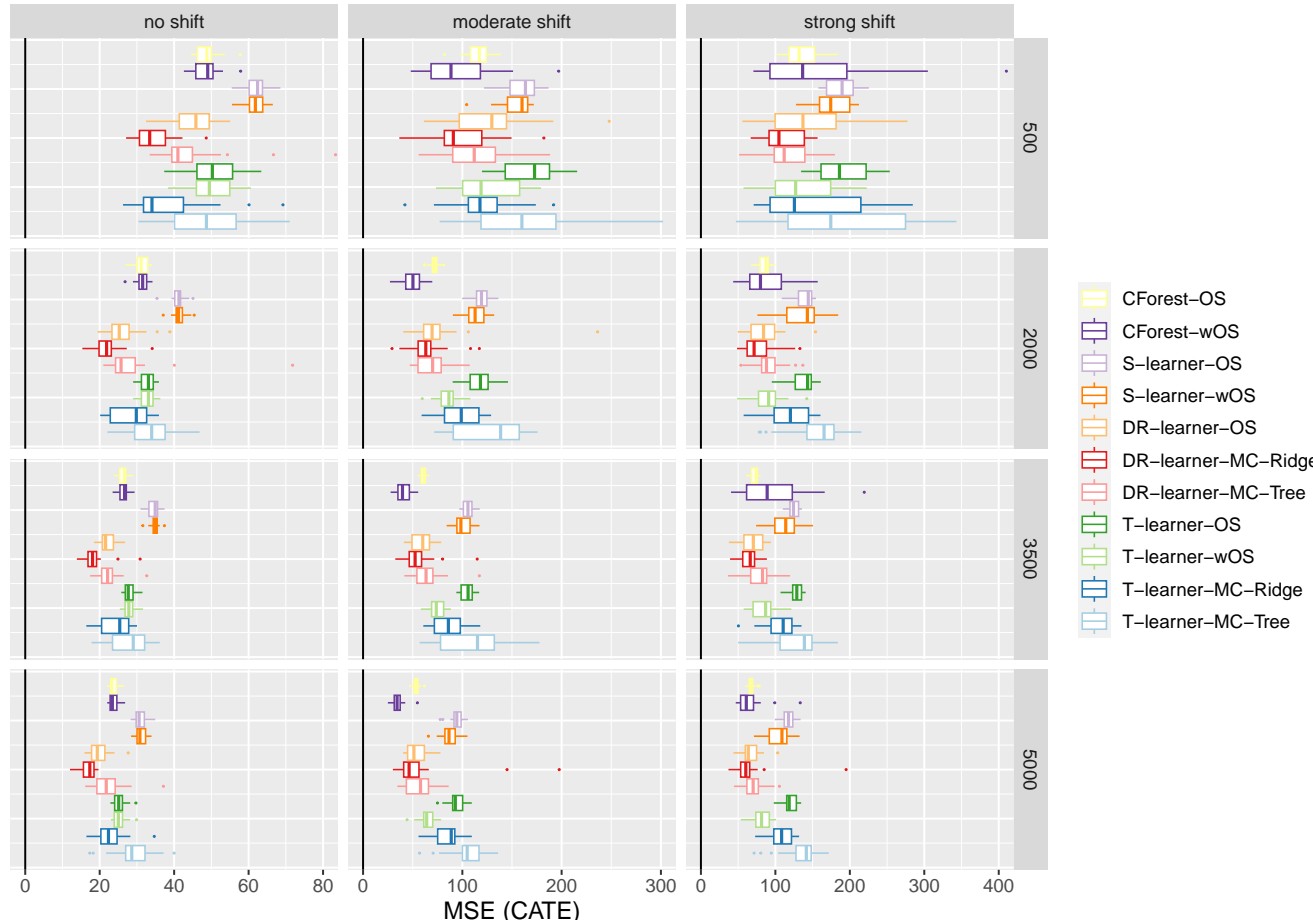

Figure 8: MSE of CATE estimation by shift intensity and training set size for different estimation methods (Simulation 1b (external shift, full linear CATE, logistic confounding)). The distribution of MSE scores over simulation runs is shown. DR-learner-MC-Ridge and T-learner-MC-Ridge consistently improve over DR-learner-OS and T-learner-OS. DR-learner-MC-Ridge performs best overall in settings with small to moderately sized training data.

### D.2.3 Observational study and RCT

We simulate a setting in which we have access to training data from an observational study (OS) and from a small randomized control trial (RCT). We consider a covariate/ external shift between the observational study, $\mathcal{D}_{os}$, and the RCT, $\mathcal{D}_{rct}$. We further assume unobserved confounding either in both data sources (D.2.3) or in the observational training data only (D.2.3). The task is to estimate the true CATE using models that learned either in the observational (training) data or in the RCT, or by using both data sources in combination.

$$(X_{train}, T_{train}, Y_{train}) \sim \mathcal{D}_{os}, (X_{audit}, T_{audit}, Y_{audit}) \sim \mathcal{D}_{rct}, (X_{test}, T_{test}, Y_{test}) \sim \mathcal{D}_{os}$$

That is, the randomized controlled trial data is crucial to obtain identification, but ultimately we seek a predictor with good performance on the covariate distribution of the *observational data.*

**Simulation 2a (confounded observational data and RCT)** In the first simulation, we consider covariate shifts from the observational to the RCT setting alone.

**Assumption 5** (Covariate shift from observational to RCT)**.**

$$P(X_{obs}) \neq P(X_{rct})$$
$$P(Y_{obs} = y \mid X, U, A) = P(Y_{rct} = y \mid X, U, A), \forall y$$

In addition to the setup in Appendix D.2, we introduce unobserved confounding. The specification is as follows:

The unobserved confounder $U$ is correlated with $x_1$:

$$u(x) = \begin{cases} 0.8 & \text{if } x_1 > \bar{x}_1 \\ 0.2 & \text{if } x_1 \leq \bar{x}_1 \end{cases}, \qquad U_i \sim \text{Bern}(u(x))$$

$$\mu_0(x) = 3x_1 + 5x_2$$
$$\mu_1(x) = \mu_0(x) + 3x_1 + 5x_2$$
$$\mu_0^*(x, u) = \mu_0(x) - u$$
$$\mu_1^*(x, u) = \mu_1(x) + 3u$$
$$e^{os}(x, u) = \frac{1}{1 + e^{(2-3u+(-2(x_1-0.5)-1(x_2-0.5)))}}$$
$$e^{rct}(x) = 0.5$$
$$z(x) = \frac{1}{1 + e^{(2(x_2-0.5)+(x_3-0.5))}}$$

**CATE Estimation** We use the following methods for estimating the CATE based on training sets of simulated observational data.

- (**CForest-OS**) Causal forest (Wager and Athey, 2018) trained in the training set of the observational data.

- (**S-learner-OS**) S-learner using random forest trained in the training set of the observational data.

- (**DR-learner-OS**) DR-learner (Kennedy, 2023) using random forest trained in the training set of the observational data.

- (**T-learner-OS**) T-learner using random forest trained in the training set of the observational data.

We estimate DR-learner and T-learner using multi-calibration boosting with simulated RCT data.

- (**DR-learner-MC-Ridge**) DR-learner using random forest in the training set of the observational data is post-processed with MCBoost using ridge regression in the randomized control trial.

- (**DR-learner-MC-Tree**) DR-learner using random forest in the training set of the observational data is post-processed with MCBoost using decision trees in the randomized control trial.

- (**T-learner-MC-Ridge**) T-learner using random forest in the training set of the observational data is post-processed with MCBoost using ridge regression in the randomized control trial.

- (**T-learner-MC-Tree**) T-learner using random forest in the training set of the observational data is post-processed with MCBoost using decision trees in the randomized control trial.

We further compare to CATE learner that are solely based on the simulated RCT data. Shift-reweighting is conducted by training a logistic regression to predict sample membership in the observational versus RCT data and calculating propensity weights $\frac{1-\hat{p}}{\hat{p}}$ based on the predicted probability of membership in the RCT data $\hat{p}$.

- (**CForest-CT**) Causal forest trained in the randomized control trial.

- (**CForest-wCT**) Causal forest trained in the shift-reweighted randomized control trial.

- (**S-learner-CT**) S-learner using random forest trained in the randomized control trial.

- (**S-learner-wCT**) S-learner using random forest trained in the shift-reweighted randomized control trial.

- (**DR-learner-CT**) DR-learner using random forest trained in the randomized control trial.

- (**T-learner-CT**) T-learner using random forest trained in the randomized control trial.

- (**T-learner-wCT**) T-learner using random forest trained in the shift-reweighted randomized control trial.

**Evaluation**  We evaluate bias in ATE and MSE in CATE estimation on a test set drawn from the observational data. In calculating the true ATE and $\tau(x)$, we marginalize over $U$ and compute $\mathrm{E}[Y_i(1)|X] = Y_i(1) + 3\,\mathrm{E}[U|X_i]$ and $\mathrm{E}[Y_i(0)|X] = Y_i(0) - \mathrm{E}[U|X_i]$.

**Results**  We plot the bias of the estimated ATE for each method by shift intensity (column panels) and training set size (row panels) in Figure 9 (see also Table 8 including KL divergence). In the absence of covariate shift (first column), naive learning in the observational data results in biased estimates of the ATE. Utilizing both data sources in combination via multi-calibration boosting allows to improve over the initial DR- and T-learner. Introducing a covariate shift between the observational data and the RCT (second column) degenerates the performance of the RCT-based estimators and the best results are achieved by multi-accurate DR- and T-learner, especially for strong shifts (third column).

Results for the MSE of the estimated CATE are shown in Figure 10 (Table 9). In the absence of covariate shift (first column), the RCT-based estimators outperform the estimators that learned from the observational data. Introducing a shift between the observational study and the RCT (second and third column) increases the MSE of the RCT-based learners considerably such that the best results can now be observed for the tree-based multi-accurate T-learner, followed by the ridge regression-based multi-accurate T-learner and causal forests learned in the observational data.

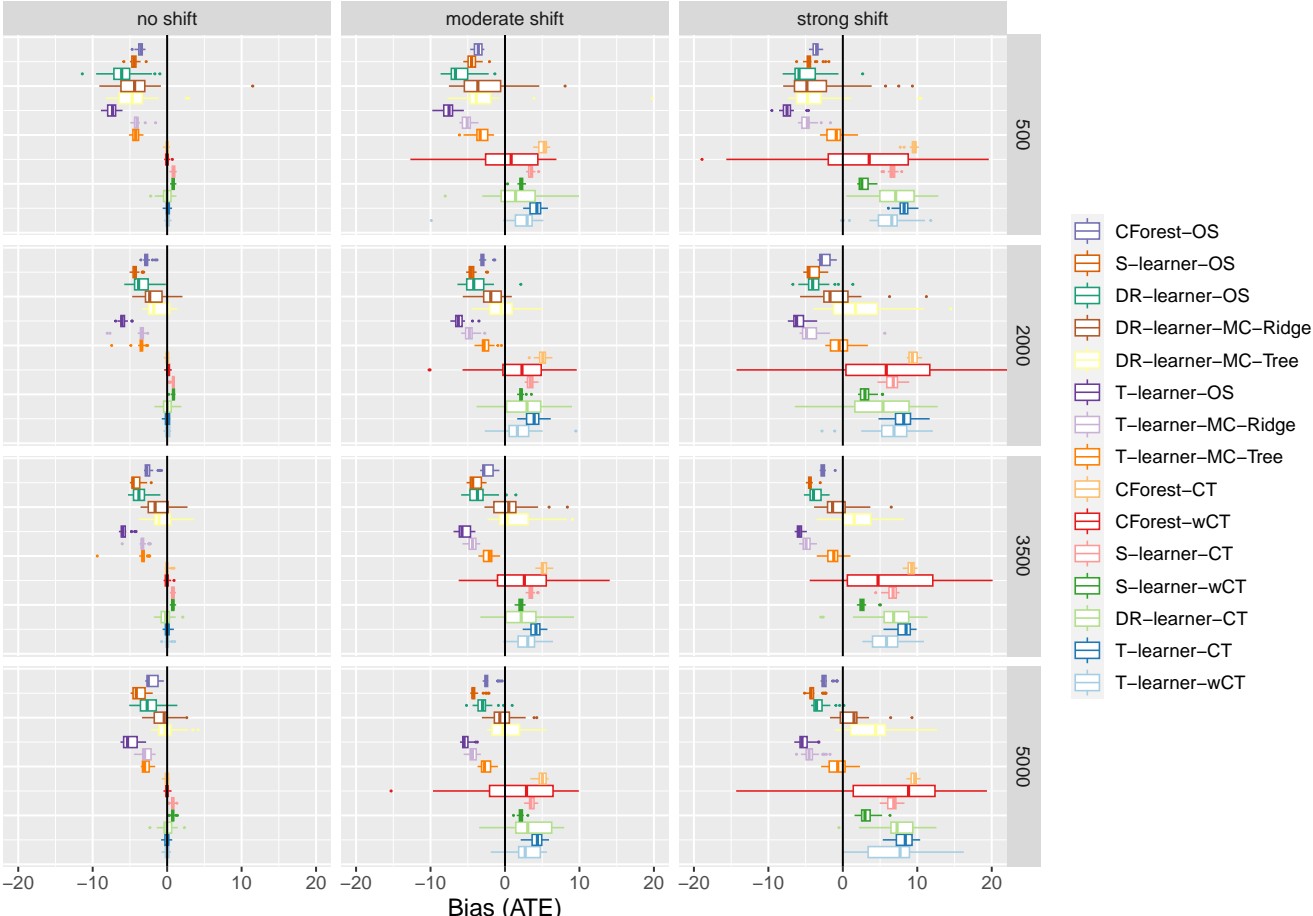

Figure 9: Bias of ATE estimation by shift intensity and training set size for different CATE estimation methods (Simulation 2a (confounded observational data and RCT)). The distribution of bias scores over simulation runs is plotted. Given moderate to strong covariate shift between the observational data and RCT, multi-accurate learner achieve the best results.

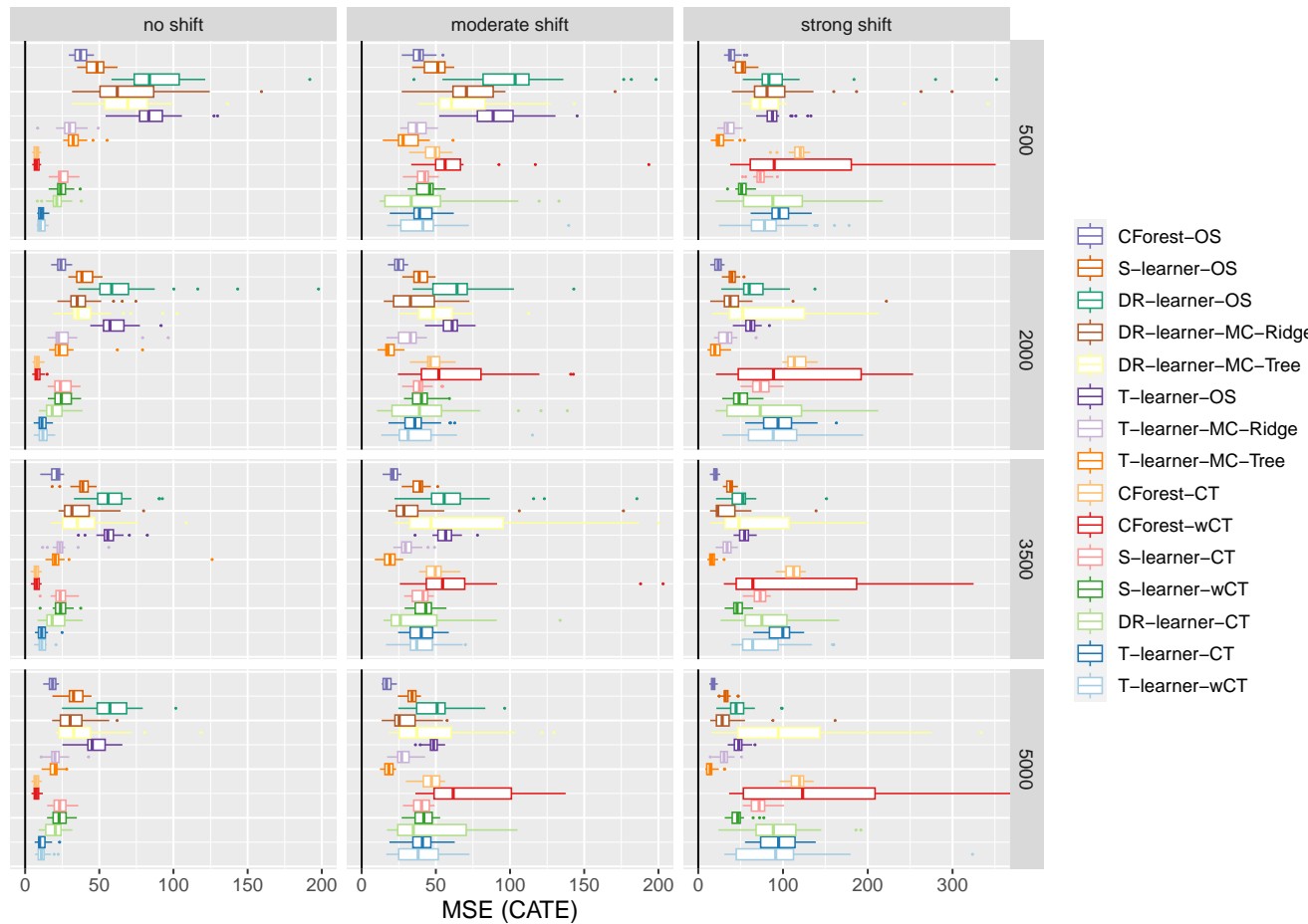

Figure 10: MSE of CATE estimation by shift intensity and training set size for different estimation methods (Simulation 2a (confounded observational data and RCT)). The distribution of MSE scores over simulation runs is shown. T-learner-MC-Tree outperform other methods in settings with shifted RCT data.

**Simulation 2b (total shift between observational data and RCT)**  In simulation 2b, we consider potentially stronger distribution shifts beyond covariate shift alone.

**Assumption 6** (Total shift from observational to RCT)**.**

$$P(X_{obs}) \neq P(X_{rct})$$
$$P(Y_{obs} = y \mid X, A) \neq P(Y_{rct} = y \mid X, A), \forall y$$

The difference between Assumption 6 and Assumption 5 is whether we allow the marginal distribution of $U$ to shift. Assumption 5 is a "conditional model invariance" assumption between the data-generating process and the RCT. A sufficient condition for this to hold is that $P(U_{obs}) = P(U_{rct})$ and the invariant conditional probability assumption above. On the other hand, the total shift of Assumption 6 could arise from shifts in the distribution of $U$. Both of these are additional covariate shifts.

The specification is as follows:

$$\mu_0(x) = \mu_0^{*rct}(x) = 3x_1 + 5x_2$$
$$\mu_1(x) = \mu_1^{*rct}(x) = \mu_0(x) + 3x_1 + 5x_2$$
$$\mu_0^{*os}(x, u) = \mu_0(x) - u$$
$$\mu_1^{*os}(x, u) = \mu_1(x) + 3u$$
$$e^{os}(x, u) = \frac{1}{1 + e^{(2 - 3u + (-2(x_1 - 0.5) - 1(x_2 - 0.5)))}}$$
$$e^{rct}(x) = 0.5$$
$$z(x) = \frac{1}{1 + e^{(2(x_2 - 0.5) + (x_3 - 0.5))}}$$

**Evaluation**  We evaluate bias in ATE and MSE in CATE estimation on a test set that follows the covariate distribution of the observational data, $\mathcal{D}_{os}$. However, in constructing the true ATE and $\tau(x)$ we use $\mu_0(x)$ and $\mu_1(x)$ as specified in the RCT, i.e. without unobserved confounders.

**Results**  The bias of the estimated ATE for each method by shift intensity (column panels) and training set size (row panels) is presented in Figure 11 (and Table 10 including KL divergence). The first set of results (first column) indicate that under unobserved confounding in the observational data only and without external covariate shift, RCT-based estimators are, as expected, unbiased. The multi-accurate DR- and T-learner that draw on both data sources are able to reduce the bias of the naive DR- and T-learner. As the external shift between the observational data and the RCT increases (second and third column), learning only on the RCT incurs bias and shift-reweighted methods as well as (tree-based) multi-accurate DR- and T-learner achieve the best results.

The results for the MSE of the CATE are shown in Figure 12 (Table 11). Similar to the results for bias, the RCT-based estimators perform best under the no external covariate shift setting (first column). Among the estimators based on the observational data, the ridge- and tree-based multi-accurate T-learner and causal forests perform best. Tree-based post-processing performs best among all methods in scenarios with strong covariate shift (third column).

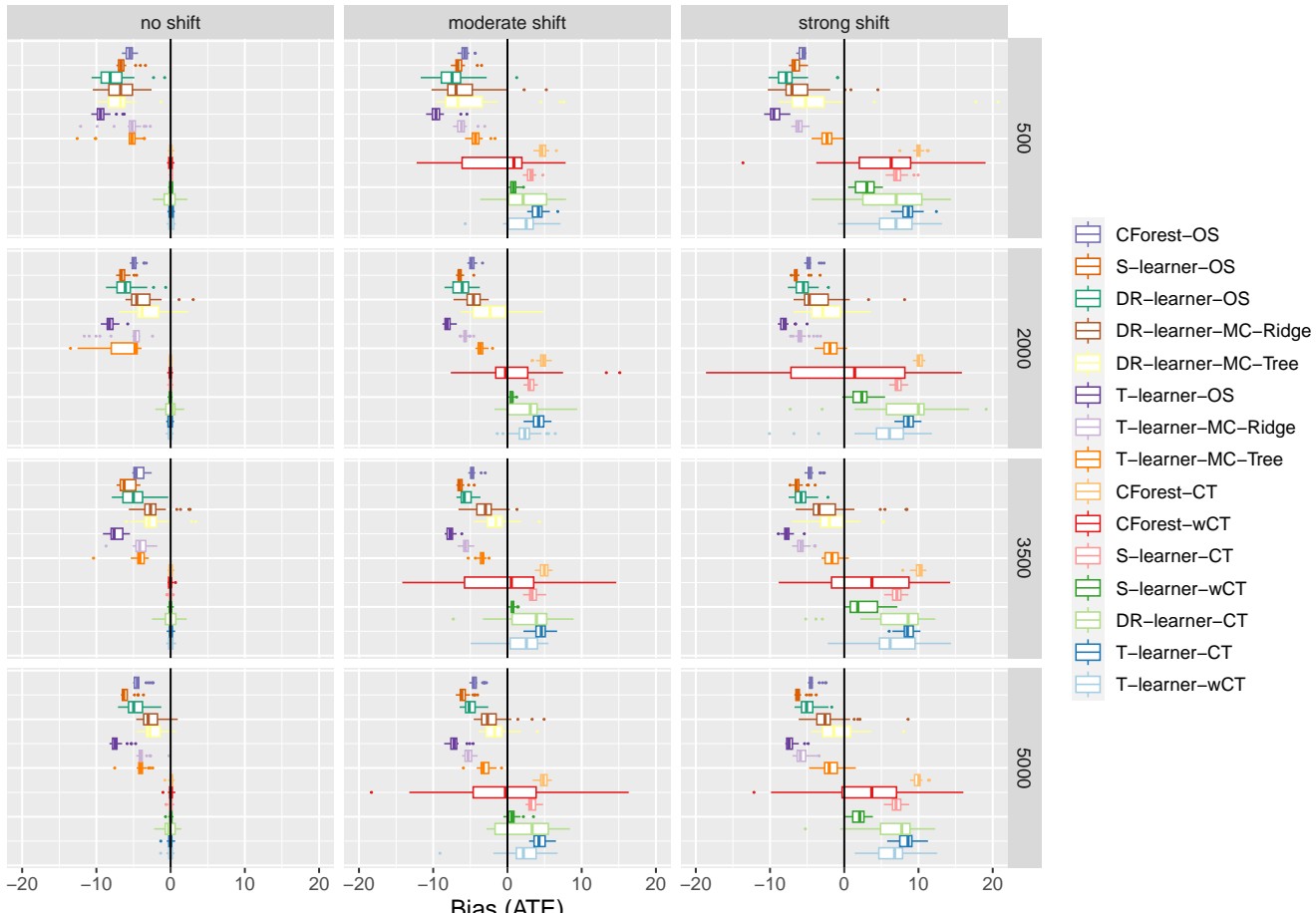

Figure 11: Bias of ATE estimation by shift intensity and training set size for different CATE estimation methods (Simulation 2b (total shift between observational data and RCT)). The distribution of bias scores over simulation runs is shown. As the external shift between the observational data and the RCT increases, multi-accurate DR-learner and T-learner-MC-Tree are competitive with shift-reweighted learning.

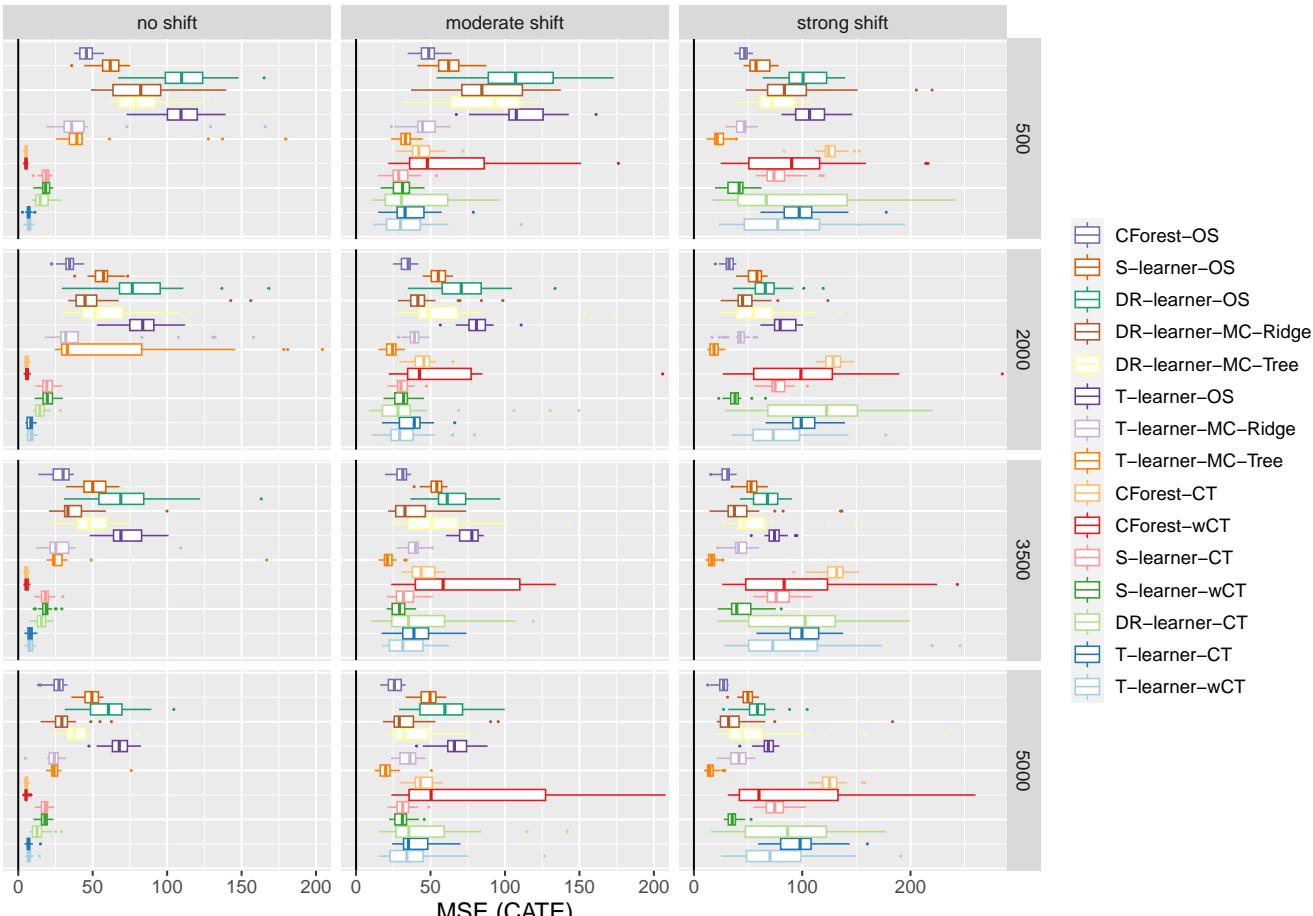

Figure 12: MSE of CATE estimation by shift intensity and training set size for different estimation methods (Simulation 2b (total shift between observational data and RCT)). The distribution of MSE scores over simulation runs is shown. T-learner-MC-Tree performs best among all methods in scenarios with strong covariate shift.

Table 4: Bias of ATE estimation by shift intensity and training set size for different CATE estimation methods, averaged over simulation runs (Simulation 1a (external shift, linear CATE, beta confounding)). For each setting, method achieving best performance printed in **bold** (second best in *italic*).

| Train size | Shift degree | KL-Div. | CForest | | S-learner | | DR-learner | | | T-learner | | | |
|---|---|---|---|---|---|---|---|---|---|---|---|---|---|
| | | | OS | wOS | OS | wOS | OS | Ridge | Tree | OS | wOS | Ridge | Tree |
| | 0 | 0.08 | 0.04 | **0** | *-0.02* | -0.02 | 0.76 | 0.35 | 0.34 | -0.09 | -0.12 | -0.15 | -0.11 |
| | 0.25 | 0.23 | 1.62 | **-0.11** | 2.13 | 1.87 | 0.83 | 1.19 | 1.39 | 1.17 | *0.61* | 0.64 | 0.84 |
| | 0.5 | 0.65 | 3.17 | **-0.09** | 4.2 | 3.39 | 1.92 | 1.88 | 1.88 | 2.34 | *1.34* | 1.57 | 1.8 |
| | 0.75 | 1.15 | 4.46 | **-0.55** | 5.96 | 4.69 | 2.43 | 2.65 | 3.4 | 3.41 | *2.17* | 2.49 | 2.45 |
| 500 | 1 | 1.62 | 5.26 | **1.52** | 7.05 | 5.6 | 3.69 | 3.28 | 3.55 | 4.24 | *2.77* | 3.01 | 3.45 |
| | 1.25 | 1.91 | 5.87 | **-2.09** | 7.65 | 5.92 | 3.89 | 3.36 | 4.42 | 4.4 | *2.55* | 2.77 | 3.08 |
| | 1.5 | 2.09 | 6.29 | **-2.33** | 8.1 | 6.38 | 2.59 | 3.68 | 3.93 | 5.09 | *2.46* | 3.4 | 3.69 |
| | 1.75 | 2.22 | 6.8 | *-2.73* | 8.45 | 7.42 | 4.28 | 4.91 | 4.93 | 5.13 | **2.26** | 3.01 | 4.21 |
| | 2 | 2.40 | 6.88 | -4.08 | 8.6 | 7.1 | 5.01 | 4.01 | 4.31 | 5.17 | **1.69** | *3.48* | 3.58 |
| | 0 | 0.02 | 0.04 | 0.04 | *0.01* | **0** | 0.31 | -0.01 | 0.21 | -0.01 | -0.01 | 0.16 | 0.2 |
| | 0.25 | 0.17 | 0.79 | *-0.34* | 1.79 | 1.35 | 0.46 | 0.61 | 0.6 | 0.74 | **0.33** | 0.5 | 0.58 |
| | 0.5 | 0.60 | 1.8 | **-0.65** | 3.62 | 2.42 | 1.41 | 1.21 | 1.35 | 1.68 | *0.98* | 1.19 | 1.24 |
| | 0.75 | 1.11 | 2.52 | *-1.41* | 5.14 | 2.87 | 1.8 | **1.4** | 1.92 | 2.39 | 1.43 | 1.77 | 2 |
| 2000 | 1 | 1.52 | 2.97 | -2.08 | 6 | 3.27 | 1.82 | **1.51** | 1.91 | 2.76 | *1.58* | -2.41 | 2.18 |
| | 1.25 | 1.86 | 3.35 | -2.12 | 6.65 | 3.96 | 2.35 | **1.92** | 2.1 | 3.26 | *1.99* | 2.2 | 2.52 |
| | 1.5 | 2.05 | 3.63 | -2.84 | 7.07 | 4.21 | 2.38 | 2.19 | *2.17* | 3.49 | **1.92** | 2.56 | 2.89 |
| | 1.75 | 2.21 | 3.81 | -3.27 | 7.29 | 4.5 | 2.29 | *2.08* | 2.27 | 3.59 | **1.86** | 2.54 | 2.84 |
| | 2 | 2.32 | 3.93 | -3.59 | 7.48 | 4.55 | 2.67 | *2.32* | 2.52 | 3.85 | **1.71** | 3.01 | 3.59 |
| | 0 | 0.02 | *0.03* | **0.02** | -0.05 | -0.05 | 0.28 | 0.14 | 0.14 | -0.07 | -0.07 | -0.18 | -0.09 |
| | 0.25 | 0.16 | 0.62 | **-0.27** | 1.65 | 1.15 | 0.39 | 0.53 | 0.43 | 0.66 | *0.3* | 0.48 | 0.57 |
| | 0.5 | 0.60 | 1.31 | **-0.57** | 3.41 | 2.11 | 0.94 | 0.99 | *0.82* | 1.51 | 0.84 | 1.03 | 1.37 |
| | 0.75 | 1.10 | 2.08 | *-1.22* | 4.77 | 2.57 | 1.5 | **1.13** | 1.37 | 2.25 | 1.38 | 1.62 | 1.81 |
| 3500 | 1 | 1.52 | 2.38 | -1.63 | 5.64 | 2.83 | 1.88 | **1.54** | 1.62 | 2.61 | *1.55* | 2 | 2.29 |
| | 1.25 | 1.82 | 2.59 | -2.1 | 6.19 | 3.03 | 1.7 | **1.51** | 1.87 | 2.91 | *1.64* | 2.16 | 2.38 |
| | 1.5 | 2.05 | 2.72 | -3.61 | 6.45 | 3.28 | 2.13 | **1.59** | 1.98 | 3.05 | *1.61* | 2.19 | 2.62 |
| | 1.75 | 2.20 | 2.94 | -3.18 | 6.7 | 3.67 | 2.18 | *1.69* | 1.85 | 3.23 | **1.67** | 2.45 | 2.73 |
| | 2 | 2.29 | 3 | -3.97 | 6.87 | 3.79 | *1.9* | 2.28 | 2.26 | 3.35 | **1.49** | 2.4 | 2.87 |
| | 0 | 0.01 | -0.06 | -0.05 | -0.11 | -0.11 | -0.06 | *0.02* | **-0.01** | -0.13 | -0.13 | -0.14 | -0.14 |
| | 0.25 | 0.16 | 0.52 | **-0.25** | 1.57 | 1.07 | 0.32 | 0.36 | 0.41 | 0.62 | *0.27* | 0.49 | 0.71 |
| | 0.5 | 0.59 | 1.1 | **-0.54** | 3.2 | 1.82 | 0.76 | *0.67* | 0.7 | 1.39 | 0.74 | 0.96 | 1.14 |
| | 0.75 | 1.10 | 1.65 | -1.38 | 4.56 | 2.28 | **1.11** | *1.11* | 1.39 | 2.07 | 1.2 | 1.56 | 1.66 |
| 5000 | 1 | 1.50 | 2.02 | -1.86 | 5.38 | 2.55 | **1.27** | *1.4* | 1.59 | 2.47 | 1.5 | 1.83 | 2.24 |
| | 1.25 | 1.81 | 2.34 | -1.66 | 5.91 | 2.97 | *1.58* | 1.69 | **1.47** | 2.76 | 1.69 | 2.02 | 2.4 |
| | 1.5 | 2.05 | 2.37 | -3.08 | 6.2 | 2.94 | 1.93 | 1.97 | **1.54** | 2.89 | *1.62* | 2.3 | 2.23 |
| | 1.75 | 2.21 | 2.55 | -3.24 | 6.39 | 3.26 | 1.91 | 1.74 | **1.45** | 3.06 | *1.68* | 2.49 | 3.14 |
| | 2 | 2.30 | 2.63 | -3.56 | 6.53 | 3.7 | *1.83* | 1.9 | 1.98 | 3.2 | **1.7** | 2.38 | 2.62 |

Table 5: MSE of CATE estimation by shift intensity and training set size for different estimation methods, averaged over simulation runs (Simulation 1a (external shift, linear CATE, beta confounding)). For each setting, method achieving best performance printed in **bold** (second best in *italic*).

| Train size | Shift degree | KL Div. | CForest OS | CForest wOS | S-learner OS | S-learner wOS | DR-learner OS | DR-learner Ridge | DR-learner Tree | T-learner OS | T-learner wOS | T-learner Ridge | T-learner Tree |
|---|---|---|---|---|---|---|---|---|---|---|---|---|---|
| 500 | 0 | 0.08 | *12.52* | **12.5** | 19.64 | 19.69 | 39.11 | 35.83 | 34.71 | 14.99 | 15.01 | 23.65 | 23.34 |
| | 0.25 | 0.23 | 16.22 | *14.87* | 25.4 | 24.31 | 39.71 | 38.62 | 33.19 | 18.39 | 17.96 | **13.76** | 23.99 |
| | 0.5 | 0.65 | 23.97 | *17.31* | 38.24 | 30.95 | 49.65 | 53.33 | 34.47 | 22.98 | 20.9 | **15.71** | 28.8 |
| | 0.75 | 1.15 | 33.76 | *22.6* | 53.54 | 39.98 | 56.69 | 43.87 | 48.8 | 29.32 | 25.36 | **21.3** | 31.88 |
| | 1 | 1.62 | 39.1 | 33.58 | 63.77 | 46.94 | 58.16 | 59.52 | 49.9 | 34.82 | *30.83* | **22.51** | 36.65 |
| | 1.25 | 1.91 | 44.13 | 50.24 | 69.13 | 47.81 | 57.9 | 58.83 | 60.97 | 34.24 | *26.37* | **22.24** | 32.55 |
| | 1.5 | 2.09 | 47.91 | 55.52 | 74.24 | 51.85 | 82.21 | 66.16 | 56.16 | 40.65 | *25.54* | **24.22** | 37.26 |
| | 1.75 | 2.22 | 53.07 | 164 | 78.5 | 63.76 | 60.46 | 64.53 | 62.66 | 39.19 | *29.42* | **27.65** | 41.72 |
| | 2 | 2.40 | 54.01 | 100.17 | 80.29 | 58.83 | 68.18 | 63.73 | 56.24 | 40.18 | *24.46* | **22.7** | 37.35 |
| 2000 | 0 | 0.02 | **5.06** | *5.1* | 14.97 | 14.92 | 15.75 | 15.83 | 15.31 | 9.56 | 9.52 | 8.5 | 15.34 |
| | 0.25 | 0.17 | *6.13* | **5.98** | 19.28 | 17.02 | 16.96 | 15.4 | 15.32 | 10.43 | 10.35 | 8.83 | 15.77 |
| | 0.5 | 0.60 | *10.47* | **9.33** | 29.76 | 20.21 | 20.28 | 18.6 | 17.6 | 14.83 | 14.49 | 11.65 | 18.63 |
| | 0.75 | 1.11 | **13.19** | 14.76 | 40.88 | 21.47 | 19.86 | 18.81 | 19.15 | 17.24 | 15.84 | *13.61* | 22.23 |
| | 1 | 1.52 | **15.44** | 23.4 | 47.64 | 23.43 | 23.09 | 21.72 | 25.05 | 19.44 | *17.07* | 541.18 | 22.88 |
| | 1.25 | 1.86 | *17.03* | 30.8 | 53.67 | 27.69 | 25.46 | 22.39 | 21.78 | 21.68 | 19.04 | **14.56** | 25.03 |
| | 1.5 | 2.05 | 18.42 | 40.07 | 57.9 | 28.71 | 26.72 | 26.79 | 24.99 | 23.04 | *17.78* | **16.42** | 25.81 |
| | 1.75 | 2.21 | 19.23 | 42.16 | 59.87 | 29.85 | 26.8 | 22.7 | 22.47 | 23.04 | *16.93* | **15.57** | 27.11 |
| | 2 | 2.32 | 19.99 | 45.5 | 62.18 | 30.1 | 24.72 | 27.07 | 23.7 | 25.36 | **16.63** | *19.02* | 29.57 |
| 3500 | 0 | 0.02 | *3.2* | **3.19** | 12.91 | 12.87 | 12.56 | 11.02 | 11.25 | 7.64 | 7.63 | 7.24 | 12.44 |
| | 0.25 | 0.16 | *4.22* | **4.17** | 16.63 | 14.13 | 12.04 | 11.28 | 10.98 | 9 | 9.05 | 7.34 | 13.86 |
| | 0.5 | 0.60 | *6.57* | **6.46** | 26.72 | 16.98 | 15.43 | 13.4 | 11.72 | 12.47 | 12.03 | 9.51 | 16.99 |
| | 0.75 | 1.10 | **9.78** | 12.25 | 36.13 | 18.59 | 14.68 | 14.92 | 14.56 | 15.35 | 14.25 | *11.21* | 17.79 |
| | 1 | 1.52 | **10.65** | 14.68 | 42.75 | 19.42 | 17.16 | 15.14 | 15.14 | 16.8 | 15 | *12.84* | 21.89 |
| | 1.25 | 1.82 | **10.83** | 23.62 | 47.2 | 19.39 | 16.18 | 16.57 | 16.01 | 17.78 | 14.77 | *12.98* | 20.17 |
| | 1.5 | 2.05 | **11.61** | 32.26 | 49.25 | 20.51 | 18.26 | 16.88 | 17.81 | 18.68 | 14.49 | *13.39* | 23.67 |
| | 1.75 | 2.20 | **12.69** | 38.13 | 51.6 | 22.77 | 17.92 | 16.71 | 16.32 | 19.75 | 15.28 | *15.07* | 21.18 |
| | 2 | 2.29 | **12.73** | 56.72 | 53.34 | 23.36 | 16.4 | 21.69 | 19.04 | 20.17 | 15.18 | *14.76* | 22.09 |
| 5000 | 0 | 0.01 | **2.62** | *2.63* | 12.45 | 12.36 | 9.31 | 8.73 | 9.23 | 6.97 | 6.99 | 6.14 | 10.51 |
| | 0.25 | 0.16 | *3.19* | **3.01** | 15.53 | 13.18 | 9.32 | 8.49 | 8.95 | 7.77 | 7.73 | 7.74 | 12.25 |
| | 0.5 | 0.59 | *4.95* | **4.58** | 23.96 | 14.14 | 10.56 | 9.48 | 10.36 | 10.57 | 10.09 | 8.2 | 14.69 |
| | 0.75 | 1.10 | **6.66** | *9.45* | 33.15 | 16.08 | 12.62 | 11.07 | 12.24 | 13.68 | 12.92 | 11.27 | 16.6 |
| | 1 | 1.50 | **7.89** | 11.84 | 38.96 | 16.78 | 13.9 | 12.46 | 12.89 | 14.89 | 13.63 | *11.27* | 18.55 |
| | 1.25 | 1.81 | **9.41** | 16.77 | 43.57 | 18.71 | 15.24 | *12.35* | 13.52 | 16.02 | 13.83 | 22.07 | 20.03 |
| | 1.5 | 2.05 | **9.06** | 23.63 | 45.78 | 17.95 | 14.93 | 13.22 | *13.17* | 16.68 | 13.61 | 13.18 | 19.91 |
| | 1.75 | 2.21 | **9.82** | 29.24 | 47.46 | 19.6 | 14.82 | *13.09* | 14.68 | 17.55 | 13.89 | 14.38 | 22.76 |
| | 2 | 2.30 | **10.36** | 40.95 | 48.85 | 22.73 | 15.09 | 14.9 | 15.18 | 18.62 | *14.32* | 14.42 | 19.83 |

Table 6: Bias of ATE estimation by shift intensity and training set size for different CATE estimation methods, averaged over simulation runs (Simulation 1b (external shift, full linear CATE, logistic confounding)). For each setting, method achieving best performance printed in **bold** (second best in *italic*).

| Train size | Shift degree | KL Div. | CForest OS | CForest wOS | S-learner OS | S-learner wOS | DR-learner OS | DR-learner Ridge | DR-learner Tree | T-learner OS | T-learner wOS | T-learner Ridge | T-learner Tree |
|---|---|---|---|---|---|---|---|---|---|---|---|---|---|
| | 0 | 0.07 | **-0.59** | *-0.66* | -2.25 | -2.27 | -2.28 | -2.32 | -2.42 | -4.59 | -4.6 | -3.73 | -4.4 |
| | 0.25 | 0.22 | *-2.94* | **-1.19** | -4.51 | -3.91 | -3.96 | -3.38 | -3.88 | -6.21 | -5.58 | -4.39 | -5.64 |
| | 0.5 | 0.60 | -5.21 | **-1.94** | -6.68 | -5.78 | *-5.14* | -5.16 | -5.36 | -7.94 | -6.79 | -6.71 | -8.1 |
| | 0.75 | 1.05 | -7.23 | **-2.63** | -8.54 | -7.44 | -6.94 | *-5.77* | -6.43 | -9.59 | -7.92 | -8.42 | -9.96 |
| 500 | 1 | 1.47 | -8.57 | **-4.95** | -10.07 | -8.73 | -8.58 | *-6.96* | -7.66 | -11.09 | -9.13 | -9.26 | -10.79 |
| | 1.25 | 1.76 | -9.07 | **-5.13** | -10.71 | -9.44 | -6.83 | *-6.66* | -7.65 | -11.62 | -9.41 | -10.17 | -11.92 |
| | 1.5 | 2.02 | -9.56 | **-6.54** | -11.14 | -9.84 | -8.59 | *-6.72* | -8.72 | -12 | -9.49 | -10.49 | -12.25 |
| | 1.75 | 2.21 | -9.71 | **-5.74** | -11.54 | -10.19 | -8.4 | *-7.83* | -8.57 | -12.48 | -9.76 | -11.2 | -12.58 |
| | 2 | 2.35 | -10.04 | *-7.48* | -11.7 | -10.16 | -9.39 | **-7.1** | -8.24 | -12.63 | -9.29 | -10.75 | -12.02 |
| | 0 | 0.02 | *0.18* | **0.17** | -2.05 | -2.06 | -1.62 | -1.63 | -1.48 | -3.51 | -3.52 | -3.3 | -3.46 |
| | 0.25 | 0.16 | *-2.01* | **-0.22** | -3.93 | -3.37 | -2.49 | -2.87 | -3.04 | -4.83 | -4.28 | -4.3 | -4.78 |
| | 0.5 | 0.53 | -3.88 | **-0.64** | -5.86 | -4.88 | *-3.36* | -3.81 | -3.81 | -6.53 | -5.42 | -6.2 | -7.06 |
| | 0.75 | 0.99 | -5.55 | **-1.76** | -7.54 | -6.37 | -4.39 | *-3.9* | -4.55 | -8.12 | -6.72 | -7.29 | -8.26 |
| 2000 | 1 | 1.39 | -6.5 | **-1.8** | -8.64 | -7.27 | -5.98 | *-5.41* | -5.71 | -9.15 | -7.28 | -8.18 | -9.26 |
| | 1.25 | 1.73 | -7.18 | **-2.12** | -9.45 | -8.04 | -6.11 | *-5.29* | -5.73 | -9.95 | -7.68 | -8.98 | -10.53 |
| | 1.5 | 1.93 | -7.6 | **-2.23** | -9.74 | -8.27 | -6.07 | *-6.02* | -6.41 | -10.26 | -7.87 | -9.3 | -10.63 |
| | 1.75 | 2.09 | -7.8 | **-3.98** | -10.05 | -8.81 | -6.2 | *-5.67* | -6.62 | -10.55 | -8.44 | -9.95 | -11.77 |
| | 2 | 2.24 | -7.87 | **-2.7** | -10.19 | -8.6 | -6.84 | *-5.42* | -6.29 | -10.63 | -7.7 | -9.79 | -11.07 |
| | 0 | 0.01 | **0.13** | *0.16* | -2.01 | -2 | -1.41 | -1.46 | -1.54 | -3.17 | -3.17 | -3.02 | -3.15 |
| | 0.25 | 0.15 | *-1.64* | **-0.01** | -3.64 | -3.09 | -2.29 | -2.4 | -2.35 | -4.37 | -3.86 | -4.1 | -4.39 |
| | 0.5 | 0.52 | -3.5 | **-0.4** | -5.53 | -4.53 | -3.6 | *-3.47* | -3.77 | -6.08 | -5.04 | -5.9 | -6.55 |
| | 0.75 | 0.98 | -4.77 | **-0.37** | -6.87 | -5.61 | -4.15 | *-4.01* | -4.16 | -7.27 | -5.73 | -6.72 | -7.8 |
| 3500 | 1 | 1.38 | -5.83 | **-0.64** | -8.12 | -6.73 | -5.24 | -4.64 | *-4.55* | -8.57 | -6.61 | -7.76 | -8.58 |
| | 1.25 | 1.70 | -6.39 | **-0.74** | -8.74 | -7.25 | *-4.73* | -5.2 | -6.17 | -9.08 | -7.02 | -8.4 | -9.62 |
| | 1.5 | 1.92 | -6.7 | **-0.03** | -9.06 | -7.42 | -6.11 | *-5.35* | -5.37 | -9.5 | -7.08 | -8.98 | -10 |
| | 1.75 | 2.10 | -7 | **-0.6** | -9.29 | -7.63 | -6.05 | -5.85 | *-5.81* | -9.68 | -7.09 | -8.8 | -10.08 |
| | 2 | 2.24 | -7.17 | **-1.68** | -9.58 | -7.77 | *-6.13* | -6.14 | -6.2 | -10.07 | -7.46 | -9.1 | -9.93 |
| | 0 | 0.01 | **0.21** | *0.22* | -1.91 | -1.89 | -1.2 | -1.35 | -1.23 | -3 | -2.99 | -2.96 | -3.28 |
| | 0.25 | 0.15 | *-1.47* | **0.09** | -3.49 | -2.94 | -2.01 | -2.14 | -2.18 | -4.14 | -3.67 | -3.93 | -4.63 |
| | 0.5 | 0.52 | -3.05 | **0.05** | -5.08 | -4.04 | *-2.9* | -2.94 | -3.08 | -5.53 | -4.47 | -5.01 | -5.58 |
| | 0.75 | 0.97 | -4.53 | **-0.21** | -6.68 | -5.4 | *-3.9* | -3.92 | -4 | -7.11 | -5.58 | -6.81 | -7.51 |
| 5000 | 1 | 1.40 | -5.38 | **0.18** | -7.65 | -6.25 | -4.79 | *-4.55* | -4.7 | -8.02 | -6.14 | -7.51 | -8.49 |
| | 1.25 | 1.69 | -5.99 | **-0.97** | -8.28 | -6.76 | *-4.67* | -5.11 | -4.95 | -8.64 | -6.45 | -8.24 | -9.2 |
| | 1.5 | 1.93 | -6.32 | **-0.21** | -8.76 | -6.99 | -5.27 | *-4.71* | -5.39 | -9.11 | -6.76 | -8.62 | -9.71 |
| | 1.75 | 2.08 | -6.42 | **-0.64** | -8.83 | -7.1 | -5.64 | *-4.39* | -5.24 | -9.28 | -6.77 | -8.57 | -9.49 |
| | 2 | 2.23 | -6.88 | **-2.61** | -9.24 | -7.47 | -6.14 | *-5.33* | -5.35 | -9.68 | -7.28 | -9.12 | -10.24 |

Table 7: MSE of CATE estimation by shift intensity and training set size for different estimation methods, averaged over simulation runs (Simulation 1b (external shift, full linear CATE, logistic confounding)). For each setting, method achieving best performance printed in **bold** (second best in *italic*).

| Train size | Shift degree | KL Div. | CForest | | S-learner | | DR-learner | | | T-learner | | | |
|---|---|---|---|---|---|---|---|---|---|---|---|---|---|
| | | | OS | wOS | OS | wOS | OS | Ridge | Tree | OS | wOS | Ridge | Tree |
| 500 | 0 | 0.07 | 48.96 | 48.83 | 61.44 | 61.47 | 49.01 | **36.24** | 41.68 | 50.37 | 50.47 | *37.92* | 48.71 |
| | 0.25 | 0.22 | 58.03 | 52.83 | 79.93 | 79.13 | 64.22 | **47.74** | 53.11 | 72.51 | 64.08 | *52.5* | 67.26 |
| | 0.5 | 0.60 | 75.47 | **59.45** | 107.39 | 107.74 | 76.5 | *65.54* | 69.39 | 103.71 | 86.43 | 80.42 | 111.06 |
| | 0.75 | 1.05 | 98.37 | **78.02** | 134.31 | 132.05 | 109.67 | *83.42* | 87.06 | 132.66 | 105.15 | 107.44 | 147.33 |
| | 1 | 1.47 | 116.47 | **94.48** | 160.77 | 154.28 | 126.34 | *110.72* | 111.87 | 164.12 | 127.02 | 121.08 | 165.23 |
| | 1.25 | 1.76 | *121.81* | 132.84 | 169.13 | 165.05 | 142.91 | **108.59** | 128.54 | 173.35 | 133.51 | 138.64 | 186.45 |
| | 1.5 | 2.02 | 128.21 | 144.18 | 177.03 | 169.83 | 133.75 | **115.82** | *119.92* | 179.79 | 137.04 | 142.15 | 194.81 |
| | 1.75 | 2.21 | 129.24 | 139.23 | 183.05 | 177.42 | 153.78 | **109.25** | *123.72* | 189.41 | 140.58 | 158.14 | 198.73 |
| | 2 | 2.35 | 136.23 | 156.07 | 187.09 | 176.18 | 143.81 | **114.75** | *133.21* | 193.61 | 136.73 | 148.47 | 192.43 |
| 2000 | 0 | 0.02 | 31.25 | 31.29 | 41.25 | 41.46 | 26.21 | **23.87** | *25.76* | 32.69 | 32.76 | 28.04 | 33.36 |
| | 0.25 | 0.16 | 35.78 | *33.1* | 56.21 | 54.81 | 37.36 | **31.1** | 35.15 | 49.45 | 42.36 | 40.36 | 51.18 |
| | 0.5 | 0.53 | 47.93 | **36.68** | 78.99 | 74.7 | 53.38 | *47.29* | 58.23 | 74.61 | 57.9 | 68 | 86.61 |
| | 0.75 | 0.99 | 62.81 | **43.77** | 102.92 | 98.62 | 67.95 | *55.98* | 59.31 | 101.23 | 78.57 | 83.52 | 107.64 |
| | 1 | 1.39 | 71.52 | **50.4** | 118.4 | 112.52 | 76.93 | *65.76* | 73.73 | 117.39 | 86.09 | 96.77 | 125.88 |
| | 1.25 | 1.73 | 78.65 | **65.43** | 131.65 | 125.92 | 75.88 | *68* | 82.42 | 130.8 | 91.11 | 111.73 | 148.89 |
| | 1.5 | 1.93 | 83.4 | 87.66 | 134.23 | 128.7 | *80.8* | **77.14** | 94 | 134.91 | 95.45 | 113.56 | 149.28 |
| | 1.75 | 2.09 | 84.4 | **80.99** | 139.06 | 138.6 | 86.48 | *81.35* | 96.41 | 139.34 | 104.81 | 125.67 | 171.48 |
| | 2 | 2.24 | 85.39 | 117.33 | 139.58 | 135.01 | 85.56 | **82.29** | *84.27* | 139.53 | 90.69 | 120.89 | 155.25 |
| 3500 | 0 | 0.01 | 26.41 | 26.48 | 34.62 | 34.77 | *22.3* | **20.45** | 22.49 | 27.94 | 27.94 | 24.09 | 28.1 |
| | 0.25 | 0.15 | 29.68 | **27.89** | 48.19 | 46.15 | 29.57 | *27.95* | 28.28 | 42.62 | 36.47 | 36.65 | 43.35 |
| | 0.5 | 0.52 | 40.64 | **32.07** | 69.51 | 64.46 | *40.27* | 44.18 | 42.4 | 66.82 | 52.49 | 62.16 | 76.77 |
| | 0.75 | 0.98 | 50.62 | **34.89** | 86.45 | 80.05 | 55.44 | *50.07* | 52.88 | 83.88 | 60.45 | 72.47 | 96.5 |
| | 1 | 1.38 | 60.8 | **40.59** | 105.13 | 99.35 | 58.89 | *55.2* | 65.03 | 104.92 | 73.67 | 88.04 | 109.7 |
| | 1.25 | 1.70 | 65.42 | **53.8** | 113.57 | 105.53 | 73.34 | *60.93* | 71.11 | 112.44 | 79.46 | 98.13 | 128.19 |
| | 1.5 | 1.92 | 67.51 | **58.93** | 115.71 | 105.15 | 66.37 | 63.82 | *62.2* | 117.23 | 78.11 | 106 | 131.51 |
| | 1.75 | 2.10 | 71.14 | 84.67 | 119.63 | 110.99 | *68.86* | **60.95** | 72.15 | 119.74 | 79.36 | 100.75 | 133.81 |
| | 2 | 2.24 | 72.57 | 94.48 | 124.32 | 112.8 | 70.37 | **68.83** | *68.96* | 127.26 | 84.85 | 106.19 | 128.92 |
| 5000 | 0 | 0.01 | 23.76 | 23.84 | 30.99 | 31.12 | 19.81 | **18.33** | *19.01* | 25.41 | 25.37 | 22.87 | 28.73 |
| | 0.25 | 0.15 | 26.53 | **25.12** | 43.11 | 40.92 | 25.98 | *25.28* | 26.89 | 38.97 | 33.2 | 36.33 | 45.83 |
| | 0.5 | 0.52 | 35.34 | **28.45** | 60.29 | 54.75 | 32.56 | *31.91* | 35.93 | 57.47 | 42.99 | 48.56 | 60.4 |
| | 0.75 | 0.97 | 46.53 | **31.58** | 81.14 | 73.84 | 48.41 | *41.51* | 53.5 | 81.54 | 58.2 | 75.15 | 91.08 |
| | 1 | 1.40 | 53.35 | **35.06** | 94.19 | 86.83 | 53.36 | *52.64* | 59.22 | 94.05 | 64.56 | 83.78 | 105.81 |
| | 1.25 | 1.69 | 58.87 | **41.09** | 102.34 | 93.53 | 54.68 | *54.36* | 56.8 | 102.96 | 68.77 | 94.5 | 117.64 |
| | 1.5 | 1.93 | 61.59 | **51.86** | 109.6 | 96.23 | *57.74* | 59.66 | 65.16 | 110.6 | 73.29 | 99.92 | 125.89 |
| | 1.75 | 2.08 | **61.85** | 64.77 | 108.83 | 97.83 | 65.31 | *62.42* | 68.62 | 111.54 | 71.68 | 97.3 | 120.56 |
| | 2 | 2.23 | 67.63 | *65.96* | 116.97 | 105 | 67.87 | **60.4** | 78.9 | 119.41 | 81.65 | 107.55 | 134.97 |

Table 8: Bias of ATE estimation by shift intensity and training set size for different CATE estimation methods, averaged over simulation runs (Simulation 2a (confounded observational data and RCT)). For each setting, method achieving best performance printed in **bold** (second best in *italic*).

| Train size | Shift degree | KL Div. | CForest OS | S-learner OS | DR-learner OS | DR-learner Ridge | DR-learner Tree | T-learner OS | T-learner Ridge | T-learner Tree | CForest RCT | CForest wRCT | S-learner RCT | S-learner wRCT | DR-l RCT | T-learner RCT | T-learner wRCT |
|---|---|---|---|---|---|---|---|---|---|---|---|---|---|---|---|---|---|
| 500 | 0 | 2.82 | -3.62 | -4.41 | -5.87 | -3.91 | -4.45 | -7.43 | -4.06 | -4.23 | *0.01* | **0** | 0.83 | 0.82 | -0.07 | 0.05 | 0.05 |
| | 0.25 | 2.94 | -3.75 | -4.43 | -5.1 | -4.29 | -3.88 | -7.52 | -4.18 | -4.16 | 0.85 | *-0.64* | 1.54 | 1.25 | 0.64 | 0.89 | **0.41** |
| | 0.5 | 3.30 | -3.59 | -4.39 | -4.91 | -4.73 | -4.16 | -7.32 | -4.36 | -3.89 | 1.87 | *-1.05* | 2.08 | 1.57 | 1.75 | 1.67 | **0.84** |
| | 0.75 | 4.07 | -3.41 | -3.99 | -4.4 | -2.55 | -3.6 | -6.73 | -4.24 | -3.32 | 3.2 | *-1.37* | 2.7 | 1.88 | 2.56 | 2.67 | **1.17** |
| | 1 | 5.44 | -3.61 | -4.38 | -6.05 | -2.71 | *-1.57* | -7.53 | -5.08 | -3.25 | 5.01 | **1.15** | 3.5 | 2.13 | 1.71 | 4.13 | 2.17 |
| | 1.25 | 6.95 | -3.44 | -4.12 | -4.73 | -3.54 | -3.64 | -7.13 | -4.95 | -2.56 | 6.56 | **0.48** | 4.43 | *2.39* | 3.17 | 5.47 | 3.83 |
| | 1.5 | 9.37 | -3.49 | -4.16 | -4.96 | -3.61 | -3.02 | -7.31 | -4.79 | **-1.77** | 7.89 | *2.07* | 5.16 | 2.53 | 4.42 | 6.26 | 3.97 |
| | 1.75 | 12.16 | -3.45 | -4.18 | -6.22 | -3.01 | -3.7 | -6.98 | -5.08 | **-1.35** | 8.81 | 5.85 | 6.01 | *2.65* | 5.95 | 7.47 | 5.69 |
| | 2 | 15.04 | -3.58 | -4.36 | -4.98 | -3.1 | -3.43 | -7.42 | -4.7 | **-0.95** | 9.5 | *2.38* | 6.6 | 2.83 | 7.1 | 8.21 | 6.07 |
| 2000 | 0 | 2.52 | -2.73 | -4.23 | -3.45 | -1.89 | -1.27 | -5.95 | -3.73 | -3.61 | **0.01** | 0.12 | 0.78 | 0.81 | 0.08 | *0.03* | 0.06 |
| | 0.25 | 2.92 | -2.59 | -4.07 | -3.36 | -1.23 | -1.4 | -5.78 | -3.19 | -3.17 | 0.79 | -1.29 | 1.44 | 1.08 | *0.46* | 0.72 | **0.16** |
| | 0.5 | 3.38 | -2.73 | -4.39 | -3.53 | -1.59 | -1.65 | -6.07 | -3.76 | -3.24 | 1.8 | -2.16 | 2.07 | 1.57 | *1.25* | 1.55 | **0.6** |
| | 0.75 | 4.09 | -2.81 | -4.4 | -4.37 | **-1.47** | -1.82 | -6.1 | -4.3 | -2.97 | 3.34 | -2.05 | 2.7 | 1.91 | 2.48 | 2.82 | *1.5* |
| | 1 | 5.26 | -2.93 | -4.38 | -3.84 | -1.86 | **-0.54** | -6.13 | -4.71 | -2.53 | 4.99 | *1.46* | 3.41 | 2.23 | 2.67 | 3.83 | 2.19 |
| | 1.25 | 6.98 | -2.71 | -4.32 | -3.81 | *-0.9* | **-0.84** | -6.01 | -4.83 | -2.42 | 6.45 | 1.47 | 4.17 | 2.2 | 2.8 | 5.03 | 3.25 |
| | 1.5 | 9.43 | -2.59 | -4.18 | -3.6 | **0.58** | 2.09 | -5.97 | -4.71 | *-1.2* | 7.99 | 4.75 | 5.38 | 2.7 | 4.07 | 6.91 | 6.16 |
| | 1.75 | 11.93 | -2.63 | -4.05 | -3.17 | **0.5** | 2.4 | -5.81 | -4.59 | *-0.87* | 9.05 | 7.45 | 6.38 | 3.15 | 7.05 | 8.17 | 6.99 |
| | 2 | 15.22 | -2.55 | -4.16 | -3.68 | *-0.68* | 2.24 | -5.93 | -4.07 | **-0.42** | 9.46 | 7.39 | 6.66 | 3.16 | 4.94 | 8.15 | 6.7 |
| 3500 | 0 | 2.93 | -2.41 | -4.18 | -3.67 | -0.93 | -0.74 | -5.66 | -3.32 | -3.41 | *0.01* | **0** | 0.76 | 0.76 | -0.19 | 0.05 | 0.04 |
| | 0.25 | 2.85 | -1.97 | -3.62 | -2.37 | -0.86 | **-0.19** | -4.96 | -3.18 | -2.97 | 0.92 | -1.51 | 1.49 | 1.12 | 0.5 | 0.85 | *0.25* |
| | 0.5 | 3.39 | -2.29 | -3.76 | -2.51 | -0.9 | **-0.37** | -5.05 | -3.22 | -2.6 | 1.96 | -1.64 | 2.09 | 1.63 | 1.93 | 1.68 | *0.62* |
| | 0.75 | 4.01 | -2.22 | -3.86 | -2.73 | *-0.37* | **0.15** | -5.2 | -3.88 | -2.56 | 3.4 | -1.98 | 2.76 | 1.95 | 2.48 | 2.87 | 1.29 |
| | 1 | 5.27 | -2.4 | -4.04 | -3.39 | **0.49** | *1.74* | -5.5 | -4.43 | -2.25 | 5.18 | 2.6 | 3.47 | 2.14 | 2.38 | 4.09 | 3.05 |
| | 1.25 | 6.79 | -2.61 | -4.21 | -3.37 | *-0.77* | **0.26** | -5.54 | -4.78 | -2.08 | 6.79 | 1.54 | 4.43 | 2.45 | 3.98 | 5.49 | 3.67 |
| | 1.5 | 9.18 | -2.23 | -3.93 | -2.59 | **1.03** | 1.9 | -5.34 | -4.41 | *-1.32* | 7.85 | 1.72 | 5.14 | 2.47 | 4.67 | 6.2 | 3.96 |
| | 1.75 | 11.94 | -2.46 | -4.1 | -3.17 | **0.71** | 2.59 | -5.43 | -4.63 | *-1.11* | 8.82 | 3.93 | 5.92 | 2.82 | 5.85 | 7.33 | 5.58 |
| | 2 | 15.15 | -2.64 | -4.41 | -3.73 | **-0.5** | 2.05 | -5.81 | -4.86 | *-1.17* | 9.21 | 6.79 | 6.58 | 2.73 | 6.2 | 8.27 | 6.16 |
| 5000 | 0 | 2.64 | -2.12 | -3.78 | -2.52 | -0.64 | *-0.02* | -5 | -2.91 | -2.89 | -0.05 | -0.04 | 0.74 | 0.74 | **0.01** | -0.04 | -0.06 |
| | 0.25 | 2.92 | -2.25 | -3.9 | -3.11 | *-0.24* | -0.55 | -5.14 | -2.91 | -2.93 | 0.81 | -1.68 | 1.49 | 1.04 | 0.97 | 0.73 | **0.12** |
| | 0.5 | 3.33 | -2.06 | -3.69 | -2.5 | **0.1** | *-0.16* | -5 | -3.3 | -2.53 | 1.88 | -1.81 | 2.04 | 1.47 | 1.42 | 1.67 | 0.59 |
| | 0.75 | 4.18 | -2.3 | -4.07 | -3 | *-0.74* | **0.02** | -5.25 | -3.8 | -2.57 | 3.42 | -1.18 | 2.73 | 2.03 | 2.2 | 2.78 | 1.22 |
| | 1 | 5.23 | -2.22 | -3.95 | -2.86 | **-0.2** | *0.41* | -5.23 | -4.33 | -2.57 | 4.96 | 1.25 | 3.51 | 2.12 | 3.17 | 4.12 | 2.68 |
| | 1.25 | 6.87 | -2.52 | -4.16 | -3.41 | **0.4** | 0.95 | -5.46 | -4.72 | -1.87 | 6.89 | *0.74* | 4.52 | 2.43 | 4.1 | 5.65 | 4.13 |
| | 1.5 | 9.18 | -2.33 | -3.97 | -2.96 | **0.17** | 1.78 | -5.1 | -4.55 | *-1.32* | 8.15 | 1.48 | 5.42 | 2.58 | 5.85 | 6.88 | 4.61 |
| | 1.75 | 12.10 | -1.86 | -3.47 | -2.45 | *1.03* | 2.74 | -4.64 | -4.07 | **-0.39** | 8.89 | 6.29 | 6.15 | 2.98 | 6.81 | 7.63 | 6.15 |
| | 2 | 15.23 | -2.25 | -3.9 | -3.03 | *1.32* | 3.8 | -5.16 | -4.27 | **-0.57** | 9.51 | 6.84 | 6.61 | 3.26 | 7.45 | 8.2 | 6.85 |

Table 9: MSE of CATE estimation by shift intensity and training set size for different estimation methods, averaged over simulation runs (Simulation 2a (confounded observational data and RCT)). For each setting, method achieving best performance printed in **bold** (second best in *italic*).

| Train size | Shift degree | KL Div. | CForest OS | S-learner OS | DR-learner OS | DR-learner Ridge | DR-learner Tree | T-learner OS | T-learner Ridge | T-learner Tree | CForest RCT | CForest wRCT | S-learner RCT | S-learner wRCT | DR-l RCT | T-learner RCT | T-learner wRCT |
|---|---|---|---|---|---|---|---|---|---|---|---|---|---|---|---|---|---|
| 500 | 0 | 2.82 | 37.43 | 47.2 | 94.84 | 100.75 | 76.41 | 86.4 | 30.54 | 33.85 | **7.71** | *7.76* | 24.31 | 24.37 | 11.11 | 21.63 | 11.12 |
| | 0.25 | 2.94 | 37.22 | 47.11 | 95.35 | 75.57 | 81.38 | 87.48 | 31.09 | 31.7 | **9.33** | 11.57 | 26.49 | 26.4 | 11.53 | 19.14 | *11.52* |
| | 0.5 | 3.30 | 38.09 | 47.77 | 93.25 | 70.52 | 66.67 | 84.78 | 35.26 | 35.56 | **17.13** | 34.55 | 33.46 | 35.15 | 18 | 24.62 | *17.87* |
| | 0.75 | 4.07 | 35.07 | 44.28 | 93.6 | 91.21 | 64.14 | 78.57 | 30.76 | 29.05 | 27.5 | 52.74 | 34.61 | 37.01 | **25.26** | 32.78 | *26.67* |
| | 1 | 5.44 | 38.69 | 49.32 | 102.71 | 95.48 | 166.78 | 91.69 | *37.94* | **31.16** | 47.8 | 84.65 | 41.38 | 43.62 | 40.63 | 40.01 | 43.95 |
| | 1.25 | 6.95 | *34.34* | 43.26 | 92.51 | 77.29 | 69.69 | 81.17 | 36.15 | **26.93** | 67.26 | 158.72 | 47.33 | 42.52 | 55.63 | 49.43 | 52.96 |
| | 1.5 | 9.37 | *34.78* | 43.29 | 117.41 | 64.09 | 82.92 | 84.48 | 34.91 | **24.9** | 88.28 | 141.6 | 54.21 | 43.13 | 65.94 | 70.72 | 55.18 |
| | 1.75 | 12.16 | 36.48 | 44.88 | 86.79 | 84.26 | 77.4 | 77.93 | *36.47* | **25.58** | 104.41 | 134.25 | 66.15 | 46.86 | 85.97 | 89.8 | 78.45 |
| | 2 | 15.04 | 40.22 | 51.41 | 117.57 | 103.47 | 108.49 | 91.47 | *35.91* | **27.33** | 117.4 | 166.51 | 73.7 | 51.99 | 96.27 | 91.76 | 85.72 |
| 2000 | 0 | 2.52 | 24.77 | 39.92 | 69.39 | 38.64 | 44.04 | 60.21 | 28.88 | 27.3 | **8.2** | *8.49* | 25.29 | 25.55 | 11.9 | 20.17 | 12.31 |
| | 0.25 | 2.92 | 22.66 | 36.62 | 61.15 | 46.68 | 40.69 | 56.5 | 21.7 | 21.79 | **9.11** | 17.4 | 25.27 | 25.67 | *12.27* | 19.06 | 13.67 |
| | 0.5 | 3.38 | 23.36 | 40.53 | 63.01 | 43.82 | 57.06 | 60.26 | 24.92 | 20.93 | **16.28** | 45.42 | 31.93 | 35.5 | *17.8* | 25.63 | 19.64 |
| | 0.75 | 4.09 | *24.91* | 41.14 | 62.5 | 37.85 | 47.54 | 61.23 | 28.29 | **20.71** | 29.3 | 72.08 | 35.42 | 39.42 | 27.35 | 38.08 | 30.23 |
| | 1 | 5.26 | *24.69* | 39.05 | 63.93 | 37.05 | 51.06 | 60.09 | 31.62 | **18.55** | 47.02 | 64.63 | 38.93 | 41.07 | 37.35 | 45.56 | 36.95 |
| | 1.25 | 6.98 | *23.26* | 38.99 | 63.13 | 41.87 | 43.79 | 59.06 | 33.76 | **20.66** | 65.93 | 100.8 | 45.44 | 43.09 | 50.5 | 51.02 | 51.02 |
| | 1.5 | 9.43 | *24.77* | 42 | 79.1 | 64.98 | 96.36 | 64.11 | 33.47 | **21.65** | 89.63 | 103.76 | 59.41 | 49.25 | 76.25 | 69.79 | 78.28 |
| | 1.75 | 11.93 | *24.54* | 39.08 | 71.37 | 74.64 | 98.74 | 59.68 | 31.85 | **19.3** | 108.56 | 158.95 | 70.25 | 52.11 | 95.57 | 103.41 | 101.76 |
| | 2 | 15.22 | *23.43* | 39.83 | 66.41 | 47.75 | 89.98 | 61.15 | 34.14 | **20.65** | 116.68 | 208.11 | 74.49 | 49.51 | 96.45 | 79.35 | 93.46 |
| 3500 | 0 | 2.93 | 20.74 | 38.4 | 57.65 | 36.75 | 40.48 | 56.42 | 24.32 | 24.65 | **7.55** | *7.69* | 24.02 | 24.18 | 11.67 | 20.97 | 11.58 |
| | 0.25 | 2.85 | 17.9 | 34.03 | 52.44 | 32.03 | 40.27 | 48.91 | 23.74 | 22.99 | **10.09** | 21.26 | 26.64 | 27.86 | *12.82* | 21.18 | 14.01 |
| | 0.5 | 3.39 | 18.72 | 33.18 | 51.65 | 27.34 | 39 | 47.32 | 19.87 | *16.82* | **16.62** | 58.54 | 29.97 | 33.15 | 17.48 | 26.29 | 19.55 |
| | 0.75 | 4.01 | *19.19* | 34.73 | 56.67 | 34.18 | 43.26 | 49.95 | 25.81 | **17.5** | 29.86 | 103.45 | 35.11 | 38.22 | 28.17 | 31.97 | 30.87 |
| | 1 | 5.27 | *21.22* | 38.5 | 65.98 | 39.1 | 76.1 | 56.8 | 31.11 | **18.87** | 49.48 | 71.97 | 40.2 | 42.44 | 39.8 | 39.28 | 40.99 |
| | 1.25 | 6.79 | *20.38* | 36.03 | 53.88 | 33.08 | 49.66 | 51.35 | 33.33 | **17.39** | 70.98 | 111.6 | 47.92 | 43.2 | 57.07 | 62.41 | 51.49 |
| | 1.5 | 9.18 | *19.63* | 35.95 | 70.68 | 58.7 | 82.49 | 51.67 | 30.39 | **18.97** | 88.1 | 120.91 | 55.55 | 45.01 | 66.91 | 66.19 | 71.98 |
| | 1.75 | 11.94 | *19.63* | 35.55 | 54.55 | 41.46 | 86.55 | 50.35 | 31.43 | **15.85** | 103.74 | 102.29 | 63.82 | 46.56 | 81.41 | 76.27 | 78.81 |
| | 2 | 15.15 | *20.44* | 37.34 | 52.04 | 33.9 | 70.82 | 53.78 | 34.3 | **16.56** | 111.67 | 136.3 | 72.53 | 47.06 | 97.04 | 80.99 | 78.24 |
| 5000 | 0 | 2.64 | 18.16 | 33.76 | 59.42 | 31.65 | 39.66 | 47.26 | 20.71 | 19.68 | **7.6** | *7.74* | 23.72 | 23.6 | 11.52 | 19.88 | 11.61 |
| | 0.25 | 2.92 | 18.81 | 34.99 | 49.79 | 31.67 | 36.22 | 49.2 | 19.26 | 19.72 | **9.88** | 20.01 | 27.91 | 28.19 | *12.14* | 21.69 | 13.87 |
| | 0.5 | 3.33 | 18.02 | 33.88 | 54.92 | 35.51 | 39.12 | 49.27 | 21.67 | *16.76* | **16.74** | 87.34 | 31.02 | 34.38 | 18.56 | 24.05 | 21.31 |
| | 0.75 | 4.18 | *17.66* | 33.95 | 44.95 | 25.29 | 38.93 | 47.43 | 23.59 | **16.58** | 30.2 | 89.63 | 36.12 | 39.9 | 27.66 | 44.06 | 30.68 |
| | 1 | 5.23 | **17.48** | 33.63 | 49.97 | 29.1 | 48.54 | 47.89 | 28.7 | *18.13* | 46.43 | 80.45 | 40.07 | 42.06 | 39.98 | 46.77 | 38.25 |
| | 1.25 | 6.87 | *20.04* | 36.54 | 53.89 | 33.91 | 50.57 | 51.67 | 32.42 | **16.86** | 72.71 | 121.44 | 49.42 | 45.18 | 58.9 | 66.74 | 56.3 |
| | 1.5 | 9.18 | *17.68* | 32.9 | 45.59 | 30.67 | 60.29 | 45.23 | 31.14 | **14.83** | 92.41 | 162.72 | 56.67 | 43.41 | 74.65 | 78.07 | 63.22 |
| | 1.75 | 12.10 | **16.91** | 32.38 | 47.65 | 36.18 | 80.13 | 45.23 | 27.74 | *16.96* | 105.55 | 123.41 | 66.73 | 48.54 | 86.49 | 92.77 | 83.16 |
| | 2 | 15.23 | *17.7* | 33.02 | 48.57 | 36.72 | 104.62 | 47.78 | 30.67 | **14.75** | 117.25 | 147.32 | 71.05 | 47.19 | 94.93 | 91.72 | 96.34 |

Table 10: Bias of ATE estimation by shift intensity and training set size for different CATE estimation methods, averaged over simulation runs (Simulation 2b (total shift between observational data and RCT)). For each setting, method achieving best performance printed in **bold** (second best in *italic*).

| Train size | Shift degree | KL Div. | CForest OS | S-learner OS | DR-learner OS | DR-learner Ridge | DR-learner Tree | T-learner OS | T-learner Ridge | T-learner Tree | CForest RCT | CForest wRCT | S-learner RCT | S-learner wRCT | DR-l RCT | T-learner RCT | T-learner wRCT |
|---|---|---|---|---|---|---|---|---|---|---|---|---|---|---|---|---|---|
| 500 | 0 | 2.98 | -5.53 | -6.38 | -7.68 | -6.73 | -6.81 | -9.25 | -5.43 | -5.59 | *0.03* | **0** | 0.07 | 0.05 | -0.05 | 0.12 | 0.11 |
|  | 0.25 | 3.19 | -5.37 | -6.13 | -7.43 | -4.82 | -5.46 | -9.03 | -5.33 | -5.31 | 0.7 | -0.78 | 0.75 | **0.24** | 0.47 | 0.84 | *0.31* |
|  | 0.5 | 3.62 | -5.51 | -6.22 | -7.39 | -5.89 | -5.74 | -9.2 | -5.3 | -4.66 | 1.73 | -1.71 | 1.55 | **0.6** | 1.61 | 1.95 | *0.93* |
|  | 0.75 | 4.57 | -5.71 | -6.56 | -7.68 | -7.19 | -6.15 | -9.6 | -6.08 | -4.79 | 3.04 | *-1.09* | 2.29 | **0.77** | 2.01 | 2.95 | 1.58 |
|  | 1 | 5.78 | -5.72 | -6.46 | -7.19 | -5.86 | -4.86 | -9.36 | -6.08 | -4.18 | 4.73 | -2.66 | 3.17 | **0.83** | 2.46 | 4.13 | *2.06* |
|  | 1.25 | 7.79 | -5.36 | -5.96 | -6.53 | -5.11 | -4.3 | -8.72 | -5.8 | -3.31 | 6.21 | *-2.69* | 4.3 | **1.05** | 4.06 | 5.46 | 3.24 |
|  | 1.5 | 10.33 | -5.55 | -6.2 | -5.96 | -4.16 | -3.03 | -9.24 | -6.04 | -3.22 | 7.82 | *2.47* | 5.49 | **1.68** | 4.29 | 6.67 | 4.99 |
|  | 1.75 | 13.82 | -5.46 | -6.35 | -7.05 | -5.34 | -4.84 | -9.26 | -6.12 | *-2.65* | 9.31 | 6.51 | 6.98 | **2.38** | 6.17 | 8.49 | 7.01 |
|  | 2 | 16.98 | -5.61 | -6.45 | -7.51 | -5.71 | -3.08 | -9.38 | -6.11 | **-2.29** | 9.97 | 4.22 | 7.23 | *2.79* | 6.33 | 8.59 | 6.87 |
| 2000 | 0 | 2.99 | -4.79 | -6.39 | -5.9 | -3.77 | -3.07 | -8.12 | -5.6 | -6.46 | -0.02 | -0.05 | *-0.01* | -0.05 | **0** | -0.05 | -0.06 |
|  | 0.25 | 3.00 | -4.45 | -5.98 | -5.77 | -2.93 | -3.04 | -7.63 | -4.14 | -4.08 | 0.64 | -1.15 | 0.66 | *0.12* | 0.57 | 0.72 | **0.11** |
|  | 0.5 | 3.67 | -4.8 | -6.45 | -5.86 | -4.22 | -3.9 | -8.1 | -4.8 | -4.22 | 1.76 | -2.54 | 1.46 | **0.45** | 2.06 | 1.8 | *0.64* |
|  | 0.75 | 4.61 | -4.6 | -6.2 | -5.22 | -2.59 | -3 | -7.92 | -5.15 | -4.06 | 3.27 | -3.66 | 2.42 | **0.62** | 2.39 | 3.3 | *1.01* |
|  | 1 | 5.99 | -4.77 | -6.39 | -6.19 | -4.61 | -2.14 | -8.03 | -5.67 | -3.56 | 4.77 | *0.86* | 3.11 | **0.6** | 2.93 | 4.18 | 2.43 |
|  | 1.25 | 7.48 | -4.68 | -6.29 | -5.66 | -3.2 | -1.78 | -8.09 | -5.92 | -3.12 | 6.73 | *1.77* | 4.66 | **1.22** | 3.23 | 6.07 | 4.55 |
|  | 1.5 | 10.33 | -4.77 | -6.3 | -5.94 | -2.86 | *-1.87* | -7.96 | -5.99 | -2.81 | 7.87 | 4.02 | 5.75 | **1.41** | 4.91 | 7 | 5.68 |
|  | 1.75 | 13.73 | -4.7 | -6.32 | -5.24 | -3 | **-0.76** | -8.07 | -5.82 | -2.3 | 9.27 | 3.31 | 6.61 | *1.9* | 5.11 | 8.01 | 6.33 |
|  | 2 | 17.68 | -4.65 | -6.32 | -5.44 | -3.35 | -2.33 | -8 | -5.75 | *-1.88* | 10.11 | **0.68** | 7.17 | 2.21 | 8.35 | 8.59 | 4.95 |
| 3500 | 0 | 2.99 | -4.32 | -5.94 | -4.98 | -2.43 | -2.51 | -7.37 | -4.13 | -4.27 | *0.03* | -0.03 | -0.03 | -0.04 | -0.08 | 0.03 | **0.01** |
|  | 0.25 | 2.97 | -4.45 | -6.06 | -5.05 | -2.79 | -2.07 | -7.54 | -4.71 | -4.5 | 0.71 | -1.46 | 0.73 | *0.07* | 0.83 | 0.74 | **0.05** |
|  | 0.5 | 3.50 | -4.22 | -5.83 | -4.95 | -2.34 | -1.98 | -7.25 | -4.33 | -3.69 | 1.84 | -2.67 | 1.43 | **0.3** | 1.72 | 1.82 | *0.65* |
|  | 0.75 | 4.45 | -4.46 | -6.16 | -5.13 | -2.75 | *-1.54* | -7.6 | -5.08 | -3.64 | 3.15 | -2.28 | 2.39 | **0.64** | 2.52 | 3.14 | 1.68 |
|  | 1 | 6.00 | -4.66 | -6.31 | -5.5 | -2.87 | *-1.3* | -7.72 | -5.62 | -3.48 | 4.92 | -2.06 | 3.43 | **0.67** | 2.96 | 4.55 | 2.02 |
|  | 1.25 | 7.50 | -4.26 | -5.88 | -4.72 | -2.18 | **0.16** | -7.29 | -5.45 | -2.86 | 6.58 | -1.34 | 4.34 | *0.95* | 4.54 | 5.46 | 3.22 |
|  | 1.5 | 10.40 | -4.2 | -5.89 | -4.68 | -1.8 | **-0.39** | -7.27 | -5.42 | -2.37 | 7.97 | 1.4 | 5.71 | *1.23* | 5.08 | 7.06 | 4.03 |
|  | 1.75 | 13.87 | -4.48 | -6.26 | -5.09 | -2.33 | **-0.46** | -7.63 | -5.78 | -2.3 | 9.11 | 4.94 | 6.44 | *1.7* | 4.99 | 7.78 | 6.21 |
|  | 2 | 17.16 | -4.47 | -6.27 | -5.68 | -1.73 | **-1.48** | -7.71 | -5.75 | -1.6 | 10.05 | 2.55 | 7.11 | 2.69 | 6.69 | 8.55 | 6.67 |
| 5000 | 0 | 3.12 | -4.28 | -5.94 | -4.76 | -2.6 | -2.26 | -7.22 | -3.78 | -4.05 | 0.07 | 0.05 | **0** | *0.01* | -0.11 | -0.04 | -0.05 |
|  | 0.25 | 3.06 | -4.14 | -5.88 | -4.72 | -2.49 | -2.18 | -7.1 | -3.99 | -3.83 | 0.72 | -1.42 | 0.73 | **0.05** | 1.2 | 0.84 | *0.19* |
|  | 0.5 | 3.59 | -4.13 | -5.76 | -4.92 | -2.25 | -1.93 | -7.03 | -4.24 | -3.65 | 1.71 | -2.44 | 1.44 | **0.26** | 2.13 | 1.85 | *0.46* |
|  | 0.75 | 4.56 | -4.3 | -5.96 | -5.1 | -2.23 | -1.5 | -7.17 | -4.75 | -3.52 | 3.17 | *-0.9* | 2.21 | **0.47** | 2.19 | 2.92 | 1.27 |
|  | 1 | 5.82 | -4.27 | -5.89 | -4.97 | -1.93 | -1.46 | -7.1 | -5.18 | -3.11 | 4.89 | **-0.22** | 3.31 | *0.7* | 2.68 | 4.36 | 1.89 |
|  | 1.25 | 7.69 | -4 | -5.64 | -4.34 | -2.04 | **0.12** | -6.86 | -5.33 | -2.63 | 6.54 | 4.26 | 4.45 | *0.92* | 4.74 | 5.65 | 4.17 |
|  | 1.5 | 10.31 | -4.3 | -6.05 | -5.1 | -2.42 | **-0.52** | -7.28 | -5.74 | -2.49 | 8.24 | 3.39 | 5.82 | *1.43* | 4.18 | 7.26 | 5.58 |
|  | 1.75 | 13.72 | -4.52 | -6.16 | -5.04 | -2.59 | **-1.04** | -7.4 | -5.88 | -2.11 | 9.14 | 4.9 | 6.47 | *1.98* | 5.79 | 7.9 | 6.7 |
|  | 2 | 17.32 | -4.27 | -6.02 | -4.84 | -1.91 | **-0.56** | -7.23 | -5.62 | *-1.67* | 9.96 | 3.5 | 7 | 1.97 | 6.6 | 8.45 | 6.43 |

Table 11: MSE of CATE estimation by shift intensity and training set size for different estimation methods, averaged over simulation runs (Simulation 2b (total shift between observational data and RCT)). For each setting, method achieving best performance printed in **bold** (second best in *italic*).

| Train size | Shift degree | KL Div. | CForest OS | S-learner OS | DR-learner OS | DR-learner Ridge | DR-learner Tree | T-learner OS | T-learner Ridge | T-learner Tree | CForest RCT | CForest wRCT | S-learner RCT | S-learner wRCT | DR-l RCT | T-learner RCT | T-learner wRCT |
|---|---|---|---|---|---|---|---|---|---|---|---|---|---|---|---|---|---|
| 500 | 0 | 2.98 | 46.53 | 61.3 | 111.73 | 82.85 | 83.54 | 108.92 | 45.41 | 51.38 | **5.22** | *5.28* | 18.27 | 18.27 | 7.07 | 16.15 | 7.33 |
| | 0.25 | 3.19 | 45.23 | 58.48 | 107.7 | 105.93 | 123.73 | 105.4 | 42.59 | 45.13 | **6.91** | *8.58* | 18.42 | 19.6 | 9.11 | 14.66 | 8.92 |
| | 0.5 | 3.62 | 47.1 | 59.48 | 104.44 | 81.08 | 100.4 | 108.1 | 36.17 | 35.04 | **13.45** | 23.04 | 20.44 | 23.6 | 15.97 | 26.37 | *13.64* |
| | 0.75 | 4.47 | 49.53 | 64.22 | 109.15 | 93.88 | 89.59 | 115.87 | 45.45 | 36.6 | 25.35 | 45.64 | 24.45 | 27.44 | *23.96* | 27.56 | **19.9** |
| | 1 | 5.78 | 48.78 | 63.54 | 124.72 | 93.16 | 108.59 | 112.39 | 45.89 | 32.91 | 43.62 | 106.32 | *30.65* | **30.28** | 36 | 40.86 | 33.74 |
| | 1.25 | 7.79 | 45.41 | 56.97 | 111.54 | 91.98 | 99.49 | 100.86 | 41.42 | **30.18** | 62.16 | 139.28 | 40.83 | *31.4* | 52.28 | 52.26 | 48.02 |
| | 1.5 | 10.33 | 48.76 | 61.09 | 120.58 | 83.81 | 152.2 | 113.71 | 45.95 | **33.42** | 88.73 | 91.14 | 56.46 | *36.34* | 70.6 | 59.34 | 63.67 |
| | 1.75 | 13.82 | 46.44 | 60.79 | 115.11 | 106.45 | 108.84 | 108.87 | 45.81 | **26.6** | 114.53 | 183.94 | 75.91 | *40.42* | 100.12 | 88.11 | 107.75 |
| | 2 | 16.98 | 46.1 | 60.06 | 113.18 | 102.82 | 174.57 | 108.76 | 44.54 | **23.56** | 126.01 | 125.37 | 78.09 | *39.27* | 99.86 | 93.8 | 85.42 |
| 2000 | 0 | 2.99 | 34.02 | 56.98 | 83.89 | 54.37 | 58.17 | 83.85 | 48.68 | 64.93 | **5.81** | *5.88* | 19.34 | 19.61 | 8.19 | 15.34 | 8.28 |
| | 0.25 | 3.00 | 29.49 | 50.98 | 76.11 | 49.77 | 60.99 | 75.93 | 25.82 | 25.96 | **6.98** | 10.82 | 18.25 | 20.02 | *9.11* | 17.66 | 9.61 |
| | 0.5 | 3.67 | 33.12 | 55.6 | 72.25 | 46.41 | 46.75 | 81.12 | 30.23 | 26.36 | **13.76** | 37.77 | 19.88 | 24.6 | 14.86 | 29.09 | *13.93* |
| | 0.75 | 4.61 | 31.86 | 54.78 | 80.56 | 49.81 | 55.36 | 81.86 | 36.61 | 28.8 | *27.04* | 154.84 | **25.33** | 28.49 | 27.44 | 41.5 | 28.09 |
| | 1 | 5.99 | 33.65 | 55.77 | 72.6 | 46.45 | 60.35 | 80.97 | 38.76 | **23.86** | 44.3 | 63.07 | *31.02* | 31.36 | 37.28 | 39.17 | 32.74 |
| | 1.25 | 7.48 | *33.01* | 56.3 | 87.1 | 50.7 | 64.66 | 84.99 | 43.36 | **23.91** | 70.32 | 67.38 | 46.2 | *33.52* | 61.94 | 67.42 | 56.73 |
| | 1.5 | 10.33 | *33.69* | 55.48 | 77.48 | 51.87 | 70.75 | 81.06 | 43.36 | **22.2** | 88.01 | 93.85 | 58.31 | *34.63* | 74.66 | 66.4 | 71.68 |
| | 1.75 | 13.73 | *31.71* | 54.91 | 79.32 | 50.4 | 76.92 | 82.71 | 42.06 | **19.67** | 111.74 | 147.67 | 68.42 | *35.19* | 89.55 | 65.77 | 82.72 |
| | 2 | 17.68 | *32.33* | 56.07 | 67.16 | 51.21 | 58.54 | 82.06 | 41.47 | **19.17** | 129.68 | 111.65 | 77.32 | *38.03* | 100.22 | 131.54 | 79.72 |
| 3500 | 0 | 2.99 | 28.05 | 51.13 | 72.69 | 38.9 | 48.91 | 72.77 | 28.94 | 31.78 | **5.44** | *5.67* | 18.39 | 18.49 | 8.02 | 15.26 | 7.95 |
| | 0.25 | 2.97 | 28.66 | 51.48 | 70 | 38.2 | 51.76 | 74.17 | 38.98 | 32.72 | **7.07** | 11.5 | 18.86 | 21.25 | *9.13* | 16.56 | 9.78 |
| | 0.5 | 3.50 | 27.74 | 50.57 | 66.08 | 38.05 | 47.27 | 72.25 | 26.5 | 24.19 | **14.89** | 46.8 | 20.9 | 27.29 | *16.06* | 22.96 | 16.65 |
| | 0.75 | 4.45 | 29.41 | 52.81 | 79.39 | 44.81 | 49.55 | 75.39 | 33.09 | *23.02* | 26.21 | 103.1 | 24.51 | 28.51 | 24.75 | 41.25 | **21.19** |
| | 1 | 6.00 | 30.62 | 53.15 | 64.92 | 38.36 | 58.88 | 75 | 39.02 | **21.75** | 45.43 | 127.32 | 33.08 | *29.67* | 41.02 | 47.21 | 34.11 |
| | 1.25 | 7.50 | *27.33* | 49.97 | 66.27 | 38.69 | 78.7 | 71.01 | 37.88 | **20.5** | 68.1 | 173.54 | 42.18 | *32.84* | 52.77 | 65.16 | 45.2 |
| | 1.5 | 10.40 | *27* | 50.26 | 68.44 | 47.33 | 68.79 | 70.92 | 38.41 | **20.9** | 89.94 | 112.42 | 57.41 | *33.69* | 75.11 | 81.29 | 60.63 |
| | 1.75 | 13.87 | *29.01* | 53.2 | 68.33 | 44.42 | 72.64 | 74.55 | 42.47 | **18.1** | 109.79 | 136.09 | 66.94 | *36.19* | 86.19 | 68.07 | 80.93 |
| | 2 | 17.16 | *29.14* | 53.3 | 66.32 | 47.85 | 58.99 | 75.38 | 41.67 | **16.48** | 128.92 | 113.56 | 77.84 | *44.29* | 100.65 | 95.59 | 91.14 |
| 5000 | 0 | 3.12 | 26.08 | 48.7 | 60.13 | 31.96 | 38.88 | 67.99 | 23.98 | 26.11 | **5.42** | *5.5* | 17.6 | 17.51 | 7.27 | 13.99 | 7.38 |
| | 0.25 | 3.06 | 25.26 | 48.91 | 59.2 | 35.38 | 41.97 | 66.92 | 24.3 | 23.91 | **7.48** | 13.7 | 19.15 | 21.72 | *9.84* | 18.6 | 10.69 |
| | 0.5 | 3.59 | 25.51 | 47.96 | 58.62 | 35.94 | 39.65 | 66.99 | 25.84 | 22.22 | **13.46** | 37.92 | 19.55 | 24.89 | *14.6* | 27.59 | 15.32 |
| | 0.75 | 4.56 | 26.09 | 48.06 | 56.54 | 37.49 | 39.72 | 66.26 | 29.67 | *20.88* | 25.74 | 53.22 | 23.56 | 27.51 | 24.02 | 40.76 | **20.13** |
| | 1 | 5.82 | *26.2* | 48.63 | 60.2 | 35.77 | 39.01 | 67.02 | 35.05 | **20.49** | 44.87 | 100.33 | 31.93 | *30.69* | 38.84 | 46.02 | 38.92 |
| | 1.25 | 7.69 | *24.7* | 47.38 | 64.71 | 42.64 | 57.49 | 65.43 | 37.81 | **19.17** | 67.81 | 108.59 | 43.38 | *33.45* | 54.84 | 61.97 | 50.91 |
| | 1.5 | 10.31 | *26.65* | 50.79 | 63.56 | 43.44 | 54.69 | 69.8 | 42.46 | **17.81** | 94.26 | 124.26 | 59.78 | *35.38* | 78.68 | 69.38 | 89.64 |
| | 1.75 | 13.72 | *28.42* | 50.98 | 62.66 | 50.51 | 59.41 | 69.61 | 42.37 | **16.1** | 110.83 | 190.54 | 68.43 | *37.49* | 89.63 | 90.15 | 88.29 |
| | 2 | 17.32 | *25.92* | 49.49 | 58.89 | 41.56 | 61.87 | 67.18 | 40.06 | **15.78** | 126.05 | 90.74 | 74.91 | *35.83* | 98.03 | 85.04 | 77.14 |

### D.3 WHI Data Application

**Data**   We consider a case study using clinical trial and observational data from the Women's Health Initiative (Machens and Schmidt-Gollwitzer, 2003). A focus of this study was to investigate the effectiveness of hormone replacement therapy (HRT) treatment in preventing the onset of chronic (cardiovascular) diseases. As the observational study and clinical trial data led to conflicting findings, the WHI study has become a prime example of how confounding in observational data can introduce bias and, in this case, suggest overly optimistic results (for more detail, see Kallus and Zhou 2018). In this setting, we study how multi-accurate CATE estimators that are "warm-started" with observational data and have access to small samples from the clinical trial compare to estimators that draw on either observational or clinical trial data only.

We aim to assess the effect of HRT treatment on systolic blood pressure as a major risk factor for cardiovascular diseases. We estimate the CATE with respect to two sets of covariates – a small (age, ethnicity) and an extended set (age, ethnicity, number of cigarettes per day, systolic blood pressure baseline, diastolic blood pressure baseline, BMI baseline; see Table 12).

In our application setting, we start with the observational study (OS) (52,335 observations) and draw a random 50% sample that serves as observational training data for (naive) CATE estimation. We split the clinical trial data (14,531 observations) into an initial 50% training set and a 50% test set. The initial training set is used to draw further random samples of size $\{250, 500, 750, 1000, 1250, 1500\}$ that serve as clinical trial (CT) training data. For each CT training set size, sampling is repeated 25 times.

**CATE estimation**   We use the following methods for estimating the CATE based on the training set from the observational study.

- (**CForest-OS**) Causal forest (Wager and Athey, 2018) trained in the training set of the observational data.

- (**S-learner-OS**) S-learner using random forest to learn a joint outcome model for treated and untreated in the training set of the observational data.

- (**DR-learner-OS**) DR-learner (Kennedy, 2023) using regression forest to learn separate outcome models for treated and untreated in the training set of the observational data.

- (**T-learner-OS**) T-learner using regression forest to learn separate outcome models for treated and untreated in the training set of the observational data.

We estimate DR-learner and T-learner using multi-calibration boosting with samples of clinical trial data. The MCBoost hyperparameter settings are shown in Table 13. Hyperparameters of the baseline CATE learner are listed in Table 14.

- (**DR-learner-MC-Ridge**) DR-learner using regression forest in the training set of the observational data is post-processed with MCBoost using with ridge regression in the training set of the clinical trial data.

- (**DR-learner-MC-Tree**) DR-learner using regression forest in the training set of the observational data is post-processed with MCBoost using with decision trees in the training set of the clinical trial data.

- (**T-learner-MC-Ridge**) T-learner using regression forest in the training set of the observational data is post-processed with MCBoost using with ridge regression in the training set of the clinical trial data.

- (**T-learner-MC-Tree**) T-learner using regression forest in the training set of the observational data is post-processed with MCBoost using with decision trees in the training set of the clinical trial data.

We further compare to the following CATE learner that are solely based on clinical trial data.

- (**CForest-CT**) Causal forest trained in the training set of the clinical trial data.
- (**S-learner-CT**) S-learner using random forest to learn a joint outcome model for treated and untreated in the training set of the clinical trial data.
- (**T-learner-CT**) T-learner using random forest to learn separate outcome models for treated and untreated in the training set of the clinical trial data.

We infer the "true" CATE by applying the following methods to the test set of the clinical trial data.

- (**RL-NET**) R-learner (Nie and Wager, 2020) using elastic net as base learner.
- (**TL-NET**) T-learner using elastic net as base learner.
- (**XL-RF**) X-learner (Künzel et al., 2019) using random forest as base learner.

**Evaluation**  We compare the outlined methods with respect to the bias in ATE and MSE in CATE estimation in the test set of the clinical trial data. To evaluate bias, we use the observed difference in outcomes by treatment condition in the clinical trial, $\hat{\text{ATE}}_{obs} = \frac{\sum TY}{\sum T} - \frac{\sum (1-T)Y}{\sum (1-T)}$, as the estimate of the true ATE and evaluate against the respective mean of $\hat{\tau}$ of the various CATE estimation methods.

$$\text{Bias} = \hat{\text{ATE}}_{obs} - \frac{1}{n}\sum \hat{\tau}(x)$$

In evaluating MSE, we use the estimated CATE function, $\tau^*(x)$, based on learners that had privileged access to the clinical trial test data (XRF, RL, TL) as a substitute for the true $\tau(x)$ and evaluate against $\hat{\tau}(x)$ of the CATE estimation methods outlined above (using the observational and/or clinical trial training data only).

$$\text{MSE} = \frac{1}{n}\sum (\tau^*(x) - \hat{\tau}(x))^2$$

**Results**  Figure 13a (small set of covariates) and Figure 13b (extended set) show the bias of the estimated ATE for each method by clinical trial training set size. As expected, learning in the clinical trial training data allows for unbiased estimation of the ATE as shown by the three CT-based methods in both settings. These estimates, however, come with high variability if the CT training data is small. Learning solely in the observational data incurs bias in ATE estimation, particularly in settings where the CATE learner only have access to a small set of covariates (Figure 13a). In this case, post-processing with clinical trial data improves upon the initial T-learner. Given an extended set of covariates the bias of the observational data-based methods decreases and post-processing is less effective (Figure 13b).

We evaluate the MSE of the estimated CATE in Figure 14 (small set of covariates) and Figure 15 (extended set) by clinical trial training set size against the three approximations of the true CATE that are based on the clinical trial test data. The observational data-based methods generally outperform the CT-based CATE estimates, indicating that the small clinical trial training sets on their own are not sufficient for accurate CATE estimation (comparing Figure 14a to 14b). Post-processing the initial T-learner via multi-calibration boosting with clinical trial data allows to achieve the smallest MSE for most CT training set sizes and true CATE estimation techniques in the limited covariate setting (Figure 14a). As the observational data-based CATE learner achieve low MSE with the extended set of covariates, post-processing shows no improvement in this case (Figure 15a).

Table 12: Sample composition (averages and proportions) of the observational study and clinical trial of the WHI data.

|  | OS | | | RCT | | |
|---|---|---|---|---|---|---|
|  | Overall | $T=0$ | $T=1$ | Overall | $T=0$ | $T=1$ |
| Treatment | 0.33 | | | 0.50 | | |
| Systolic blood pressure | 124.83 | 125.88 | 122.68 | 125.54 | 125.30 | 125.78 |
| Systolic blood pressure baseline | 125.09 | 126.24 | 122.75 | 127.65 | 127.69 | 127.61 |
| Diastolic blood pressure baseline | 74.56 | 74.78 | 74.12 | 75.68 | 75.78 | 75.59 |
| BMI baseline | 26.83 | 27.29 | 25.88 | 28.52 | 28.53 | 28.50 |
| Age | 62.52 | 63.43 | 60.68 | 63.37 | 63.37 | 63.37 |
| Cigarettes per day | | | | | | |
| 0 | 0.53 | 0.54 | 0.51 | 0.52 | 0.52 | 0.51 |
| <1 | 0.02 | 0.02 | 0.02 | 0.02 | 0.02 | 0.02 |
| 1-4 | 0.09 | 0.09 | 0.09 | 0.08 | 0.07 | 0.08 |
| 5-14 | 0.15 | 0.15 | 0.15 | 0.15 | 0.16 | 0.15 |
| 15-24 | 0.13 | 0.12 | 0.14 | 0.14 | 0.14 | 0.15 |
| 25-34 | 0.04 | 0.04 | 0.05 | 0.05 | 0.05 | 0.05 |
| 35-44 | 0.03 | 0.03 | 0.03 | 0.03 | 0.03 | 0.03 |
| 45+ | 0.01 | 0.01 | 0.01 | 0.01 | 0.01 | 0.01 |
| Ethnicity | | | | | | |
| White | 0.89 | 0.87 | 0.92 | 0.84 | 0.84 | 0.84 |
| Black | 0.05 | 0.07 | 0.02 | 0.07 | 0.07 | 0.06 |
| Hispanic | 0.03 | 0.03 | 0.02 | 0.05 | 0.05 | 0.05 |
| American Indian | 0.00 | 0.00 | 0.00 | 0.00 | 0.00 | 0.00 |
| Asian/Pacific Islander | 0.02 | 0.02 | 0.03 | 0.02 | 0.02 | 0.02 |
| Unknown | 0.01 | 0.01 | 0.01 | 0.01 | 0.01 | 0.01 |

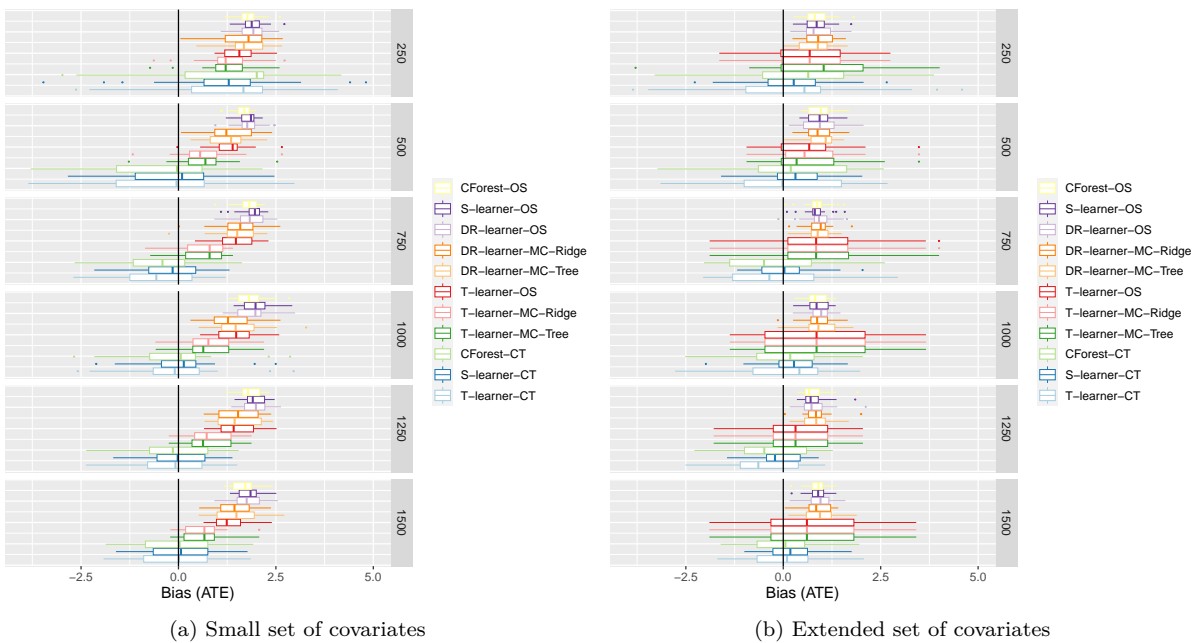

(a) Small set of covariates  (b) Extended set of covariates

Figure 13: Bias by clinical trial training set size (WHI Data Application). The distribution of bias scores over sampling repetitions is plotted. Post-processing initial T-learner with clinical trial data improves over T-learner-OS in the limited covariate setting.

Table 13: Hyperparameter settings for post-processing using MCBoost. Default settings are used for parameters not listed.

(a) T-learner MC

| Method | Implementation | Hyperparameter | Value |
|--------|----------------|----------------|-------|
| Ridge | `mcboost` | `max_iter` | 10 |
| | | `alpha` | 1e-06 |
| | | `eta` | 0.1 |
| | | `weight_degree` | 2 |
| | `glmnet` | `alpha` | 0 |
| | | `s` | 1 |
| Tree | `mcboost` | `max_iter` | 10 |
| | | `alpha` | 1e-06 |
| | | `eta` | 0.1 |
| | | `weight_degree` | 2 |
| | `rpart` | `maxdepth` | 3 |

(b) DR-learner MC

| Method | Implementation | Hyperparameter | Value |
|--------|----------------|----------------|-------|
| Ridge | `mcboost` | `max_iter` | 5 |
| | | `alpha` | 1e-06 |
| | | `eta` | 0.1 |
| | | `weight_degree` | 2 |
| | `glmnet` | `alpha` | 0 |
| | | `s` | 1 |
| Tree | `mcboost` | `max_iter` | 5 |
| | | `alpha` | 1e-06 |
| | | `eta` | 0.1 |
| | | `weight_degree` | 2 |
| | `rpart` | `maxdepth` | 3 |

Note: `eta = 0.01` in extended set of covariates setting.

Table 14: Hyperparameter settings of (baseline) CATE learners. Default settings are used for parameters not listed.

| Method | Implementation | Hyperparameter | Value |
|---|---|---|---|
| CForest | `grf` | `num.trees` | 2000 |
| | | `mtry` | `sqrt(p)+20` |
| | | `sample.fraction` | 0.5 |
| | | `honesty.fraction` | 0.5 |
| | | `min.node.size` | 5 |
| T-,DR-learner | `regression_forest` | `num.trees` | 2000 |
| | | `mtry` | `sqrt(p)+20` |
| | | `sample.fraction` | 0.5 |
| | | `honesty.fraction` | 0.5 |
| | | `min.node.size` | 5 |
| S-learner | `ranger` | `num.trees` | 500 |
| | | `mtry` | `sqrt(p)` |
| | | `min.node.size` | 5 |
| | | `replace` | TRUE |
| | | `sample.fraction` | 1 |

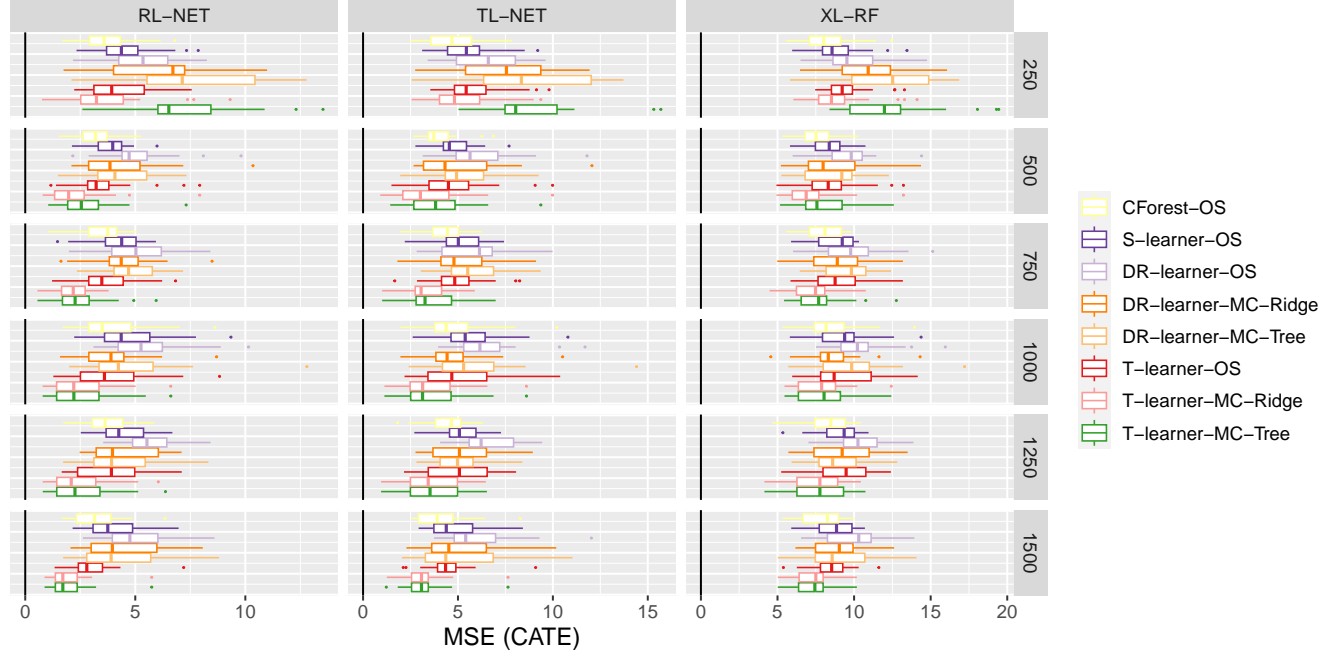

(a) Observational data-based and multi-accurate CATE estimation

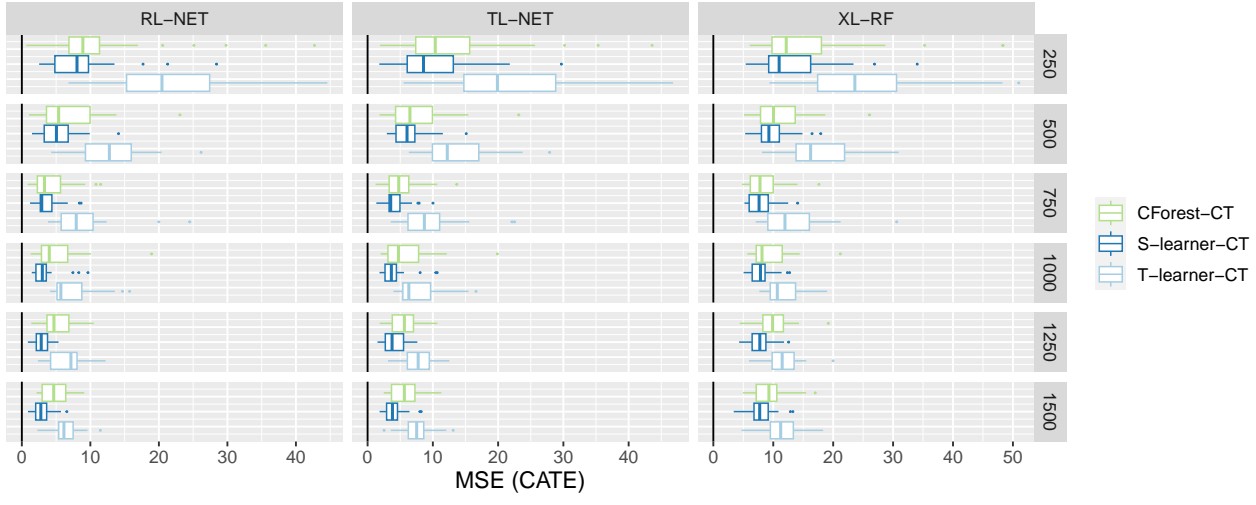

(b) Clinical trial data-based CATE estimation

Figure 14: MSE by 'true' CATE estimation method and clinical trial training set size with small set of covariates (WHI Data Application). The distribution of MSE scores over sampling repetitions is plotted. T-learner-MC-Ridge outperforms other methods for most CT training set sizes and true CATE estimation techniques RL-NET and TL-NET.

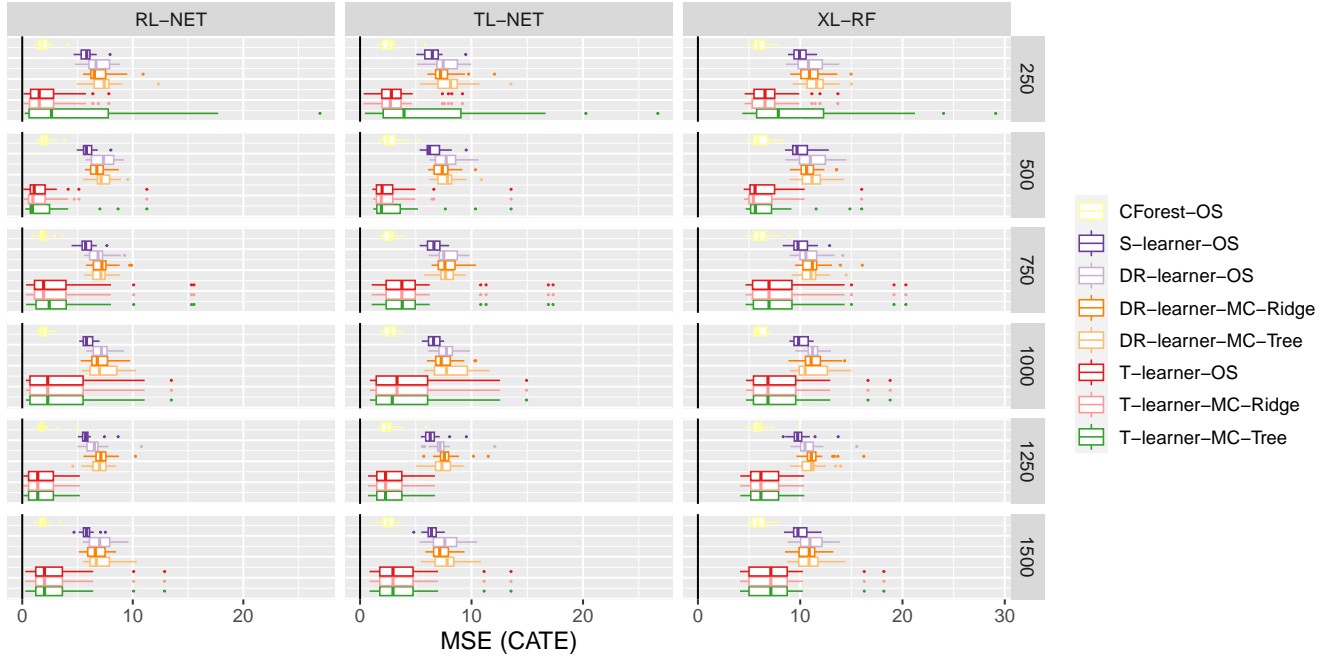

(a) Observational data-based and multi-accurate CATE estimation

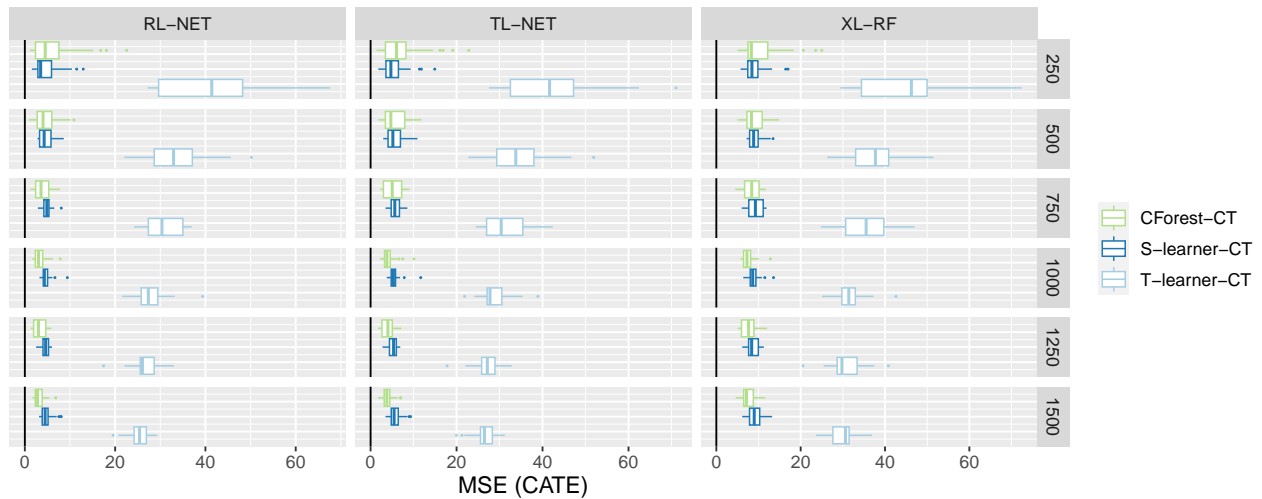

(b) Clinical trial data-based CATE estimation

Figure 15: MSE by "true" CATE estimation method and clinical trial training set size with extended set of covariates (WHI Data Application). The distribution of MSE scores over sampling repetitions is plotted. Multicalibration boosting yields little improvement as CForest-OS and T-learner-OS already achieve low MSE.

