# OpenReview forum: "Multi-Accurate CATE is Robust to Unknown Covariate Shifts"
_TMLR — Accepted by TMLR_

### Review · Reviewer_Hf3b · 2024-07-31

**Summary Of Contributions:**

The authors propose a method to learn multi-accurate predictors to postprocess CATE T-learners (differenced regressions) to become robust to unknown covariate shifts at the time of deployment.

**Audience:**

Yes

**Claims And Evidence:**

Yes

**Requested Changes:**

Please refer to the weaknesses.

**Strengths And Weaknesses:**

Strengths: The method works in general for pseudo-outcome regression, such as the DR-learner. They show how this approach can combine (large) confounded observational and (smaller) randomized datasets by learning a confounded predictor from the observational dataset, and auditing for multi-accuracy on the randomized controlled trial.


They show improvements in bias and mean squared error in simulations with increasingly larger covariate shift, and on a semi-synthetic case study of a parallel large observational study and smaller randomized controlled experiment.


Overall, the authors establish a connection between methods developed for multi-distribution learning and achieve appealing desiderata (e.g. external validity) in causal inference and machine learning.

Weaknesses:

What are the hyperparameters setup?

Theoretically speaking, what are the advantages of the proposed method?

---

> ### Author Response · Authors · 2024-08-29
> **Response to review**
>
> We clarified the hyperparameter setup of all baseline and multi-calibrated CATE learners used in both the simulations and the real data applications. Details are given in Table 2, 3 (simulation setup) and Table 13, 14 (WHI data application) in the revised appendix D. Note that appendix D.1 also lists all software packages used and links to an OSF repository which includes all code files needed for replication.
>
> The theoretical advantages of this method are provable robustness of prediction error bias under *unknown covariate shifts*, so long as the true shift function lies within the set of test functions for multi-accurate learning. We show that theoretically this can occur without loss of efficiency, so long as the set of test functions $\mathcal{H}$ is rich/complex enough: our Proposition 3 shows the multi-accurate learning procedure implicitly approximates a "simpler" version of a state-of-the-art CATE learner, the DR-learner. This is an important contribution because multi-accuracy *alone* *only* implies control of prediction bias, not variance.
>
> Thanks for the question, we have added this summary clarification in the introductory discussion of contributions.

---

### Review · Reviewer_wUVC · 2024-08-16

**Summary Of Contributions:**

The paper applies recent advances in algorithmic fairness to the estimation of CATE. Two settings are considered: 1) a CATE estimator is trained on an unconfounded dataset and the goal to deploy it on another unconfounded dataset; and 2) a CATE estimator is trained on a large but confounded dataset and the goal is to deploy it on a small RCT dataset.  In both cases, it is expected that the estimators are valid on the target datasets, after post-processing estimators on target datasets by the multi-calibration technique from the algorithmic fairness literature.

**Audience:**

Yes

**Claims And Evidence:**

No

**Requested Changes:**

I think the whole project should be reworked, and a whole new paper should be written. So please take my following comments as suggestions for future work.

### Method

The current paper apparently says multi-calibration can deal with arbitrary covariate shifts. However, the problem addressed by Kim et al. (2022) is roughly the adaptation of the averaged outcome estimator from one of the treatment/control groups to the other group. That is, the difference between the two groups (populations) can be *completely specified* by the true propensity score, although the true propensity score is *unknown*.

Perhaps the most obvious mistake in the current paper is that the multi-calibration technique is applied to the conditional outcome $\mu_t(X)$. However, *what is multi-calibrated should be a propensity score* which, as stated in the 1st point, lies at the core of the technique.
- The domain of the function $\tilde{p}$ in Def 1 and 2, taken from the previous work, also shows this clearly. This is an example of taking cursory notes from the previous work without much thinking (or it can be noticed that multi-calibration should be applied to propensity scores).

The identification results in the current paper make no sense. In practice, Kim et al. (2022) address the problem of ATE error caused by the error of propensity score estimation, so that the former has a known bound if the latter is bounded (Universal Adaptability). That is, *the original work deals with the problem of estimation but not identification*, and, in fact, the identification, including under those settings in the current paper, is just textbook results of identification under unconfoundedness.

### Writing

The paper does not use consistent terminology. For example, reference population and target population refer to the same thing.

Another general problem is unexplained terminology and symbols. For example,

- reference/target population is not really explained in the Intro and their meaning can only be guessed from later parts;
- "<<" and "P-a.s." in Setting 1.

Some explanations are not in the right place. Examples:

- “pseudo-outcome regression” is mentioned in the Intro and Background, but only defined as late as page 11;

- “MC-Boost” and “Auditing” are in Figure 1, but are explained nearly 3 pages later, and this makes Figure 1 not very meaningful.

Insufficient/misleading explanations. Examples:

- why it is required that the conditional expectation equals to 1 in the Setting 1?

-  in Figure 1 for Setting 2, "unknown $Q$" is mentioned, while it is only used in the main text for Setting 1 but not Setting 2;

- at the beginning of Sec 3.1, the eqs do not show the difference between S and T learners because $\mu_t(x)$ is an equivalent way to write $\mu(x, t)$; and it puzzles me why S-learner is mentioned here because S-learner is never mentioned again in the main text.

- the paper spends quite some space on MCBoost and auditing while it adds no intuition but confusion, e.g., "residuals" and "twisting" are not clearly explained.

**Strengths And Weaknesses:**

### Strengths

Applying multi-calibration technique to the estimation of CATE is an interesting research direction.

### Weaknesses

Unfortunately, the proposed method is based on misunderstandings of the original work (Kim et al. 2022) and is fundamentally flawed.

The writing is very sloppy, and the paper reads like cursory notes based on a quick read and application of the previous work.

---

> ### Author Response · Authors · 2024-08-29
> **Response to review**
>
> *Alleged misconception of Kim et al. (2022).*
>
> The reviewers assessment of Kim et al. (2022) and of its relation to the present paper is incorrect. Universal Adaptability (UA) as originally introduced is not about the adaptation of the average outcome estimator from one of the treatment/control groups to the other group. It is about the general ability of an average outcome estimator, based on a prediction function trained with source data, to adapt to arbitrary (external) covariate shifts, i.e. when deployed under *multiple* different target distribution. The difference between the source distribution and *each* target distribution can be specified by a propensity score function. A predictor is universally adaptable if its (average outcome) estimation error is competitive with the error of the *respective target-specific* propensity score-based estimator for any target distribution - a property which can be achieved by multi-accurate predictors.
>
> Most importantly, in Kim et al. (2022), as in our paper, the multi-calibration technique is applied to the *outcome prediction model(s)*, not to a propensity score model, since the goal is to achieve target-independent robustness. ($p$ simply stands for predictor, not propensity score.) While the multi-accuracy criterion (Def 1) has indeed been originally introduced by Kim et al. (2019) in the classification context, the associated post-processing procedure can be (and has been) readily applied to other types of outcome variables (see Pfisterer et al. (2021) and the documentation of the mcboost package). That is, the grounds on which you base your assessment of the paper are simply factually incorrect.
>
> In the present work, we show that multi-accuracy also implies robust estimation of the ATE (i.e., robust to "internal" shifts) and this result is used as a stepping stone to obtain universally adaptable CATE functions, which again focus on the *multi-target* populations setting (i.e., robust to "external" shifts). Both (re-interpretation of UA as robust ATE estimation and extension to multi-target CATE) are contributions solely of the present paper which we acknowledge was not cleanly distinguished in the original manuscript but is now clarified.
>
> In summary, we understand that cleanly introducing the setup is crucial and have carefully revised our presentation of the original work of Kim et al. (2022) in light of our extensions/ adaptations throughout the paper. We hope our clarifications allow the reviewer to re-consider their (strong) claims, which unfortunately are themselves rooted in misconceptions of this line of work.
>
> We have uploaded the revision and our clarifications early to allow enough time for any potential follow-up questions or additional clarifications we may provide.

---

> > ### Author Response · Authors · 2024-08-29
> > **Response to review**
> >
> > *Identification results.*
> >
> > *identification.. is just textbook .. unconfoundedness*. We believe this is a simple misunderstanding. Yes, the identification in the warm-up Proposition 1 is based on standard doubly-robust estimation of the ATE. What is **surprising** is that this is **achievable** with the estimator based on **multi-accurate outcome** models, under assumptions of how the test function class $\mathcal{H}$ relates to inverse propensity scores. Indeed, while our estimator, $\mathrm{E}[\tilde{\mu}_1(X)-\tilde{\mu}_0(X)],$ does not explicitly use propensity scores, Proposition 1 illustrates how it nonetheless approximates a DR estimator which does use propensity scores. Multi-accuracy ensures that $ \mathrm{E}[(Y-\tilde{\mu}_1)h(X) \mid T=1] \leq \epsilon, \forall h \in \mathcal{H}.$ Indeed, concluding -- showing that the DR estimator approximates the ATE -- is standard. But the rest is a novel perspective on the connection between multi-accurate estimation and causal inference.
> >
> > We believe your confusion exactly reflects the novelty and significance of our developments. It *is surprising* that estimating the ATE with multi-accurate outcome models can approximate a DR estimator with propensity scores, which is why we start in a standard ATE example where unconfoundedness holds. We will add the following notes to clarify; when introducing our estimator:
> >
> > > This represents estimating the ATE by imputing potential outcomes via regression... Notably, the estimator \textit{does not} explicitly use propensity scores. Nonetheless, we show how the robust test functions in multi-accurate auditing  approximate inverse propensity score functions. Next we develop identification results based on this perspective.
> >
> > and right after Proposition 1:
> >
> > > Although identification of the ATE with doubly-robust estimator is standard, note that our estimator that uses multi-accurate outcome models *does not* explicitly estimate the propensity scores. Nonetheless, the novelty of Prop. 1 is that averaging multi-accurate outcome models, because multi-accuracy implies $\mathrm{E}[(Y-\tilde{\mu}_t(X)) h(X) ]\leq \epsilon, \forall h(X),$ leads to direct approximation of a doubly-robust estimator with the \textit{multi-accurate regression-based} estimator $\mathrm{E}\left[\tilde{\mu}_1(X)-\tilde{\mu}_0(X)\right]$, under conditions on how the test function class $\mathcal{H}$ relates to inverse propensity score functions. Because this is a novel perspective, we establish it for ATE estimation first before showing how it applies to the robust covariate shifts that motivate our method.

---

> > > ### Author Response · Authors · 2024-08-29
> > > **Response to review**
> > >
> > > *Terminology.*
> > >
> > > We thank the reviewer for pointing out inconsistencies in the terminology and in the placement of some explanations, which were addressed as follows.
> > >
> > > - "Target" population is a somewhat standard term in covariate shift, interchangeable with reference population. To be absolutely clear we revised the manuscript to only use the term target population.
> > >
> > > - "$\ll$" and "P-a.s."
> > > This is a minor interdisciplinary wrinkle: these are both standard notations in measure theory, "$\ll$" describes absolute continuity of two probability measures, and "P-a.s." means ``almost surely with respect to the measure $P$". Recognizing we should be self-contained, we replace with the more explicit statement about finiteness of the likelihood ratio where distribution $P$ is supported, $P(x)>0$:
> > >
> > > \begin{equation}
> > > \mathcal{L_1} := \left[ \frac{d Q_X(x)}{d P_{X_1}(x)}: \mathbb{P} \left[ \frac{d Q_X(x)}{d P_{X_1}(x)} < \infty,\forall x \text{ s.t. } P(x)>0 \right] =1,
> > > E[ \frac{d Q_X(X)}{d P_{X_1}(X)}] = 1 \right]
> > > \end{equation}
> > >
> > > i.e. $\mathcal{L}_1$ is the set of valid likelihood ratios. Though this is more cumbersome we believe it addresses your concern about shorthand. We will further include the explanation:
> > >
> > > > $\mathcal{L}_1$ is the set of valid covariate shift likelihood ratios, such that $Q$ is absolutely continuous w.r.t $P$, i.e. the likelihood ratio is finite whenever $P$ has nonzero support (almost surely, with probability 1 under the measure $P$), and marginalizes to $1$.
> > >
> > > - We clarified that "pseudo-outcome" refers to the doubly robust score in the DR learner setup in the introduction.
> > >
> > > - We clarified the term "MC-Boost auditing" in the caption of Figure 1, as it relates to the initial discussion of multi-accuracy in the introduction section.

---

> > > > ### Author Response · Authors · 2024-08-29
> > > > **Response to review**
> > > >
> > > > *Re: ``Insufficient/misleading explanations."*
> > > >
> > > > We respectfully disagree that any of these are misleading. These are minor stylistic concerns that we are happy to smooth over, which we can do so with very minor adjustments.
> > > >
> > > > - *... conditional expectation equals to 1 in the Setting 1?*
> > > >
> > > > This is an overly general notation that could encapsulate *outcome shifts* such as $\frac{Q(Y\mid X)}{P(Y\mid X)}.$ Then the conditional expectation equals 1 by definition of the expectation, then $\int_y \frac{P(y\mid x)}{Q(y\mid x)} P(y\mid x) dy=1$. However since in the main text we focus on *covariate shift* we propose to omit that constraint for clarity. See  $\mathcal{L}_1$ above for the updated statement.
> > > >
> > > > - *in Figure 1 for Setting 2, ``unknown $Q$" is mentioned, while it is only used in the main text for Setting 1 but not Setting 2*
> > > >
> > > > This is not true: it's used in the main text in the section on observational and randomized data, see ``Identification of target-independent CATE" under Section 3.4.
> > > >
> > > > In the short introduction of Setting 2 in pg 6, it is implied that analogous considerations carry over, which we omit for brevity. To be absolutely clear we will add:
> > > >
> > > > > For covariate shift, consider likelihood ratio functions with respect to a target population $Q$. Again we seek an estimator $\hat\tau(X)$ with low bias under $Q:$ $\left|\mathrm{E}_Q[(\hat{\tau}(X)-\tau(X))]\right| \leq \epsilon.$.
> > > >
> > > > - *at the beginning of Sec 3.1, the eqs do not show the difference between S and T learners because  $\mu_t(X)$ is an equivalent way to write $\mu(x, t)$; and it puzzles me why S-learner is mentioned here because S-learner is never mentioned again in the main text.*
> > > >
> > > > This was simply to provide a comprehensive background. We removed mention of the $S$-learner.
> > > >
> > > > - *the paper spends quite some space on MCBoost and auditing while it adds no intuition but confusion, e.g., "residuals" and "twisting" are not clearly explained.*
> > > >
> > > > We included paragraphs (bottom of submitted page 7) with additional intuitions for MCBoost, the full pseudocode for which we included in the appendix. Respectfully, "residuals" is a standard term in machine learning: it means $(Y-p(X))$, i.e. the prediction errors.
> > > >
> > > > Again, for an interdisciplinary paper some terms like "twicing" will be familiar to people with certain backgrounds (i.e, statistical). We hear your concern that those with different backgrounds will not be aware of these other interdisciplinary terms. We can include more explanations to make these interdisciplinary messages more self-contained.
> > > >
> > > > To expand intuitions, we have added:
> > > >
> > > > > "Twicing" is a one-step boosting procedure: given a predictor function $p(x)$, fit the residual errors $c^*\in \arg\min_c\mathrm{E}[((Y-p(X)) -c(X) )^2]$ and return $p(x)+c^*(x)$ as the final prediction.}  ...    {For auditing over regression prediction functions, for step $k$, this is a regression problem: the auditing step solves $\min_{c\in\mathcal{C}} \mathrm{E}[((Y-p_k(X)) - c(X))^2]$. }The predictor is then updated with a multiplicative-weights update based on the worst-case test function $c^*(X)${, that is, the next predictor $p_{k+1}(x)$ is upweighted or downweighted based on the predicted error $c^*(x)$, $p_{k+1}(x) \propto e^{-\Delta_c \cdot c^*(x) / 2} \cdot p_k(x),$ where $\Delta_c$ is the miscalibration error computed on a validation set, $\mathrm{E}_V[c^*(X)(Y-p_k(X))]$. Therefore the MCBoost procedure adjusts the original predictor based on the predicted errors ($c^*(x)$), adjusting more strongly in regions of higher error, as well as if the predictor incurs higher error on average.

---

> > ### Comment · Reviewer_wUVC · 2024-09-02
> > **Thanks for the rebuttal**
> >
> > After reading this part of the rebuttal and a re-reading of Kim et al. (2022), I admit there are some confusions to be cleared before making a final evaluation. And I hope the authors could help in this regard.
> >
> > I think my main confusion is, how can $\tilde{p}$, the range of which is $[0, 1]$, be a valid “surrogate estimator” of $Y$ which is on $\mathbb{R}$? Is this line of work only for binary outcomes? I couldn’t find an answer in Kim et al. (2022) and your paper.
> >
> > Related to the above, are the “outcome prediction models” you mentioned just $\tilde{p}$? But, if so, are the T-learner and DR-learner also restricted to binary outcomes?
> >
> > I would appreciate it if you could refer to specific equations/paragraphs in related work or your paper in the discussion.

---

> ### Author Response · Authors · 2024-09-04
> **clarifications**
>
> 1. ```how can $\tilde p$, the range of which is $[0,1]$, be a valid “surrogate estimator” of $Y$ which is on $\mathbb{R}$? ```
>
> To summarize --- the methodology works for bounded outcomes without loss of generality, the code works for [0,1] so in the empirics we preprocess outcomes, dividing by the max range, and multiple predicted outputs by the max range. The methodology is _not_ limited to binary outcomes: It works for real-valued outcomes that are guaranteed finite. We agree this can be a source of confusion and revised the manuscript to clarify. It is quite common in theoretical machine learning to assume rewards (e.g, in reinforcement learning), etc, are bounded within $[0,1]$ (note that is the continuous interval) for standardization with the understanding that this holds without loss of generality for bounded variables.
>
> We uploaded this revision.
>
> First, let us explain the preprocessing steps. Because the ```mcboost``` package was developed initially for multi-calibration, which focused first on the case of calibration of probabilistic predictions $\in [0,1]$, some components of the package assume the input is $\in [0,1]$. This would be needed to create ``calibration buckets'' for multi-calibration auditing, not for multi-accuracy. Thus, this is totally without loss of generality for regression with bounded outcomes in our case. We simply pre-process the data, divide by the max range of outcomes, run multi-accuracy boosting, and report predictions that multiply back to the original range (see functions scale() and rev\_scale() in the R files, e.g. lines 156-162 in 1b\_simu.R).
>
> So, our method _in this paper_ is fully general for regression with _bounded outcomes_. Indeed Kim et al. (2022) studied outcome models with binary outcomes (voting, mortality risk). The code implementation we used assumes a particular range $[0,1]$ -- again without loss of generality from bounded outcomes.
>
> As written, we acknowledge the way we wrote Definition 1 was a bit too literal in the sense of reflecting the _code implementation_ limitations. This is likely confusing to an outside reader and we agree the definition should reflect our methodological generality.
>
> We have refactored _our writing_ in the following way: add assumption of boundedness, adjust Definition 1 to state $p \in [-B,B]$, add a preprocessing step in the algorithm explicitly that scales from [-B,B] to [0,1]. The uploaded revision reflects these changes.
>
> Again, these edits are in the presentation of the algorithmic procedure. Methodologically, the method works for bounded outcomes.
>
> Thank you for prompting this clarification in the presentation.
>
> 2. ```are the outcome prediction models you mentioned just $\tilde p$?```
>
> Yes. In this paper we typically reserve $\tilde{p}$ for the output of multi-accuracy post-processing.
>
> 3. ```But, if so, are the T-learner and DR-learner also restricted to binary outcomes?```
> *_No_*, see above: they work for bounded outcomes.
>
>
>
> The response window is fast closing. Please do let us know if we can provide any further clarifications.

---

> ### Author Response · Authors · 2024-09-04
> **further discussion of clarification**
>
> We would like to briefly include additional discussion on the generality of multi-accuracy post-processing for bounded outcomes.
>
> The primary context in which Multiaccuracy has been studied has been for binary prediction, but the techniques, algorithm, and analysis all generalize from the setting of discrete outcome prediction ${0,1}$ to real-valued probability prediction in $[0,1]$ to any bounded range $[-B,+B]$.  This fact is well-known in the community who studies Multiaccuracy, but admittedly, may not be very well-documented, particularly in our paper.  We will happily make this more explicit.  One reference, which handles the case of multiaccuracy and related notions of (moment) multicalibration for real-valued predictions, is that of "Moment Multicalibration for Uncertainty Estimation" of [Jung, Lee, Pai, Roth, Vohra; COLT/PMLR, 2021].
>
> Let us provide a self-contained explanation here.  Multiaccuracy requires that a predictor $p(x)$ passes all of a certain class of statistical tests.  These tests $E[(Y-p(X))*c(X)] < \epsilon$, require that $Y$ and $p(X)$ are close to one another in expectation on the range of $c(X)$.  So, $Y$ and $p(X)$ should live in the same space; if Y lives in $[0,1000]$, then so should $p(X)$.  We will correct all typographical errors that say that $p(X)$ lives in $[0,1]$, when Y lives in a larger range.
>
> To learn Multiaccurate predictors with a larger range, the algorithm and analysis are exactly the same.  The algorithm works by identifying a constraint defined by some $c \in C$ that is violated, and adding $\eta * c(x)$ to the predictor $p(x)$ for step size $\eta$.  The analysis shows that, in each update, the predictor $p(x)$ moves closer in squared distance to the true mean predictor $p*(x) = E[Y | X = x]$.  Of course, to learn a predictor $p$ of a random variable $Y$ that lives in a larger range $[-B,B]$ to the same absolute precision eps, the number of iterations will scale up accordingly (by a factor of $B^2$).  If you are ok with relative error of $\epsilon*B$, then the _rate_ of convergence is unaffected. It only affects the constants, in the same way that bounded outcomes affects rates of convergence elsewhere in learning theory
>
> We are happy to add an explicit discussion of this matter to the manuscript, in accordance with reviewer feedback.  We apologize that our typos seem to have led to conceptual confusions.
>
> Please let us know if there's anything additional we can clarify.

---

> > ### Comment · Reviewer_wUVC · 2024-09-05
> > **On bounded outcomes**
> >
> > Thanks for the replies. It makes sense that the method works only for bounded outcomes.
> >
> > But I am not sure if this is a "mild assumption", particularly in practice. First, without domain knowledge, we can only estimate the lower bound of $B$; and even with domain knowledge, it could be hard to judge if and how the outcome is bounded. Second, because of the imbalance ( $p(t=0|x)$ near 0 or 1 for some $x$ ), the data are too small to estimate the counterfactual range of the outcome; e.g., is it enough to *double* the bound of $Y$ estimated from data? I think we cannot answer this without further assumptions. This is even harder when your method aims to work under distribution shits.

---

> > > ### Author Response · Authors · 2024-09-05
> > > **clarification re bounded outcomes**
> > >
> > > We fully acknowledge that there can exist settings where the bounded outcomes assumption is not satisfied. We would like to clarify that what we mean by ``mild", as the term is used colloquially in theoretical ML, is that 1) the assumption imposes some restrictions, but is still relevant in a lot of settings, 2) is commonly assumed, 3) and relaxing it is more of a technicality than removing a conceptual bottleneck or core methodological issue.
> > >
> > > Re 1): In some applications, domain knowledge determines the bounds. Eg in our WHI example, the outcome was blood pressure, with known bounds from medical knowledge.
> > >
> > > 2): a number of foundational papers on causal inference also assume this, such as (but not limited to) Towards optimal doubly robust
> > > estimation of heterogeneous causal
> > > effects (Kennedy), Doubly Robust Policy Evaluation and
> > > Optimization (Dudik, Erhan, Langford, Li).
> > >
> > > 3): see our continuing response for an approach to relax the assumption.
> > >
> > > ```Second, because of the imbalance (  near 0 or 1 for some  ), the data are too small to estimate the counterfactual range of the outcome; e.g., is it enough to double the bound of  estimated from data? I think we cannot answer this without further assumptions.```
> > >
> > > Note that the overlap assumption implies that asymptotically the bounds can be determined by data. We think you raise valid finite sample concerns.
> > >
> > > Is doubling sufficient? Actually, we suggest an approach based on doubling + concentration inequalities will provide high-probability bounds on estimation of the upper (or lower) bound of outcomes. Consider the following procedure: if a predictor returns something at the upper bound, when there is not domain-level reason to believe in probability mass at the upper bounds, double until all predictions are at the interior of the upper bound. Then add an amount based on a concentration inequality for estimation of a maximum.
> > >
> > > Indeed, this is a separate technical problem, that could require some additional distributional assumptions such as subgaussianity. However, it is a technicality that would apply for problems of general regression with bounded outcomes, and is altogether separate from the conceptual contributions and focus of this paper.
> > >
> > > Finally, the assumption of bounded outcomes is also a mild technicality for the multi-accuracy algorithm. the only assumptions we need for the algorithm to converge are
> > > (a)  A bound on the expected squared distance between initial predictor $p_0(X)$ and $E[Y|X]$, where $p_0$ is the initial predictor we choose (arbitrarily or based on a prior)
> > > (b)  The ability to estimate statistical queries $E[c(x)*Y]$ from samples
> > > For (a), bounded magnitudes on $Y$ suffice, but are not necessary if we have some other auxiliary information. (Getting this additional assumption, such as a high-probability confidence bound on the estimation error is indeed harder under distribution shift, but possible under similar assumptions). For (b), we need some form of concentration; with bounded magnitudes, we get Chernoff-style convergence, but weaker forms of concentration would suffice as well, it would just affect the sample complexity.
> > >
> > >
> > > To summarize: we agree with you that sometimes there are settings where this assumption is violated. Relaxing it is a matter of independent interest separate from the contributions of this paper. We leave it for future work. Importantly, it _does hold in many applications, including the one in the paper_, with a blood pressure outcome.

---

> > > > ### Comment · Reviewer_wUVC · 2024-09-06
> > > >
> > > > Thank you, these make sense and I strongly suggest elaborating them with references (e.g., on "concentration inequality for estimation of a maximum" and "subgaussianity") in a future revision.

---

> > > > > ### Author Response · Authors · 2024-09-06
> > > > > **update**
> > > > >
> > > > > It looks like we are able to update with a revision. We added this discussion with pointers to classical references on the problem.

---

> ### Comment · Reviewer_wUVC · 2024-09-05
> **On identification**
>
> Although there are many different "identifications", at a high level they all say the following:
>
> Given a model and certain observational distribution, the estimand is in some sense determined (e.g., unique, confined to a countable set, bounded etc).
>
> In particular, there is no discussion of convergence rate which is a property of an estimator.
>
> And the identification under unconfoundedness says (there are other versions, of course):
>
> Given the model $Y=\mu(X, T)+\epsilon$ and $p(X, Y, T)$, under unconfoundedness, $\mu(x, t)=E(Y|x ,t)$.
>
> That's it! And it applies to all the settings in this paper, no? It seems to me that you confused identification with estimation (convergence in particular) and that what you refer to as "multi-accurate outcome models" are actually "multi-accurate outcome estimators".

---

> ### Author Response · Authors · 2024-09-05
> **clarification to on identification**
>
> ``` identification vs. estimation```
>
> Indeed our results blend both aspects of identification and estimation. What is surprising and interesting is that a particular estimation approach (multi-accuracy) that was motivated by pure regression-adjustment identification is equivalent to a doubly-robust estimator of a certain type, which mixes identification via regression adjustment and via IPW.
>
> We think a complete way to position our results relative to identification vs. estimation is that:
>
>   *robust identification + regression adjustment estimation with well-specified multi-accurate estimation = target-independent doubly-robust estimation*
>
>
> Prop 1 establishes that regression adjustment estimation with well-specified multi-accurate estimation = doubly-robust estimation. Prop 2 establishes that regression adjustment estimation with well-specified multi-accurate estimation = target-independent identification of the ATE. Putting these together yields the above.
>
> We will clarify this flow. We are now unable to upload revisions OpenReview. We will modify the first heading "Target-independent identification of the ATE under unconfoundedness'', which combines the conclusions of multiple sections, to *"Robust estimation of the ATE under unconfoundedness: regression adjustment with well-specified multi-accurate outcome predictors = doubly-robust estimation"*. However, when we discuss identification of the ATE under unknown distribution shifts, it is correct to refer to this as *identification* under distribution shift with the *unknown propensity score*. So we will clarify the next heading and change it from "Identification of the ATE under universal adaptability" to *"Robust target-independent identification and estimation of the ATE under universal adaptability"*.
>
> Overall indeed these topics are quite blended in the paper. We think adding the above summary will clarify. We will avoid emphasizing identification alone except in discussion of Prop 2.
>
> ```  Given the model $Y=\mu(X, T)+\epsilon$ and $p(X, Y, T)$, under unconfoundedness, $\mu(x, t)=E(p(Y \mid x, t))$ That's it! And it applies to all the settings in this paper, no? ```
>
> I'm sorry, no, we don't quite understand what you mean by this.
>
> Indeed, there are no underlying technical difficulties in Prop. 1: it applies the multi-accuracy definition and interprets it in the context of doubly-robust estimation. However, it does imply interesting properties, so we write it up as such.
>
> And, all our settings do use the same estimator $E[\tilde{p}(X)]$: what changes from setting to setting are simply the _assumptions on the multi-accuracy test function class_, which arise from whether we require properties of Prop. 1 (approximating a doubly-robust estimator), Prop. 2 (robust identification), or properties of multi-accuracy in interpreting test functions as indicators for subgroups.

---

> ### Comment · Reviewer_wUVC · 2024-09-06
>
> Even if under Setting 1 with the unknown $Q$, the outcome model (e.g., $Y=\mu(X, T)+\epsilon$) should still hold, and the difference is only how $X$ affects $T$, i.e., the propensity score. Indeed, a consistent estimator (e.g., regression of $Y$ on $(X, T)$) under $P$ should be consistent under $Q$; in both cases, the estimator converges to $\mu(x, t)=E(Y|x ,t)$, and this is what I meant by "the identification applies to all the settings in this paper". That is, the problems studied in this line of work are no problems under infinite data. However, what we care about here, for example, is *how fast* an estimator fitted on $P$ converges on $Q$ (under finite data), which is not about identification.

---

> ### Author Response · Authors · 2024-09-06
> **response**
>
> OK, we better understand what you mean.
>
> Yes, we _already_ discussed this in the paragraph right above Section 2.2, on page 6:
> > If suitably nonparametric CATE estimation indeed recovered the Bayes-optimal predictor in finite samples, there would be no issue of unknown deployment shifts. But because in finite samples it generally does not, modifying estimation to protect against unknown deployment shifts can protect against misspecification and finite-sample issues. For example, misspecified CATE estimation is vulnerable to unknown covariate shift. The conventional mean-squared error MSE can be nonzero for the Bayes-optimal predictor $\mu_1(X) = E[Y (1) |
> X]$. If the conditional bias or variance in Y is heteroskedastic (i.e. varies in x), the prediction MSE changes as external shifts change the marginalizing covariate distribution. Later on we will use multi-accurate learning to post-process CATE estimates to ensure robustness against covariate shifts represented by a function class of likelihood ratios.
>
> ```That is, the problems studied in this line of work are no problems under infinite data.```
>
> It's not a problem under infinite data under the true conditional expectation function. It is a problem under infinite data under a _misspecified_ outcome model, $\mu \neq \mu*$. See the discussion above. And it is a problem in finite data due to sample selection bias.
>
> We _already discussed this in the paper_ and also in the interpretation of the experiments, where we acknowledge our method is expected to perform better in small-sample regimes, which are nonetheless relevant.

---

> ### Comment · Reviewer_wUVC · 2024-10-09
> **Identification and Misspecification**
>
> AFAI understand, the quote is still discussing estimation because you say "mean-squared error MSE can be nonzero", which is a property of the estimator. Let me try to clear this up below.
>
> Setting: True model $\mathcal{P}:=\\{f_{\theta}:\theta \in \Theta\\}$ ($\theta$ includes the conditional outcome functional parameter $\mu$), Misspecified model $\mathcal{Q}:=\\{g_{\phi}:\phi \in \Phi\\}$ ($\phi$ includes the conditional outcome functional parameter $\nu$), True parameter $\theta^* \in \Theta$. Note that I used $f,g$ to explicitly indicate different parameterizations (and $\phi$ might not include any conditional outcome functional parameters, though I think this is not what you thought).
>
> Misspecification means $\mathcal{P}\neq\mathcal{Q}$ (which implies that it can be the case that $f_{\theta^*} \notin \mathcal{Q}$). Under misspecification, the main goal is usually to show that $\min_{\phi\in \Phi} D(f_{\theta^*}, g_{\phi}$) is small for some divergence $D$. I am not sure if you have such results in your paper, but this is not "identification" anyway.
>
> Now, you want to discuss "identification under misspecification". Frankly, I did not know such previous work but share my thoughts below.
>
> First, you could simply show that $\forall \phi_1, \phi_2 \in \Phi  (g_{\phi_1}=g_{\phi_2} \implies \phi_1=\phi_2)$, which means "the misspecified model is identifiable". But this is not very useful if we cannot bound the divergence mentioned above. So, on the line of your thought, I think we want something like the following:
>
> Preliminary step ("benign" misspecification): show that $f_{\theta^*} \in \mathcal{Q}$;
>
> Identification step: $\forall \phi_1, \phi_2 \in \Phi (g_{\phi_1}=g_{\phi_2}=f_{\theta^*} \implies \nu_1=\nu_2=\mu^*)$. A straightforward way to achieve this is to show that "the misspecified model is identifiable" *and* that $\forall \theta \in \Theta,\phi \in \Phi (g_{\phi}=f_{\theta} \implies \nu=\mu)$, but the latter is actually "the outcome model is well-specified on $\mathcal{P}\cap\mathcal{Q}$" in your terms, which makes the problem trivial.
>
> Finally, the bottom line is that I did not see any analysis like the above, but you proved convergence rates/error bounds, which, as I said repeatedly, are about estimation.

---

### Review · Reviewer_Gt65 · 2024-08-21

**Summary Of Contributions:**

This paper proposed a novel method for conditional average treatment effect estimation that is robust to unknown covariate shifts. The extensive experiments show the efficiency and excellent bias control of the proposed method under distribution shifts. This approach can also adapt to multiple kinds of covariate distribution shifts.

**Audience:**

Yes

**Claims And Evidence:**

Yes

**Requested Changes:**

Please refer to the Weaknesses.

**Strengths And Weaknesses:**

Strengths:
1. The robustness of unknown covariates shifts in CATE estimation is important and challenging, and it also has promising applications.
2. The proposed method is novel, combining the strengths of multi-accurate learning to ensure a low estimation bias.
3. The experiment evaluations are extensive, including a simulation study, a semi-synthetic case study, and an RCT experiment.
4. The authors employed a lot of examples to introduce their motivation, which is easy for readers to understand.


Weaknesses:
1. The related work discussion in the introduction is somewhat scattered.
2. It is advised to list contributions separately in the introduction.
3. It is suggested that the definition of the propensity score be moved to before the overlap assumption.
4. I suggest authors put the background in an individual section, rather than the Method section.
5. Since the paper aims to improve the robustness of CATE under covariate distribution shifts, I suggest the authors discuss the quantified criterion of distribution shift (e.g. Kullback-Leibler Divergence) in the simulations section.

---

> ### Author Response · Authors · 2024-08-29
> **Response to review**
>
> We thank the reviewer for their comments, which we have addressed as follows:
>
> - *Related work discussion in introduction* We have restructured the discussions, compressing discussions of related work in the discussion. See below where we introduce a new Background section. Now the only discussion of related work in introduction comprises of \textit{citations} that \textit{justify and make concrete} our motivations, as well as brief mentions of most closely related areas (CATE estimation and multi-accuracy).
>
> - *Separate listing of contributions*
> Sure, we have done this to clarify. We have added a new itemize environment to clarify contributions and go into more detail.
>
> - *Definition of propensity score to before overlap assumption*
> Thanks for noting; fixed and moved.
>
> - *Background in individual section*
> Sure: we have broken out the subsection ``Background on estimation" to its own section prior to Method, where we introduce our meta-algorithm.
>
> - *Quantification of shift*
> We have extended the simulation sections and quantify the distribution shifts in all simulation scenarios by presenting the KL divergence between the training (simulation 1a, 1b)/ RCT (simulation 2a, 2b) and test distributions in Tables 4--11.

---

### Author Response · Authors · 2024-08-29
**Response to all reviewers**

We thank the reviewers for recognizing the relevance of the problem of external validity and CATE, the methodological merits of connecting multi-accurate (or multi-distribution learning) with conditional average treatment effect estimation, and our extensive empirical assessment of a method new to the causal inference literature. Indeed, both of these areas have seen tremendous growth recently, though from different disciplines; our paper aims to provide common language and discussions relevant to either perspective.

We have uploaded a revision which includes further clarifications and expositions. We thank reviewers for noting particular points which could be confusing to readers with different backgrounds, and useful suggestions regarding writing. We have addressed this with additional self-contained explanations. Our revision thoroughly addresses these points and we thank the reviewers for prompting these improvements.

Reviewer wUVC has strong claims that the paper is fundamentally flawed. Unfortunately, Reviewer wUVC's assessment is based on factually incorrect statements: indeed Kim et al. (2022) also applies multi-accurate post-processing to *outcome models*. We believe this misunderstanding could arise from us understating the novelty of the identification perspective we put forth, where the multi-accurate *outcome model* approximates a *doubly-robust model* (hence with propensity score), even though the original estimator doesn't include the propensity score explicitly. (The connection arises via the multi-accuracy procedure and the auditing/test functions). To prevent similar future misunderstandings, we have added extensive clarifications both in describing our estimator and after Proposition 1.

---

### Comment · Reviewer_wUVC · 2024-10-08
**Post rebuttal update**

It seems I cannot update my main review now, so I added a comment.

After the discussion, I admit my initial evaluation that the paper "is fundamentally flawed" was too strong. However, some concerns remain:

- The paper didn’t mention the essential assumption of a bounded outcome, which was also the main reason I made the initial evaluation. Even after the revision and discussion, I feel this point is not elaborated enough.
- There is confusion on the fundamental concept of identification, and the author(s) confuse identification with convergence analysis and "multi-accurate outcome estimators" with "multi-accurate outcome models". (Please see my latest (and final) reply)
- The writing is not good as per my reading. I strongly suggest the author(s) have another 1 or 2 pass over the paper to polish it, particularly in the latter half.

I hope that the author(s) could make an effort to improve the final version and that the merits of the work override my concerns.

Congrats on the acceptance! And thanks to the AC and other reviewers for their work.

---

### Decision · Action_Editor_iAHJ · 2024-09-29

**Recommendation:** Accept with minor revision

**Comment:**

First, let me thank the reviewers (particularly wUVC) and authors for the fruitful discussion, which has notably improved the submission.

After reviewing the submission, the reviewers' comments, and the authors' responses, I recommend acceptance with minor revision.
The authors revised their submission according to the reviewers' comments, convincingly addressed concerns raised during the response period, and improved their submission. Any remaining issues can addressed in the final revision.

Specifically, given the lively discussion, I request the authors

* have one more pass over the manuscript to ensure any remaining inconsistencies are resolved (e.g. just after Assumption 4 a $\mathcal{D}_\text{o}$ has snuck in instead of $\mathcal{D}_\text{obs}$);
* add an outline to the paper in §1 and also an outline to the Method section (§4) to give an overview of their exposition;
* have another pass over the paper to ensure that concepts are explained or referenced for general readers in machine learning (estimation vs identification, etc) and assumptions are clear.

I'm looking forward to the camera-ready version of the paper. It makes a valuable contribution with practical implications and theoretical insights that will be of interest to the TMLR audience.

**Audience:**

Yes. Covariate shifts are an important concern to applying causal inference "in the wild." Making causal inference methods more robust to covariate shifts with theoretical guarantees is important to researchers and practitioners.

**Claims And Evidence:**

Yes. The submission proposes a method that improves the robustness of Conditional Average Treatment Effect (CATE) estimators under unknown covariate shifts using recently proposed approaches on multi-accurate predictors. The empirical evaluation includes simulations with varying amounts of covariate shift and a case study using real-world data. Concerns raised by the reviewers have been addressed in the revisions.

---

> ### Author Response · Authors · 2024-10-15
>
> Thank you very much for the comprehensive efforts to AE the submission and your expertise and judgment. We are delighted to resubmit this minor revision, which addresses your crucial and thoughtful suggestions. We have uploaded a revision addressing these final expository concerns. (Diff omitted due to small changes throughout).
>
>
> Responses to AE requests:
>
> Thanks for pointing out, we have addressed this and other typographical inconsistencies.
>
> Re outline in  §1  and outline in Method,  §4: we have added these, the new last paragraph of  §1 and a new preface to  §4 (3 paragraphs) walking through the discussion of the results to come.
>
> Re explaining concepts for general readers: We have done this throughout. We have added additional references to classical notions in covariate shift literature (Shimodaira, 2000; Buja et al., 2019) connecting model misspecification to dependence of the population minimizer (i.e, infinite-data) of the MSE on the covariate distribution, i.e. vulnerability to covariate shift (p 6). We do choose to continue with conventions in the machine learning and causal inference literature that an "outcome model" is another term for "estimator". We believe these issues of terminology are minor, stylistically grounded in the relevant subfields. We have made sure to inline explanations for terminology otherwise, balancing length with a self-contained presentation.
>
> We greatly appreciate the efforts and comments from the review team, which have helped us improve the presentation of our work.